# ON DIFFUSION MODELING FOR ANOMALY DETECTION

**Victor Livernoche** [1 3 *]   **Vineet Jain**[1 3 *]   **Yashar Hezaveh**[2 3]   **Siamak Ravanbakhsh**[1 3]

[1] School of Computer Science, McGill University
[2] Department of Physics, University of Montreal
[3] Mila - Quebec AI Institute
* Equal contribution

## ABSTRACT

Known for their impressive performance in generative modeling, diffusion models are attractive candidates for density-based anomaly detection. This paper investigates different variations of diffusion modeling for unsupervised and semi-supervised anomaly detection. In particular, we find that Denoising Diffusion Probability Models (DDPM) are performant on anomaly detection benchmarks yet computationally expensive. By simplifying DDPM in application to anomaly detection, we are naturally led to an alternative approach called Diffusion Time Estimation (DTE).[1] DTE estimates the distribution over diffusion time for a given input and uses the mode or mean of this distribution as the anomaly score. We derive an analytical form for this density and leverage a deep neural network to improve inference efficiency. Through empirical evaluations on the ADBench benchmark, we demonstrate that all diffusion-based anomaly detection methods perform competitively for both semi-supervised and unsupervised settings. Notably, DTE achieves orders of magnitude faster inference time than DDPM, while outperforming it on this benchmark. These results establish diffusion-based anomaly detection as a scalable alternative to traditional methods and recent deep-learning techniques for standard unsupervised and semi-supervised anomaly detection settings.

## 1 INTRODUCTION

Anomaly detection seeks to identify observations that differ from the others to such a large extent that they are likely generated by a different mechanism (Hawkins, 1980). This is a longstanding research problem in machine learning with applications in various fields ranging from medicine (Pachauri & Sharma, 2015; Salem et al., 2013), finance (Ahmed et al., 2016b), security (Ahmed et al., 2016a), manufacturing (Susto et al., 2017), particle physics (Fraser et al., 2022) and geospatial data (Yairi et al., 2006). Despite its significance and potential for impact (e.g., leading to the discovery of new phenomena), to this day traditional anomaly detection methods, such as nearest neighbours, reportedly outperform deep learning techniques on various benchmarks (Han et al., 2022) by a significant margin. This is true for unsupervised, semi-supervised, and supervised anomaly detection tasks. However, the growing number of applications involving high-dimensional data and massive datasets are beginning to challenge the classical, and in particular non-parametric, techniques, and there is a need for scalable, interpretable, and expressive deep learning techniques for anomaly detection.

In recent years, denoising diffusion probabilistic models (DDPMs) (Ho et al., 2020) have received much attention as a powerful class of generative models. While these models have been successfully utilized for anomaly detection in domain-specific image datasets (Wolleb et al., 2022; Zhang et al., 2023a; Wyatt et al., 2022), a comprehensive exploration of their applicability for general-purpose anomaly detection across diverse tabular, image, and natural language datasets is notably absent.

Our starting point is the observation that DDPM exhibits competitive performance compared to previous approaches for unsupervised and semi-supervised anomaly detection. These are some of

---

[1]Code available at https://github.com/vicliv/DTE

the most challenging settings, where either an unlabelled mix of normal and anomalous samples are available for training or, at best, the training data only includes normal samples. However, the expressivity and interpretability of DDPM come with a considerable computational cost. This computational complexity poses challenges for anomaly detection tasks involving large datasets or data streams.

In anomaly detection using DDPM, we deterministically "denoise" the input and measure the distance to its denoised reconstruction; a large distance indicates an anomaly. Since we only use this distance for outlier identification, in order to reduce the complexity of the diffusion-based approach, we propose to directly estimate this distance, which is correlated with diffusion time.

More precisely, we estimate the posterior distribution of diffusion time (or noise variance) for a given input. This estimated distribution serves as a guide for identifying anomalies, as they are anticipated to exhibit higher posterior density at larger time steps compared to normal samples. In particular, we use the mode or mean of this distribution as the anomaly score. We derive an analytical form for this posterior distribution, enabling its non-parametric estimation. We see that the non-parametric approximation produces a ranking for anomalies that is identical to k-Nearest Neighbours (kNN) for anomaly detection. We then propose a parametric model, a deep neural network, allowing us to leverage the generalization capability and efficient inference time of deep learning.

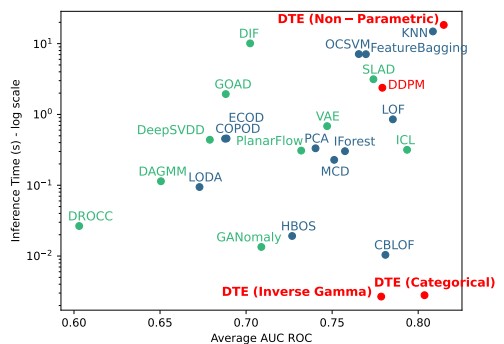

Figure 1: Average inference time vs. average AUC ROC for all 57 ADBench datasets in the semi-supervised setting. Lower right is better (DTE Categorical). Colour scheme: red (diffusion-based), green (deep learning), blue (classical).

We provide an extensive evaluation compared to classical and other deep models for different anomaly detection settings on more than 57 datasets from ADBench (Han et al., 2022). Our empirical results suggest that using a single deep neural network architecture across all datasets and settings makes the diffusion model competitive with classical and other deep models. Figure 1 shows the efficiency and effectiveness of different anomaly detection algorithms across all datasets in ADBench. Notably, our proposed method surpasses the direct application of DDPMs, achieving substantial improvements in inference time.

The contributions of our work are summarized as follows:

- Evaluation of denoising diffusion probabilistic models on various anomaly detection tasks encompassing tabular data and embeddings of images and natural language datasets.
- Development of a simplified approach that models the posterior distribution over diffusion time as a proxy for anomaly detection.
- Derivation of an analytical form of the posterior distribution of diffusion time and development of a non-parametric estimator that leads us to kNN.
- Introduction of a parametric approach utilizing a deep neural network for improved generalization and scalability.
- Implementation of additional baselines and extensive evaluation on 57 datasets from AD-Bench, showcasing competitive performance compared to classical and existing deep-learning-based anomaly detection algorithms.
- Investigation into the interpretability of diffusion-based methods, including our novel approach, highlighting their strengths and limitations.
- Exploration of optimal representation selections for image datasets with diffusion methods.

## 2 PRELIMINARIES

A classification of anomaly detection methods is based on the availability of labelled data. *Supervised* setting is similar to binary classification with unbalanced classes since the number of anomalies

in the data is generally a small fraction of the total number of samples. This setup is limited to the identification of known anomalies. The more challenging *unsupervised* setting assumes that the data is a mix of normal and anomalies, without access to labels. Methods in this category often make assumptions about the data-generation process. Therefore, embedding techniques and deep generative models are prime candidates. However, a challenge for deep models is the fact that they tend to model the anomalies within the input data more easily, making the task of identifying them harder. A middle ground between supervised and unsupervised is *semi-supervised* or *one-class classification* setting, where one has access to purely normal samples during training, yet anomalies of unknown nature can exist at inference time. Perhaps confusingly, the term semi-supervised is also used when partial labelling of anomalies is available during the training. In this work, we are interested in identifying anomalies with an unknown distribution and therefore do not assume access to any label information for outliers. That is we consider both unsupervised and the one-class classification version of semi-supervised anomaly detection.

## 2.1 Diffusion Probabilistic Models

A diffusion process is a stochastic process characterized by a probability distribution that evolves over time, governed by the diffusion equation. Diffusion probabilistic models (Sohl-Dickstein et al., 2015; Ho et al., 2020) are latent variable probabilistic models where the state at time steps larger than zero are considered latent variables. Let $\mathbf{x}_0 \sim q(\mathbf{x}_0)$ denote the data and $\mathbf{x}_1, \ldots, \mathbf{x}_T$ denote the corresponding latent variables. The forward diffusion process is generally fixed to add Gaussian noise at each timestep according to a variance schedule $\beta_1, \ldots, \beta_T$. The approximate posterior $q(\mathbf{x}_{1:T} \mid \mathbf{x}_0)$ is given by,

$$q(\mathbf{x}_{1:T}|\mathbf{x}_0) := \prod_{t=1}^{T} q(\mathbf{x}_t|\mathbf{x}_{t-1}), \qquad q(\mathbf{x}_t|\mathbf{x}_{t-1}) := \mathcal{N}(\mathbf{x}_t; \sqrt{1-\beta_t}\mathbf{x}_{t-1}, \beta_t\mathbf{I}) \tag{1}$$

Choosing the transitions as Gaussian distributions enables sampling $\mathbf{x}_t$ at any time in closed form. Let $\alpha_t := 1 - \beta_t$ and $\bar{\alpha}_t := \prod_{s=1}^{t} \alpha_s$, then,

$$q(\mathbf{x}_t|\mathbf{x}_0) := \mathcal{N}(\mathbf{x}_t; \sqrt{\bar{\alpha}_t}\mathbf{x}_0, (1-\bar{\alpha}_t)\mathbf{I}). \tag{2}$$

Diffusion probabilistic models then learn transitions that reverse the forward diffusion process. Starting at $p(\mathbf{x}_T) = \mathcal{N}(\mathbf{x}_T; 0, \mathbf{I})$, the joint distribution of the reverse process $p_\theta(\mathbf{x}_{0:T})$ is given by,

$$p_\theta(\mathbf{x}_{0:T}) := p(\mathbf{x}_T) \prod_{t=1}^{T} p_\theta(\mathbf{x}_{t-1}|\mathbf{x}_t), \qquad p_\theta(\mathbf{x}_{t-1}|\mathbf{x}_t) := \mathcal{N}(\mathbf{x}_{t-1}; \boldsymbol{\mu}_\theta(\mathbf{x}_t, t), \boldsymbol{\Sigma}_\theta(\mathbf{x}_t, t)) \tag{3}$$

This parameterized Markov chain also called the reverse process, can produce samples matching the data distribution after a finite number of transition steps.

## 3 Diffusion Time Estimation

Denoising diffusion probabilistic models (DDPM), as introduced in (Ho et al., 2020), can be used to generate samples matching the data distribution even in high-dimensional spaces. The reverse diffusion process implicitly learns the score function of the data distribution and can be used for the likelihood-based identification of anomalies. A common approach used in prior works on anomaly detection using diffusion models (Wolleb et al., 2022; Zhang et al., 2023a; Wyatt et al., 2022) is to reconstruct input samples by simulating the reverse diffusion chain and then using the reconstruction distance to identify anomalies. This is particularly useful where anomalies are localized in the image, and the difference between the input and its reconstruction identifies this localized anomaly. While all previous works focus on this scenario in image data, we consider the broader problem of identification of anomalous samples without assumptions on data type or the nature of the anomaly.

Toward this objective, we evaluate the reconstruction-based approach using DDPMs on the AD-Bench benchmark, which comprises 57 datasets, including tabular, image, and natural language data. We observe that the choice of timestep at the start of reverse diffusion is arbitrary, yet it can significantly affect the anomaly detection performance. We found that using 25% of the maximum timestep globally leads to good results; see the Appendix A for an ablation.

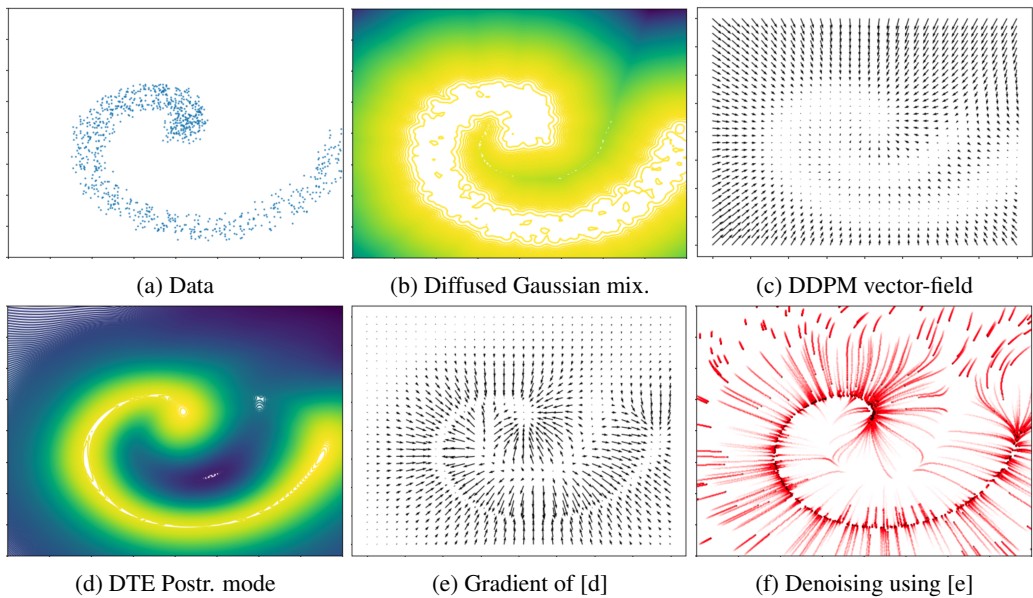

(a) Data          (b) Diffused Gaussian mix.          (c) DDPM vector-field

(d) DTE Postr. mode          (e) Gradient of [d]          (f) Denoising using [e]

Figure 2: DDPM and DTE on a toy dataset shown in (a). (b) shows the Gaussian density function associated with the lowest timestep of DDPM and (c) shows the vector field corresponding to the gradient of this density. (d) plots the mode of the DTE posterior distribution over diffusion time, which we show in subsequent sections is an inverse Gamma distribution. (e) shows the gradient of (d), and (f) shows the flow associated with this gradient, showing that random samples are mapped toward the data manifold.

As anticipated, the expressivity of these models allows them to perform competitively compared to prior work. However, inference for a single data point involves simulating the reverse diffusion chain in its entirety, making this approach computationally expensive. By quantifying the disparity between the reconstructed output and the original input, the objective is to effectively capture the deviations of anomalous samples from the underlying data manifold. We contend that modeling the score function by learning the reverse process is unnecessary if the objective is only the identification of anomalies.

Building upon this idea, we propose a much simpler approach that does not require modeling the reverse diffusion process but instead models the distribution over diffusion time corresponding to noisy input samples. Assuming anomalies are distanced from the data manifold, the density for larger timesteps should have a higher value for anomalies, enabling their probabilistic identification. This can be seen as a direct estimation of reconstruction error.

More concretely, we simulate anomalous samples using a diffusion process and train a neural network to predict the diffusion time corresponding to the noisy samples. Provided that the noisy samples cover the entire feature space, this procedure should also capture potential anomalies. Figure 2 contrasts DDPM and DTE on a toy dataset. The success of our method in using diffusion for anomaly detection is due to the space-filling property of the diffusion process; different regions of the space are sampled at different rates, depending on their proximity to the data manifold. To our knowledge, this is the first setting that uses this property of diffusion beyond its application in learning time-dependent score functions for generative modelling. While in that setting, the estimated score is able to meaningfully approximate the true score over the entire space, we show that we are able to approximate the diffusion time for arbitrary points, including normal or anomalous points.

### 3.1 POSTERIOR DISTRIBUTION OF DIFFUSION TIME

Assuming $\mathbf{x}_s \in \mathbb{R}^d$ is produced through a diffusion process, starting from the data manifold, our goal in this section is to identify the distribution over its diffusion time, as a surrogate for its distance from the manifold. The diffusion process described by Equation (2) specifies a distribution corresponding to each timestep. First, let us assume the dataset consists of a single data point at the

origin. Denote the variance at time $t$ as $\sigma_t^2 = 1 - \bar{\alpha}_t$, and consider the $d$-dimensional zero mean Gaussian distribution at each timestep $\mathcal{N}(\mathbf{0}, \sigma_\mathbf{t}^2)$. The posterior distribution over $\sigma_t^2$ given $\mathbf{x}_s$ is:

$$p(\sigma_t^2|\mathbf{x}_s) \propto p(\mathbf{x}_s|\sigma_t^2)\, p(\sigma_t^2) = \mathcal{N}(\mathbf{x}_s; \mathbf{0}, \sigma_\mathbf{t}^2) \propto \sigma_\mathbf{t}^{-\mathbf{d}} \exp\left(-\frac{\|\mathbf{x_s}\|^2}{2\sigma_\mathbf{t}^2}\right)$$

This is an *inverse Gamma distribution* $p(\sigma_t^2; a, b) = \frac{b^a}{\Gamma(a)}\left(\frac{1}{\sigma_t^2}\right)^{a+1}\exp\left(-\frac{b}{\sigma_t^2}\right)$ with parameter values $a = d/2 - 1$ and $b = \|\mathbf{x}_s\|^2/2$.

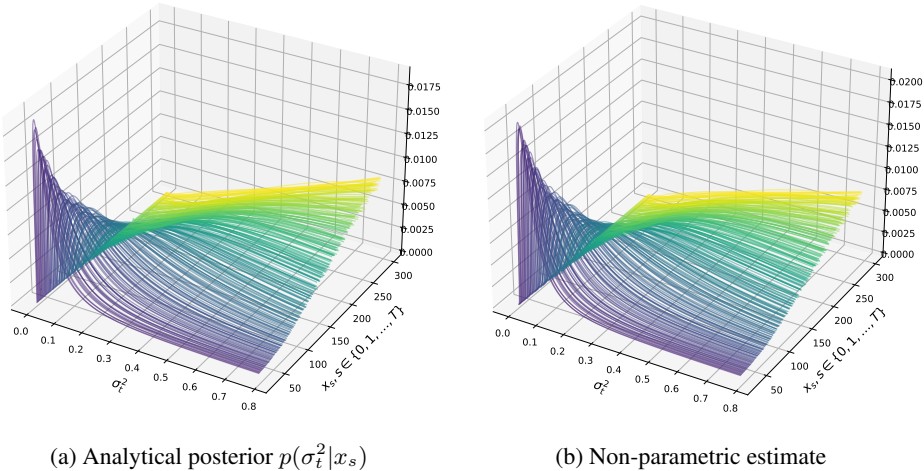

(a) Analytical posterior $p(\sigma_t^2|x_s)$          (b) Non-parametric estimate

Figure 3: Posterior timestep distribution $p(\sigma_t^2|\mathbf{x}_s)$, where $\mathbf{x}_s$ is produced using diffusion with different time steps $s \in \{1, \ldots, T\}$, averaged over the `vertebral` dataset. (a) shows the analytical distribution computed by placing Gaussian distributions of different variances at each point in the dataset, and (b) shows the inverse Gamma distribution with scale parameter value depending on the average distance to the k-nearest neighbours ($k = 32$).

If instead of a single data point at the origin, we have a dataset $\mathcal{D}$, with the corresponding data distribution $p(\mathbf{x})$, we have

$$p(\sigma_t^2|\mathbf{x}_s) \propto p(\mathbf{x}_s|\sigma_t^2)p(\sigma_t^2) = \sum_{\mathbf{x}_0} p(\mathbf{x}_s|\mathbf{x}_0, \sigma_t^2)p(\mathbf{x}_0) = \sum_{\mathbf{x}_0 \in \mathcal{D}} \mathcal{N}(\mathbf{x}_s; \mathbf{x}_0, \sigma_t^2\mathbf{I}). \qquad (4)$$

We refer to Equation (4) as the analytic estimator in subsequent sections since it is the exact posterior distribution. The posterior distribution can be interpreted as adding the likelihoods of Gaussian distributions centered around data points $\mathbf{x}_0 \in \mathcal{D}$ with different (time-dependent) variances. Substituting the Gaussian density function and simplifying, we get

$$p(\sigma_t^2|\mathbf{x}_s) \propto \sum_{\mathbf{x}_0 \in \mathcal{D}} \sigma_t^{-d} \exp\left(-\frac{\|\mathbf{x}_s - \mathbf{x}_0\|^2}{2\sigma_t^2}\right) = \sigma_t^{-d} \exp\left(\log\left(\sum_{\mathbf{x}_0 \in \mathcal{D}} \exp\left(-\frac{\|\mathbf{x}_s - \mathbf{x}_0\|^2}{2\sigma_t^2}\right)\right)\right).$$

We can approximate the log-sum-exp term using `max` function:

$$p(\sigma_t^2|\mathbf{x}_s) \gtrsim \sigma_t^{-d} \exp\left(\max_{\mathbf{x}_0 \in \mathcal{D}} -\frac{\|\mathbf{x}_s - \mathbf{x}_0\|^2}{2\sigma_t^2}\right) = \sigma_t^{-d} \exp\left(-\frac{1}{\sigma_t^2}\min_{\mathbf{x}_0 \in \mathcal{D}}\frac{\|\mathbf{x}_s - \mathbf{x}_0\|^2}{2}\right) \qquad (5)$$

The posterior over diffusion time approximately has the form of an inverse Gamma distribution with the shape parameter $a = d/2 - 1$ depending only on the dimensionality of the data and the scale parameter $b = \min_{\mathbf{x}_0 \in \mathcal{D}}\frac{\|\mathbf{x}_s - \mathbf{x}_0\|^2}{2}$ depending on the distance of the input point to the closest point in the dataset. Note that, as $a > 0 \implies d > 2$, this analysis is only valid for three or higher dimensions.

## 3.2 NON-PARAMETRIC MODEL

The posterior over diffusion time given by Equation (5) can potentially be used as a non-parametric approach to anomaly detection. The approximation of log-sum-exp using the maximum value (nearest neighbour) becomes less accurate for larger timesteps, in which a point has a comparable distance to several points in the dataset. We found that instead of setting the scale parameter $b$ based on the distance to the closest point, approximating log-sum-exp using the average distance to k-nearest neighbours of the input point works better in practice. The non-parametric estimator is then:

$$p(\sigma_t^2|\mathbf{x}_s) \propto \sigma_t^{-d} \exp \left( -\frac{1}{\sigma_t^2} \cdot \frac{1}{K} \sum_{\mathbf{x}_0 \in \text{kNN}(\mathbf{x}_s)} \frac{\|\mathbf{x}_s - \mathbf{x}_0\|^2}{2} \right) \quad (6)$$

Figure 3 shows the analytical posterior distribution obtained using Equation (4) and the non-parametric estimator given in Equation (6) for a real dataset.

The upshot is that, given a point $\mathbf{x}_s$, this method approximates the scale parameter of the inverse Gamma distribution using the average distance to its $k$-nearest neighbours. The anomaly score is the mean of this distribution over diffusion time. As seen in Figure 3, points $\mathbf{x}_s$ that are produced using diffusion with larger time-steps also have a higher posterior mean, on average, enabling us to identify them as points that are far from the manifold. Interestingly, this method closely resembles the classical $k$-nearest neighbours (kNN). In fact, the anomaly rankings given by these methods are identical. In our experiments, the difference in score comes from the distance calculation: for DTE non-parametric, we take the mean distance from the k-nearest neighbours as opposed to (a variation of) kNN that takes the distance from the kth-nearest neighbour.

## 3.3 PARAMETRIC MODEL

The non-parametric estimator of diffusion time becomes compute and memory-intensive when dealing with large datasets due to the need to find the k-nearest neighbours for each input sample in the entire dataset. To tackle the scalability problem, we employ deep neural networks to estimate the posterior distribution, which also enhances generalization capabilities. The full training procedure for both parametric models is available in Appendix D.2.

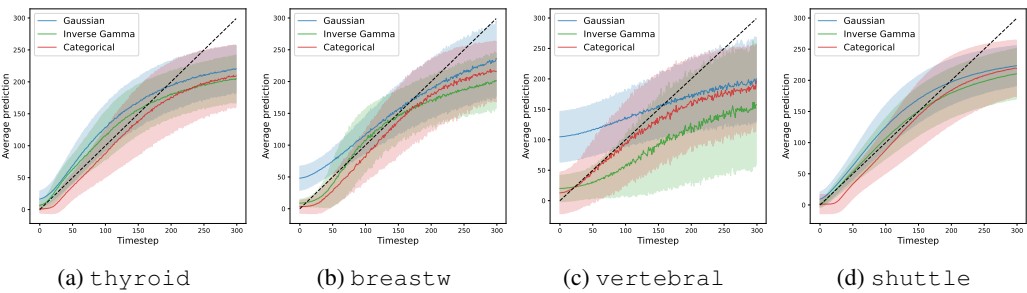

(a) `thyroid`   (b) `breastw`   (c) `vertebral`   (d) `shuttle`

Figure 4: Predicted diffusion time against ground truth diffusion time for Gaussian model ($\ell_2$-regression), Inverse Gamma model, and categorical model (with seven bins) on the test set for various datasets. The maximum length of the diffusion Markov chain is $T = 300$. The shaded region indicates the standard deviation in predictions across the dataset.

**Inverse Gamma model** In Section 3.1 we saw that the posterior distribution over time-dependent variance has the form of an inverse Gamma distribution. We train a deep neural network parameterized by $\theta$, which we denote by $f_\theta$, to predict the scale parameter $b$ of the inverse Gamma distribution, given the noisy sample $\mathbf{x}_t$. Since the shape parameter $a$ depends only on the dimensionality of the data, it is a known fixed parameter. We minimize the negative log-likelihood given by:

$$\mathcal{L}(\theta) := -\mathbb{E}_{t,\mathbf{x}_0} \left[ a \log f_\theta(\mathbf{x}_t) - (a+1) \log \sigma_t^2 - f_\theta(\mathbf{x}_t)/\sigma_t^2 \right] \quad (7)$$

The expectation is over data samples $\mathbf{x}_0 \sim p(\mathbf{x})$ and timesteps $t \sim \mathcal{U}[1, T]$. The mode of the distribution is used as anomaly score.

Figure 4 shows the predicted timestep for the inverse Gamma model applied to different datasets, with the length of Markov chain $T = 300$. Compared to standard $\ell_2$ regression which assumes that the output variable is Gaussian distributed, the inverse Gamma model has a much lower bias for diffusion time prediction for smaller timesteps, which empirically validates our analysis. However, this model suffers from high bias and high variance for larger timesteps. The high bias can be attributed to the approximation error of log-sum-exp using k-nearest neighbours, which becomes inaccurate for larger timesteps. The high variance is a consequence of the shape of the inverse Gamma distribution, which becomes flat for large values of the scale parameter (see Figure 3).

**Categorical model**    The inverse Gamma model while analytically accurate, can restrict the expressivity of the neural network. In order to provide more flexibility in learning the diffusion time distribution, we can model it as a categorical distribution over $T$ classes, where $T$ is the length of the Markov chain associated with the diffusion process. This approach does not assume any parametric distribution over diffusion time and requires the model to accurately predict the full distribution. Let $y_t \in \{0, 1\}^T$ denote the one-hot vector with one at coordinate $t$, and $f_\theta$ denote the deep neural network that predicts the class probabilities, $f_\theta : \mathcal{X} \to [0, 1]^T$. We minimize the cross-entropy loss function, which is equivalent to maximizing the log-likelihood of the categorical distribution:

$$\mathcal{L}(\theta) := \mathbb{E}_{t,x_0} \left[ -\sum_{k=0}^{K} y_t^{(k)} \log \left( f_\theta(\mathbf{x}_t)^{(k)} \right) \right] \tag{8}$$

In practice, we simplify the learning task by combining timesteps into bins and training a model to predict the correct bin. If $B$ denotes the number of bins, then the corresponding bin for a timestep $t$ would be $\lfloor \frac{t \cdot B}{T} \rfloor$. Figure 4 shows the predicted timestep for the categorical model on different datasets. Compared to the inverse Gamma model, it suffers from significantly less bias across the entire range of timesteps. The score calculation is described in Appendix D.3 with the training algorithm in Appendix D.2.

## 4    EXPERIMENTS

**Setting**    We perform experiments on the ADBench benchmark (Han et al., 2022), which comprises a set of popular tabular anomaly detection datasets as well as newly created tabular datasets made from images and natural language tasks, all described in Appendix D.1. The implementation details are provided in Appendix D, with the training algorithm, model architecture, hyperparameters, and comparison of the run-time. Some ablation studies are in Appendix A. We implement and compare the results of the various approaches proposed in Section 3: the non-parametric, the parametric inverse Gamma, and the parametric categorical DTE.

**Baselines**    We compare against all the unsupervised learning methods included in ADBench. These include classical methods, namely CBLOF (He et al., 2003), COPOD (Li et al., 2020), ECOD (Li et al., 2022), FeatureBagging (Lazarevic & Kumar, 2005), HBOS (Goldstein & Dengel, 2012), IForest (Liu et al., 2008), kNN (Ramaswamy et al., 2000), LODA (Pevný, 2016), LOF (Breunig et al., 2000), MCD (Fauconnier & Haesbroeck, 2009), OCSVM (Schölkopf et al., 1999), and PCA (Shyu et al., 2003). The deep learning-based methods include DeepSVDD (Ruff et al., 2018), and DAGMM (Zong et al., 2018). Outside of ADBench, we also compare against some more recently proposed deep learning-based approaches such as DROCC (Goyal et al., 2020), GOAD (Bergman & Hoshen, 2020), ICL (Shenkar & Wolf, 2022), SLAD (Xu et al., 2023b) and DIF (Xu et al., 2023a); see Section 5 for a brief overview. For each method, we picked the best-performing set of hyperparameters given in their original paper. We also have four additional generative baselines: normalizing flows with planar flows (Rezende & Mohamed, 2015) to identify anomalies based on the log-likelihood, DDPM, VAE (Kingma & Welling, 2013) and GAN (Goodfellow et al., 2014) to reconstruct the input and compare it with the original input to identify anomalies.

**Results**    Figure 5 shows the overall performance of these different methods on 57 tasks in AD-Bench, each limited to 50,000 data points. The results for each individual dataset are provided in Appendix F. We report the mean AUC ROC and its standard deviation over five different seeds for each method. For the unsupervised setting, we used bootstrapping over the whole dataset for training, while inference is made on the full dataset. For the semi-supervised setting, we used 50% of

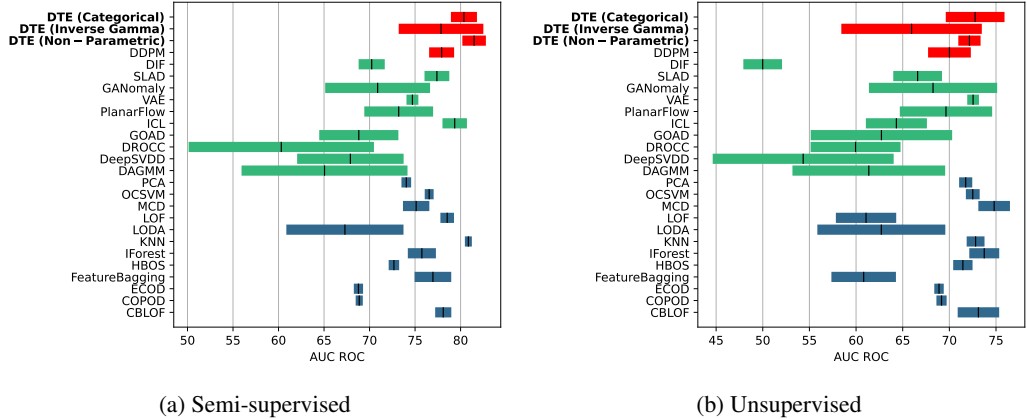

(a) Semi-supervised          (b) Unsupervised

Figure 5: AUC ROC means and standard deviations on the 57 datasets from ADBench over five different seeds for a) the semi-supervised setting using normal samples only for training and b) the unsupervised setting with bootstrapped training instances. Colour scheme: red (diffusion-based), green (deep learning methods), blue (classical methods). DTE outperforms all baselines for the semi-supervised setting apart from kNN. It is also competitive in the unsupervised setting.

the normal samples for training, while the test set contains the rest of the normal samples and all anomalous samples. The proposed method is among the few competitive in both semi-supervised and unsupervised settings. In particular, our method outperforms all previous deep learning-based approaches in both settings significantly and also outperforms the DDPM model. Unsurprisingly, deep learning methods have a higher variance than non-parametric methods. Using bagging can be a way to help reduce the variance at the cost of more training and inference time.

Figure 1 compares our method's performance and inference time with the other baselines. In some applications, such as medical and network monitoring, fast inference time is crucial as the algorithm must detect anomalies in real time. Our method uses a forward pass through a simple neural network for predictions, which gives it the shortest inference time over all the methods considered here. Training time, inference time and compute amounts are available in Appendix D.4.

**Choice of representation** ADBench's image datasets use vector representation derived from pre-trained ImageNet embeddings. We investigated the impact of representation quality for semi-supervised anomaly detection across several datasets: VisA (Zou et al., 2022), CIFAR-10, and MNIST. We observe that different methods, including DDPM, kNN and DTE, perform better when applied to image embeddings rather than raw images. In particular, embeddings produced through self-supervision are generally of higher quality when compared to those produced for classification, and the embeddings that are specialized or fined-tuned to the target dataset produce the best results. The results are reported in Appendix E.

We also observe that kNN remains a top-performing algorithm for anomaly detection, where its only disadvantage remains its scalability. As explained in Section 3.2, the non-parametric method gives the same anomaly ranking as kNN. DTE can thus be approximately interpreted as a parametric $k$-nearest neighbours algorithm which can be beneficial for large datasets that require smaller inference time. To understand the anomalies, both DDPM and DTE are able to identify a "denoised" data point; DDPM depends on an initial time step hyper-parameter, whereas DTE does not, by using deterministic ODE flow. However, DDPM outperforms in denoising, being explicitly trained for it. Further interpretability discussion, illustrated with a toy example, is in Appendix B.

## 5 RELATED WORK

We refer the reader to the following surveys for a comprehensive review (Pang et al., 2021; Chandola et al., 2009; Ruff et al., 2021; Hodge & Austin, 2004). Although recently, the spotlight has shifted towards deep learning methodologies, classical techniques such as kNN (Ramaswamy et al., 2000)

persistently exhibit strong performance. We compared our method with some of these techniques in Section 4. Clustering and nearest neighbour algorithms use the distance to score instances, making them easily interpretable. Clustering algorithms, such as CBLOF (He et al., 2003) and k-means (MacQueen, 1967), assume that anomalies are either not part of cluster, are part of smaller clusters than normal instances, or lie further away from the cluster centroid. In contrast, nearest neighbour algorithms use the distance between points or relative density with respect to their neighbourhood.

As anomalies can be more difficult to detect in high-dimensional spaces and complex data distributions (Pang et al., 2021), the development of deep anomaly detection algorithms has been increasing over the past few years (Ruff et al., 2021). In particular, several works combine autoencoders with other classical techniques (Zhou & Paffenroth, 2017; Kim et al., 2020; An & Cho, 2015; Erfani et al., 2016; Sakurada & Yairi, 2014; Xia et al., 2015). Other notable methods include DeepSVDD (Ruff et al., 2018), DAGMM (Zong et al., 2018); Lunar (Goodge et al., 2022), DROCC (Goyal et al., 2020), GOAD (Bergman & Hoshen, 2020), SO-GAAL and MO-GAAL (Liu et al., 2019), SLAD (Xu et al., 2023b) and DIF (Xu et al., 2023a). Deep kNN methods (Pang et al., 2018; Sun et al., 2022) learn representations to apply kNN. ICL (Shenkar & Wolf, 2022), which uses contrastive representation learning reported competitive results for ODDS datasets, for the semi-supervised setting.

**Diffusion-based Techniques** While diffusion models have been previously used for anomaly detection in image and video (Yan et al., 2023; Flaborea et al., 2023; Tur et al., 2023) data for a one-class setting (semi-supervised), their application in the context of tabular data and the unsupervised setting was unexplored. Wolleb et al. (2022) proposed an encoding method using a diffusion process followed by a denoising procedure guided by a classifier. Zhang et al. (2023a) synthesizes anomaly samples to train the denoising network for anomaly repair. AnoDDPM employs a specific diffusion noise to train a denoising network for normal image reconstruction (Wyatt et al., 2022). Similarly, Graham et al. (2023) utilized a DDPM to reconstructs an image for multiple different timesteps combined together to make anomaly scores. Liu et al. (2023) introduced a diffusion method that reconstruct an image by in-painting the input masked by a checkerboard pattern. Lastly, Zhang et al. (2023b) used a latent diffusion model trained with simulated anomalous samples on images.

## 6 CONCLUSION

This paper investigates the applicability of diffusion modelling for unsupervised and semi-supervised anomaly detection. We observe that specific design choices in DDPMs, although somewhat arbitrary, significantly influence their performance. Despite the expressivity and interpretability of DDPMs, they come with notable computational overhead compared to existing parametric techniques. For anomaly detection, DDPM essentially estimates the distance between the input and its "denoised reconstruction"; we observe that one could directly produce this estimate, or equivalently estimate the diffusion time. We first observe that the distribution of diffusion time given a noisy input, follows an inverse Gamma distribution. This forms the basis for our non-parametric approach that accurately predicts the diffusion time and turns out to create the same anomaly score ranking as kNN. A subsequent parametric strategy leverages a deep neural network, harnessing its generalization and rapid inference capabilities for large datasets. We evaluate the effectiveness of DTE on ADBench, a benchmark comprising popular anomaly detection datasets. Our results demonstrate competitive performance compared to prior work while improving the inference time by several orders of magnitude. Furthermore, we find that using pre-trained embeddings for images considerably improves the performance of diffusion-based methods, showing the potential advantage of using latent space diffusion.

## 7 LIMITATIONS AND FUTURE WORK

While our approach, DTE, achieves excellent performance with low inference time, it is important to acknowledge that in terms of interpretability, DTE falls behind DDPM as we explain in Appendix B and Section 4. This may pose challenges for practitioners seeking to understand the underlying mechanisms and behaviours of the data. Evaluating DTE in handling larger and more complex real-world datasets remains an avenue for future exploration. While here, we only address point anomalies, applications of diffusion modelling for group and contextual anomalies remain a high-impact unexplored area that we plan to investigate in the future.

## 8 REPRODUCIBILITY STATEMENT

We have made efforts to ensure that our method is reproducible. Appendix D.1 provides a description of all datasets included in ADBench, along with the preprocessing steps. Appendix D.2 presents a formal algorithm for parametric DTE and Appendix D.3 provides a detailed description of the network architecture and hyperparameters. We provide full results for both the unsupervised and semi-supervised settings with additional metrics, for all individual datasets and baselines in Appendix F as a reference for researchers to reproduce our experimental results. We are releasing the code as part of the supplemental material with detailed explanations to run the experiments.

## ACKNOWLEDGEMENTS

We want to thank Mehran Shakerinava for his input in the early stages of this project and Katelin Schutz for helpful discussion. The NSERC NFRF program and CIFAR AI Chairs partly support this research. Mila and the Digital Research Alliance of Canada provide computational resources.

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

# A ABLATION STUDIES

We perform several ablation studies to understand DDPM and the proposed DTE method.

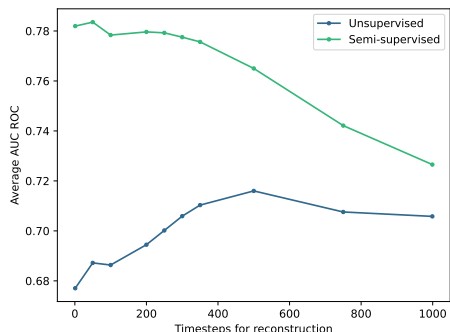 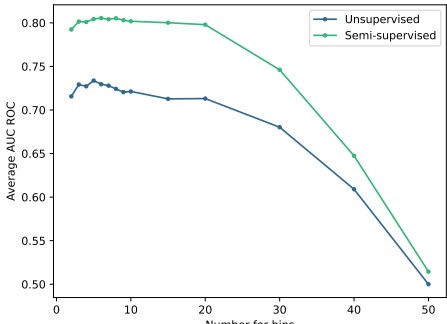

Figure 6: Average AUC ROC over the 57 ADBench datasets for different reconstruction timesteps of the DDPM model.

Figure 7: Average AUC ROC over the 57 AD-Bench datasets for different number of bins of the DTE categorical model.

**Reconstruction timestep in DDPM** When using DDPMs for anomaly detection based on the reconstruction distance, the denoising model requires an input timestep to create the reconstruction. We found that this somewhat arbitrary hyperparameter choice can significantly affect performance as shown in Figure 6.

For the unsupervised setting, we found that a value close to 50% of the maximum timestep results in the highest AUC ROC score on average. For the semi-supervised, the AUC ROC decreases as we increase the reconstruction timestep. Since the model is trained only on normal samples, the anomalies are sufficiently distanced from the learned data manifold for minor changes to result in a large reconstruction error while a larger timestep decreases the precision on normal samples.

**Number of bins in categorical DTE** As discussed in Section 3.3, we implement categorical DTE by combining multiple timesteps into bins. This turns out to be an important hyperparameter as it affects the final performance significantly. Figure 7 shows that a low number of bins leads to better performance. This can be attributed to the fact that we calculate the mean of the predicted timestep distribution rather than the mode to calculate anomaly scores and that adding more bins increases the complexity of the learning task.

**Maximum timestep in DTE** We study the effect of changing the maximum timestep in the noising diffusion process. As seen in Figure 8, the maximum timestep affects performance until roughly $T = 250$, since for very low values of $T$, the noisy samples might not resemble standard Gaussian noise and might not cover all potential anomalies in the dataset. We also note categorical DTE is more robust to the value of $T$ than the inverse Gamma DTE.

Figure 9 shows the value of standard deviation versus timestep as we increase the maximum timestep $T$. We observe that for values of $T \geq 250$, the final timestep corresponds to a standard deviation close enough to 1.0 that the data resembles samples drawn from a standard Gaussian distribution.

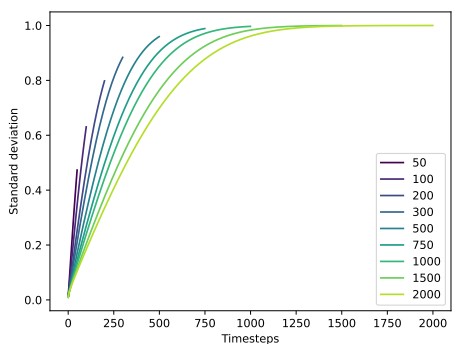

Figure 9: Standard deviations versus timestep for different values of the maximum timestep $T$.

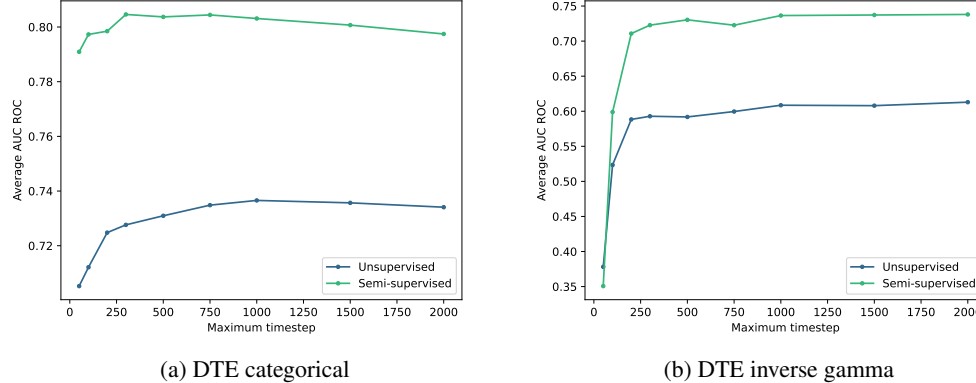

(a) DTE categorical      (b) DTE inverse gamma

Figure 8: Average AUC ROC over the 57 ADBench datasets for different maximum timestep T for the categorical and inverse gamma DTE models on both semi-supervised and unsupervised settings.

## B  INTERPRETABILITY

In certain applications, the mere identification of anomalies in the dataset is insufficient; it is imperative to understand the underlying reasons for flagging specific data points as anomalies. Both DDPM and DTE can provide interpretability by identifying a corresponding "denoised" or normal data point. In DDPM this is achieved using the deterministic ODE flow, which is (rather arbitrarily) initialized at some large time step. We found the initial time step to be an important hyper-parameter, which impacts both anomaly detection and interpretability for DDPM. In practice, $T' = .25 \times T$ performs well as the initial time-step. DTE has the benefit of avoiding such hyper-parameters, where one could use the gradient flow associated with the mode of the posterior to denoise a given input; see Figure 2 (d, e, and f).

Figure 10 shows another example, this time using the categorical likelihood on the MNIST dataset. We artificially introduce a gray patch as an anomaly (Figure 10 (a)) and perform the gradient descent procedure reducing the mean

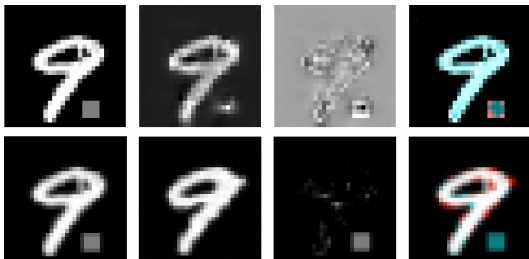

(a) Original    (b) Denoised  (c) Difference  (d) Gradient

Figure 10: Interpretability in DTE (first row) and DDPM (second row) on MNIST. Visual interpretation of a gray patch anomaly on an MNIST image using the categorical diffusion model with a simple convolution network on the first row and a DDPM on the second for comparison. a) original anomalous image, b) the denoised version using gradient descent c) difference between the original and the denoised image, d) visualization of the gradient on top of the original image.

of the posterior density. We observe that this procedure indeed partially eliminates the patch (Figure 10 (b)). We also note that since it is explicitly trained to remove the noise from a noisy input, DDPM performs better in removing the patch.

As detailed in Section 3.2, the non-parametric DTE yields the same anomaly score as kNN. Thus, the parametric DTE can be viewed as an approximate parametric kNN algorithm. This perspective enhances DTE's interpretability: the neural network's score represents the estimated distance of a point to the manifold. Although we can't pinpoint which training set instance most closely matches an input, interpreting the score as a distance to a certain neighbourhood offers a straightforward insight into the method's functioning.

## C    Non-parametric Estimation of Timestep Distribution

In Figure 3, we visualize the analytical posterior distribution along with the non-parametric estimate. The difference between these distributions is shown in Figure 11. While the two distributions are quite similar, their shape is very peaked for low values of the diffusion timestep. The slight misalignment between the peaks of the analytical and the non-parametric estimate gives rise to the spiky shape seen in the difference. For higher values of the diffusion timestep, the difference is very close to zero, demonstrating that the non-parametric estimate based on k-nearest neighbours is a very close approximation to the true posterior distribution of timestep.

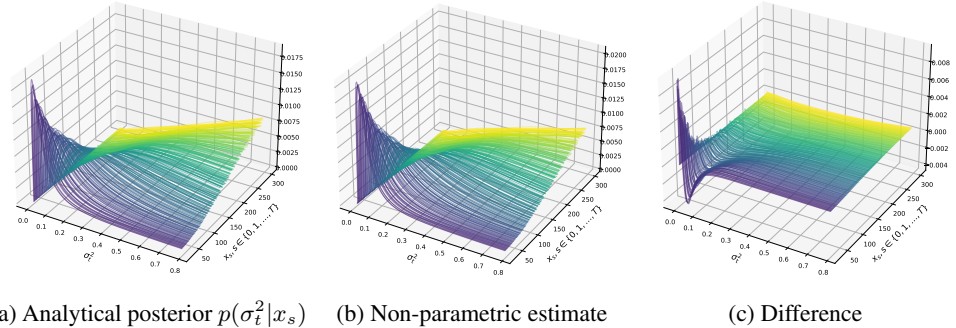

(a) Analytical posterior $p(\sigma_t^2|x_s)$      (b) Non-parametric estimate      (c) Difference

Figure 11: Posterior timestep distribution $p(\sigma_t^2|\mathbf{x}_s)$, where $\mathbf{x}_s$ is produced using diffusion with different time steps $s \in \{1, \ldots, T\}$, averaged over the `vertebral` dataset. (a) shows the analytical distribution computed by placing Gaussian distributions of different variances at each point in the dataset, (b) shows the inverse Gamma distribution with scale parameter value depending on the average distance to the k-nearest neighbours ($k = 32$), and (c) shows the difference between (a) and (b).

## D    Implementation Details

### D.1    Datasets and Preprocessing

**Datasets description**    We show the results from our methods and baselines over multiple datasets from ADBench (Han et al., 2022) described in Table 1. There are 47 tabular datasets ranging from multiple different applications. There are also five datasets composed of extracted representations of images after the last average polling layer from a Resnet-18 (He et al., 2015) model pre-trained on ImageNet. Similarly, there are five datasets composed of extracted embedding of NLP tasks from BERT (Devlin et al., 2019). We also show results on VisA (Zou et al., 2022), which is a dataset composed of images of 12 different objects where the anomalies are various flaws on the objects.

**Training and test data configuration**    For ADBench, the semi-supervised setting, we use half of the normal data in the training set, and the other half is in the test set with all the anomalies. For the unsupervised setting, we sample the whole dataset with replacement for the training data, while the test data is the whole dataset. This bootstrapping method allows us to test the variance over the training dataset for each method.

**Preprocessing**    We standardize the input samples based on the mean and standard deviation calculated over the training data, to ensure consistency across the input values and mitigate the impact of potential outliers or scale variations. For VisA, 90% of the normal instances are making the training data, while the anomalies and the remaining 10% are in the test set. For CIFAR-10 and MNIST, One class is set as the anomaly while the others are part of the training data. 80% of the normal instances are in the training data while the remaining 20% and the anomalies are in the test data. For ADBench, CIFAR-10, MNIST-C, SVHN, and FashionMNIST are made up of one class for the normal sample, while the anomalies are the rest of the classes downsampled to make up 5% of the total data.

**On the importance of standardization for diffusion models**  Throughout the course of our investigations, we discovered the critical importance of standardization. This is due to the fact that the incorporated Gaussian noise operates under the assumption that each feature is centered at zero with unit standard deviation. Consequently, implementing standard scaling facilitates the comprehensive

Table 1: Description of all datasets in ADBench

| Dataset Name | # Samples | # Features | # Anomaly | % Anomaly | Category |
|---|---|---|---|---|---|
| ALOI | 49534 | 27 | 1508 | 3.04 | Image |
| annthyroid | 7200 | 6 | 534 | 7.42 | Healthcare |
| backdoor | 95329 | 196 | 2329 | 2.44 | Network |
| breastw | 683 | 9 | 239 | 34.99 | Healthcare |
| campaign | 41188 | 62 | 4640 | 11.27 | Finance |
| cardio | 1831 | 21 | 176 | 9.61 | Healthcare |
| Cardiotocography | 2114 | 21 | 466 | 22.04 | Healthcare |
| celeba | 202599 | 39 | 4547 | 2.24 | Image |
| census | 299285 | 500 | 18568 | 6.20 | Sociology |
| cover | 286048 | 10 | 2747 | 0.96 | Botany |
| donors | 619326 | 10 | 36710 | 5.93 | Sociology |
| fault | 1941 | 27 | 673 | 34.67 | Physical |
| fraud | 284807 | 29 | 492 | 0.17 | Finance |
| glass | 214 | 7 | 9 | 4.21 | Forensic |
| Hepatitis | 80 | 19 | 13 | 16.25 | Healthcare |
| http | 567498 | 3 | 2211 | 0.39 | Web |
| InternetAds | 1966 | 1555 | 368 | 18.72 | Image |
| Ionosphere | 351 | 32 | 126 | 35.90 | Oryctognosy |
| landsat | 6435 | 36 | 1333 | 20.71 | Astronautics |
| letter | 1600 | 32 | 100 | 6.25 | Image |
| Lymphography | 148 | 18 | 6 | 4.05 | Healthcare |
| magic.gamma | 19020 | 10 | 6688 | 35.16 | Physical |
| mammography | 11183 | 6 | 260 | 2.32 | Healthcare |
| mnist | 7603 | 100 | 700 | 9.21 | Image |
| musk | 3062 | 166 | 97 | 3.17 | Chemistry |
| optdigits | 5216 | 64 | 150 | 2.88 | Image |
| PageBlocks | 5393 | 10 | 510 | 9.46 | Document |
| pendigits | 6870 | 16 | 156 | 2.27 | Image |
| Pima | 768 | 8 | 268 | 34.90 | Healthcare |
| satellite | 6435 | 36 | 2036 | 31.64 | Astronautics |
| satimage-2 | 5803 | 36 | 71 | 1.22 | Astronautics |
| shuttle | 49097 | 9 | 3511 | 7.15 | Astronautics |
| skin | 245057 | 3 | 50859 | 20.75 | Image |
| smtp | 95156 | 3 | 30 | 0.03 | Web |
| SpamBase | 4207 | 57 | 1679 | 39.91 | Document |
| speech | 3686 | 400 | 61 | 1.65 | Linguistics |
| Stamps | 340 | 9 | 31 | 9.12 | Document |
| thyroid | 3772 | 6 | 93 | 2.47 | Healthcare |
| vertebral | 240 | 6 | 30 | 12.50 | Biology |
| vowels | 1456 | 12 | 50 | 3.43 | Linguistics |
| Waveform | 3443 | 21 | 100 | 2.90 | Physics |
| WBC | 223 | 9 | 10 | 4.48 | Healthcare |
| WDBC | 367 | 30 | 10 | 2.72 | Healthcare |
| Wilt | 4819 | 5 | 257 | 5.33 | Botany |
| wine | 129 | 13 | 10 | 7.75 | Chemistry |
| WPBC | 198 | 33 | 47 | 23.74 | Healthcare |
| yeast | 1484 | 8 | 507 | 34.16 | Biology |
| CIFAR10 | 5263 | 512 | 263 | 5.00 | Image |
| FashionMNIST | 6315 | 512 | 315 | 5.00 | Image |
| MNIST-C | 10000 | 512 | 500 | 5.00 | Image |
| MVTec-AD | 5354 | 512 | 1258 | 23.50 | Image |
| SVHN | 5208 | 512 | 260 | 5.00 | Image |
| Agnews | 10000 | 768 | 500 | 5.00 | NLP |
| Amazon | 10000 | 768 | 500 | 5.00 | NLP |
| Imdb | 10000 | 768 | 500 | 5.00 | NLP |
| Yelp | 10000 | 768 | 500 | 5.00 | NLP |
| 20newsgroups | 11905 | 768 | 591 | 4.96 | NLP |

coverage of the anomaly detection space by the noise. This proved to be an essential component of the proposed anomaly detection method.

## D.2 ALGORITHM

---

**Algorithm 1** Training Process for parametric DTE

---

    **Parameters:** $T$ : maximum timestep, $\lambda$ : learning rate
    **Input:** Training data $\mathcal{D}$
1:  $\theta \leftarrow \theta_0$                                               $\triangleright$ Initialize weights of the model
2:  $\beta_0, \beta_1, ..., \beta_{T-1} \leftarrow \text{linear}(0, 0.01)$            $\triangleright$ Define the $\beta$ schedule for forward diffusion
3:  **for all** $t < T$ **do**
4:     $\bar{\alpha}_t \leftarrow \prod_{s=1}^{t}(1 - \beta_s)$                               $\triangleright$ Compute the $\bar{\alpha}$
5:     $\sigma_t \leftarrow \sqrt{1 - \bar{\alpha}_t}$                        $\triangleright$ Set standard deviation for each timestep
6:  **end for**
7:  **for** num_epochs **do**
8:     **for all** $\mathbf{x}_0$ in $\mathcal{D}$ **do**
9:         $t \sim \mathcal{U}(0, T - 1)$                        $\triangleright$ Sample timestep $t$ uniformly
10:        $\epsilon \sim \mathcal{N}(0, 1)$                       $\triangleright$ Sample standard Gaussian noise
11:        $\mathbf{x}_t \leftarrow \mathbf{x}_0 + \sigma_t\,\epsilon$               $\triangleright$ Compute noisy sample of $x$ at timestep $t$
12:        $\mathcal{L} \leftarrow \text{loss}(f_\theta(\mathbf{x}_t))$    $\triangleright$ Equation (8) for inverse Gamma or Equation (9) for categorical
13:        $\theta \leftarrow \theta - \lambda \nabla_\theta \mathcal{L}$                    $\triangleright$ Update model parameters
14:     **end for**
15: **end for**

---

## D.3 MODEL ARCHITECTURE AND HYPERPARAMETERS

We first found the hyperparameters using different training splits for the semi-supervised setting on the shuttle and thyroid datasets (network architecture, maximum timestep, batch size, number of epochs). We then tuned some of them over all the datasets using different training seeds than the ones used for the final results (number of bins and learning rate). This is the case for the diffusion methods and the normalizing flow method. For the other baselines, we picked the set of hyperparameters from the original papers that provided the best results over the whole benchmark.

**DTE** For the non-parametric DTE, the score is calculated based on the approximate posterior distribution in Equation (6) with $k = 5$ for the semi-supervised setting and $k = 32$ for the unsupervised setting. The anomaly score is the mean of the posterior to avoid having an anomaly score that is restricted by the maximum variance using the mode. The be consistent, we selected the same $k$ for the kNN baseline.

For the DTE parametric approach, we employ a multi-layer perceptron (MLP) neural network. We use a common architecture and set of hyperparameters across all datasets. When training on images, we used a ResNet-50 architecture.

For the categorical model, we found that using the mean over each output probability bin provided the best results. That is, the anomaly score for each individual $\mathbf{x}$ is computed as follow:

$$score = f_\theta(\mathbf{x}) \begin{bmatrix} 0 \\ 1 \\ 2 \\ \vdots \\ B - 1 \end{bmatrix} \tag{9}$$

where $B$ is the number of bins and $f_\theta(\mathbf{x}_t)$ is the output probability vector of the network using a softmax, which is an $N \times B$ matrix, where the sum across each row equates to one and $N$ is the batch size. The score for each instance will be a value between 0 and $B - 1$. The higher the score is, the more anomalous an instance is.

Employing the mode as a measurement metric proved suboptimal given the disproportionate representation of the first bin, a pattern that remained consistent even among anomalous instances.

Consequently, it was observed that while the probabilities could be diffusely distributed across the remaining bins, the mode predominantly remained in the first bin. In contrast, utilizing the mean allowed us to effectively account for this distribution characteristic, enabling an inclusive weighting scheme across all bins. Additionally, the mean offered a continuous scoring system as opposed to the integer values provided by the mode, thereby affording a more nuanced understanding of the anomalous data.

Table 2: Hyperparameters for parametric DTE model

| Hyperparameter | Value |
| --- | --- |
| Hidden layer sizes | [256, 512, 256] |
| Activation function | ReLU |
| Optimizer | Adam |
| Learning rate | 0.0001 |
| Dropout | 0.5 |
| Batch size | 64 |
| Number of epochs | 400 |
| Maximum timestep | 300 |
| Number of bins | 7 |

**DDPM** For the DDPM model, we used a modified ResNet for tabular data (Gorishniy et al., 2021) with added time embedding before each block, inspired by the work done for TabDDPM (Kotelnikov et al., 2022). Recognizing that learning noise at each timestep presents a considerably complex task, the necessity for a more sophisticated model than a simple MLP became evident to optimize the efficacy of our method. Furthermore, the lack of research on diffusion models for tabular data has constrained our ability to apply a model of comparable strength to the U-net model typically used for images, to our benchmark datasets. This presents an interesting direction for further research, with the potential to significantly enhance the performance of machine learning models on tabular datasets. In contrast to prior work (Wyatt et al., 2022; Wolleb et al., 2022), we do not add noise to the data point before reconstructing it as we found that it leads to overall slightly better results. This is a minor change, one intuition for the boost of performance could be that adding noise can modify the images toward anomalous data, thus increasing the amount of false positives.

Table 3: Hyperparameters for DDPM model

| Hyperparameter | Value |
| --- | --- |
| Number of blocks | 3 |
| Main layer size | 128 |
| Hidden layer size | 256 |
| Time embedding dimensions | 256 |
| Optimizer | Adam |
| Learning rate | 0.0001 |
| Dropout layer 1 | 0.4 |
| Dropout layer 2 | 0.1 |
| Batch size | 64 |
| Number of epochs | 400 |
| Maximum timestep | 1000 |
| Reconstruction timestep | 250 |

**Normalizing Flows Baseline** We compare our diffusion methods with a normalizing flows baseline that uses planar flows (Rezende & Mohamed, 2015). Normalizing flows allow to compute the exact likelihoods of data point, which allow to easily assign anomaly scores. Once trained, the model can estimate the density of any data point in the input space. This is done by passing the data point through the inverse of the learned transformation and then computing the density of the transformed point under the simple target distribution. The density of the original point under the complex data distribution can be computed from this using the change-of-variables formula.

Table 4: Hyperparameters for PlanarFlow model

| Hyperparameter | Value |
|---|---|
| Number of transformations | 10 |
| Optimizer | Adam |
| Learning rate | 0.002 |
| Batch size | 64 |
| Number of epochs | 200 |

### D.4 COMPUTE

The total amount of compute required to reproduce our experiments with five seeds, including all of the baselines and the proposed DTE model amounts to 473 GPU-hours for the unsupervised setting and 225 GPU-hours for the semi-supervised setting on an RTX8000 GPU with 48 gigabytes of memory for running the ADBench datasets.

Figure 12 shows the training and inference times averaged over all datasets in ADBench over five seeds for all methods discussed in Section 4. As expected, deep learning-based methods have significantly higher training times compared to classical methods but comparable inference times. In particular, the inference time for the parametric DTEs is orders of magnitude lower than all other methods. The non-parametric variant of DTE has no training phase, so we show the inference time in both plots.

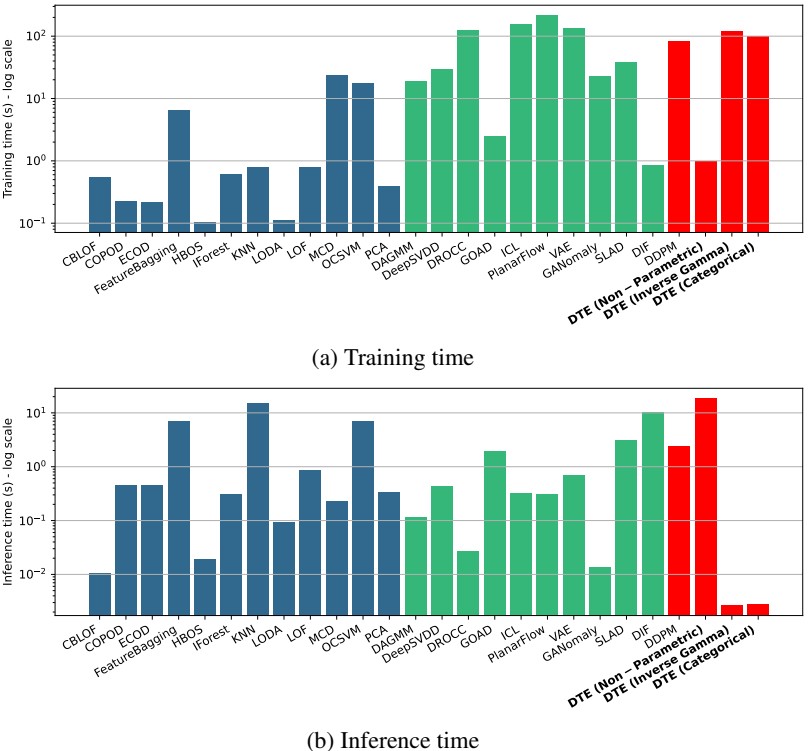

(a) Training time

(b) Inference time

Figure 12: Mean training and inference time on the 57 datasets from ADBench over five different seeds for the semi-supervised setting using normal samples only for training. Colour scheme: red (diffusion-based), green (deep learning methods), blue (classical methods).

## E  CHOICE OF REPRESENTATION FOR IMAGES

In this section, we compare the effect of choice of representation on the performance of diffusion-based anomaly detection techniques. Three choices considered are 1) pixel space representation, 2) self-supervised embedding, and 3) embedding produced by a classifier. Results for three image datasets are reported in Table 5. The datasets and preprocessing are described in Appendix D.1 and the full results are in Appendix F. As expected, using pre-trained embeddings leads to better results than pixel space for all methods considered. Tables 9 to 12 report other experiments that lead to a similar conclusion.

In particular, using self-supervised embedding for CIFAR-10, significantly improved the anomaly detection performance as the pre-training was done on CIFAR-10 itself. Note that all the other pre-training were supervised classification using ResNet-34 on ImageNet and not directly on the datasets. Overall, pre-training improves the results for all methods and all datasets with the exception of kNN and the non-parametric DTE (DTE-NP) on MNIST. This result can be attributed to the simplicity of the MNIST dataset when adapted to anomaly detection tasks. As a reminder, DTE-NP is equivalent to kNN, but corresponds to the variation that uses the mean distance of the k-nearest neighbours instead of the distance to the kth-nearest neighbour.

Zou et al. (2022) highlighted the advantages of tailoring specialized self-supervised learning techniques to specific datasets, exemplified by their method for VisA. As our methods are not explicitly designed for these datasets, our results for all diffusion-based methods reported here lag behind those of methods specialized to this dataset. In particular, VisA dataset contains images that are quite similar with the exception of highly localized anomalies.

Table 5: Average AUC ROC and standard deviations for the different subsets of each dataset, average across 5 runs, semi-supervised setting using different pre-training algorithms.

|  | DTE-NP | DTE-C | DDPM | kNN |
|---|---|---|---|---|
| VisA, supervised ImageNet pre-training | 83.63(10.50) | 81.07(11.01) | 80.47(12.47) | 83.26(10.64) |
| VisA, VicReg ImageNet pre-training | 83.36(12.44) | 81.89(12.26) | 83.14(13.76) | 83.68(13.54 |
| VisA, no pre-training | 75.96(10.54) | 64.53(19.61) | 57.85(21.74) | 75.40(9.85) |
| CIFAR10, supervised ImageNet pre-training | 53.91(7.16) | 52.57(5.53) | 52.96(7.05) | 54.42(7.56) |
| CIFAR10, VicReg pre-training | 80.92(10.81) | 63.36(11.92) | 54.22(10.26) | 79.01(11.53) |
| CIFAR10, no pre-training | 51.53(14.81) | 50.25(3.34) | 50.50(7.67) | 51.64(14.90) |
| MNIST, supervised ImageNet pre-training | 78.07(12.48) | 64.34(11.93) | 60.62(10.26) | 76.86(11.54) |
| MNIST, no pre-training | 81.94(16.46) | 49.02(16.51) | 51.29(18.85) | 84.14(15.60) |

## F  FULL RESULTS

We provide the full table of results corresponding to the AUC ROC box-plots in Section 4. We report additional metrics including F1 score and area under the precision-recall curve (AUC PR) along with the corresponding box-plots. All results are shown averaged across five seeds along with standard deviations in brackets for all 57 datasets in ADBench. In the subsequent tables, DTE-NP refers to the non-parametric DTE estimator, DTE-IG refers to the parametric inverse Gamma model, and DTE-C refers to the parametric categorical model. Tables 9 to 12 show the results for three methods when using pre-trained embeddings on CIFAR-10 and SVHN compared to trained directly on the images, as it is set up in ADBench. The difference with Table 5 is that instead of having one class as an anomaly, here we have one class as normal while the rest of the classes are downsampled to produce the anomalies.

### F.1 SEMI-SUPERVISED SETTING

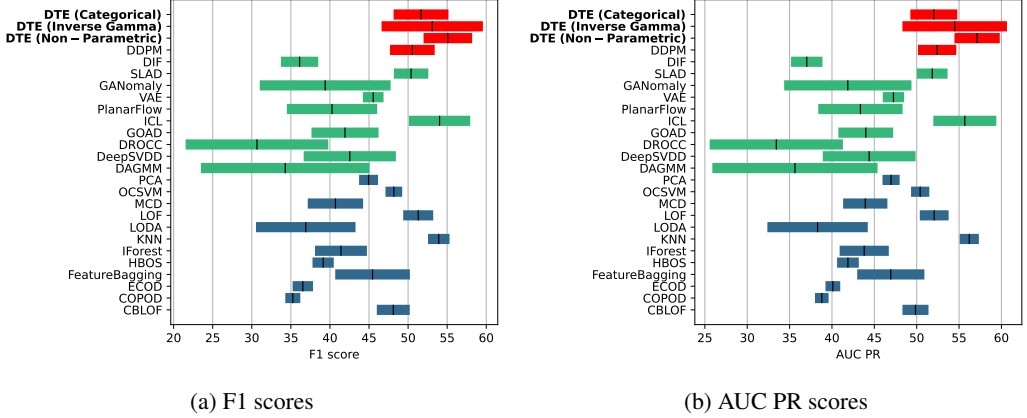

| (a) F1 scores | (b) AUC PR scores |
|---|---|

Figure 13: F1 score and AUC PR means and standard deviations on the 57 datasets from ADBench over five different seeds for the semi-supervised setting using normal samples only for training. Colour scheme: red (diffusion-based), green (deep learning methods), blue (classical methods).

Table 6: Average AUC ROC and standard deviations for 5 runs of the VisA dataset, semi-supervised setting using embeddings of supervised ResNet-34 pre-trained on ImageNet with the same training split.

|  | **DTE-NP** | **DTE-C** | DDPM | kNN |
|---|---|---|---|---|
| candle | 90.88(0.0) | 89.26(3.37) | 87.37(0.26) | 90.76(0.0) |
| capsules | 62.77(0.0) | 56.04(3.51) | 66.65(0.6) | 62.67(0.0) |
| cashew | 93.7(0.0) | 87.17(2.95) | 89.69(0.25) | 93.24(0.0) |
| chewinggum | 93.88(0.0) | 94.69(1.53) | 92.55(0.26) | 93.52(0.0) |
| fryum | 87.38(0.0) | 81.25(3.91) | 85.84(0.17) | 87.68(0.0) |
| macaroni1 | 70.27(0.0) | 71.9(4.94) | 64.33(0.69) | 69.34(0.0) |
| macaroni2 | 67.65(0.0) | 66.77(2.36) | 51.6(0.64) | 66.35(0.0) |
| pcb1 | 90.14(0.0) | 85.49(1.72) | 94.38(0.37) | 90.67(0.0) |
| pcb2 | 88.88(0.0) | 81.32(2.76) | 83.24(0.4) | 87.77(0.0) |
| pcb3 | 80.83(0.0) | 81.72(1.98) | 79.61(0.2) | 81.25(0.0) |
| pcb4 | 93.77(0.0) | 91.05(2.46) | 88.7(0.56) | 93.07(0.0) |
| pipe fryum | 83.24(0.0) | 86.19(2.38) | 81.72(0.36) | 82.86(0.0) |
| mean | 83.63(10.50) | 81.07(11.01) | 80.47(12.47) | 83.26(10.64) |

Table 7: Average AUC ROC and standard deviations for 5 runs of the VisA dataset, semi-supervised setting using embeddings of VicReg pre-trained on ImageNet with the same training split.

| | DTE-NP | DTE-C | DDPM | kNN |
|---|---|---|---|---|
| candle | 82.97(0.0) | 84.72(0.8) | 85.27(0.07) | 85.44(0.0) |
| capsules | 65.63(0.0) | 68.24(1.08) | 69.01(0.6) | 65.92(0.0) |
| cashew | 90.7(0.0) | 82.75(5.28) | 90.26(0.43) | 90.74(0.0) |
| chewinggum | 97.98(0.0) | 97.78(0.12) | 97.78(0.08) | 98.0(0.0) |
| fryum | 88.88(0.0) | 79.62(2.69) | 89.13(0.23) | 88.82(0.0) |
| macaroni1 | 70.03(0.0) | 68.17(1.19) | 64.09(0.17) | 69.52(0.0) |
| macaroni2 | 52.06(0.0) | 55.81(2.06) | 51.93(0.27) | 52.16(0.0) |
| pcb1 | 93.02(0.0) | 91.07(0.5) | 93.19(0.08) | 93.22(0.0) |
| pcb2 | 85.7(0.0) | 83.77(0.96) | 83.68(0.17) | 85.69(0.0) |
| pcb3 | 83.26(0.0) | 81.57(0.75) | 82.03(0.13) | 83.05(0.0) |
| pcb4 | 98.6(0.0) | 98.21(0.35) | 98.35(0.03) | 98.66(0.0) |
| pipe fryum | 91.44(0.0) | 90.98(1.29) | 93.05(0.05) | 92.96(0.0) |
| mean | 83.36(12.44) | 81.89(12.26) | 83.14(13.76) | 83.68(13.54) |

Table 8: Average AUC ROC and standard deviations for 5 runs of the VisA dataset, semi-supervised setting using the images directly with the same training split.

| | DTE-NP | DTE-C | DDPM | kNN |
|---|---|---|---|---|
| candle | 77.48(0.0) | 83.03(4.42) | 51.96(6.2) | 77.38(0.0) |
| capsules | 63.75(0.0) | 72.61(7.91) | 33.19(0.37) | 68.02(0.0) |
| cashew | 90.14(0.0) | 79.5(27.78) | 96.26(0.56) | 93.32(0.0) |
| chewinggum | 66.92(0.0) | 56.99(4.66) | 68.82(1.02) | 65.66(0.0) |
| fryum | 74.32(0.0) | 77.28(10.99) | 25.24(1.22) | 74.5(0.0) |
| macaroni1 | 68.67(0.0) | 54.52(20.66) | 74.7(1.14) | 70.11(0.0) |
| macaroni2 | 74.04(0.0) | 54.48(8.11) | 37.04(0.48) | 77.02(0.0) |
| pcb1 | 83.59(0.0) | 51.53(16.31) | 72.02(0.97) | 80.53(0.0) |
| pcb2 | 87.4(0.0) | 74.04(15.82) | 77.56(0.55) | 78.87(0.0) |
| pcb3 | 71.75(0.0) | 40.86(8.75) | 68.11(2.12) | 66.03(0.0) |
| pcb4 | 94.5(0.0) | 73.61(11.82) | 28.46(2.18) | 92.94(0.0) |
| pipe fryum | 58.96(0.0) | 56.04(19.1) | 60.95(5.78) | 60.42(0.0) |
| mean | 75.96(10.54) | 64.53(19.61) | 57.85(21.74) | 75.40(9.85) |

Table 9: Mean AUC ROC and standard deviation over 5 seeds for different methods trained on the images directly versus trained on embeddings generated by a pre-trained ResNet-18 on ImageNet for the unsupervised setting on the CIFAR-10 dataset.

| | DDPM | DTE-C | kNN |
|---|---|---|---|
| Images | 54.72(4.55) | 48.95(8.33) | 57.45(1.55) |
| Embeddings | 66.34(0.14) | 62.87(1.57) | 66.17(0.33) |

Table 10: Mean AUC ROC and standard deviation over 5 seeds for different methods trained on the images directly versus trained on embeddings generated by a pre-trained ResNet-18 on ImageNet for the unsupervised setting on the SVHN dataset.

| | DDPM | DTE-C | kNN |
|---|---|---|---|
| Images | 54.97(2.41) | 49.07(3.06) | 56.29(1.22) |
| Embeddings | 61.48(0.24) | 59.96(1.24) | 61.17(0.28) |

Table 11: Mean AUC ROC and standard deviation over 5 seeds for different methods trained on the images directly versus trained on embeddings generated by a pre-trained ResNet-18 on ImageNet for the semi-supervised setting on the CIFAR-10 dataset.

|  | DDPM | DTE-C | kNN |
|---|---|---|---|
| Images | 55.96(4.69) | 52.66(6.28) | 59.10(1.80) |
| Embeddings | 67.91(0.13) | 68.53(1.59) | 67.53(0.0) |

Table 12: Mean AUC ROC and standard deviation over 5 seeds for different methods trained on the images directly versus trained on embeddings generated by a pre-trained ResNet-18 on ImageNet for the semi-supervised setting on the SVHN dataset.

|  | DDPM | DTE-C | kNN |
|---|---|---|---|
| Images | 57.28(2.75) | 48.78(4.23) | 55.92(1.17) |
| Embeddings | 61.37(0.08) | 62.91(1.1) | 61.69(0.0) |

## F.2 UNSUPERVISED SETTING

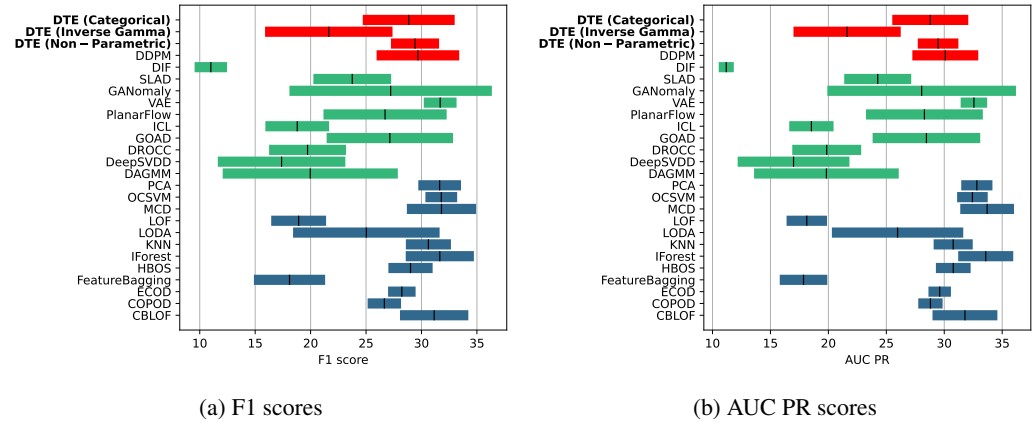

(a) F1 scores

(b) AUC PR scores

Figure 14: F1 score and AUC PR means and standard deviations on the 57 datasets from ADBench over five different seeds for the unsupervised setting with bootstrapped training instances. Colour scheme: red (diffusion-based), green (deep learning methods), blue (classical methods).

Table 13: Average AUC ROC and standard deviations over five seeds for the semi-supervised setting on ADBench.

| | CBLOF | COPOD | ECOD | FeatureBagging | HBOS | IForest | kNN | LODA |
|---|---|---|---|---|---|---|---|---|
| aloi | 53.69(0.15) | 49.51(0.0) | 51.73(0.0) | 49.07(0.53) | 52.23(0.0) | 50.74(0.68) | 51.04(0.0) | 49.24(2.75) |
| amazon | 58.17(0.12) | 56.78(0.0) | 53.79(0.0) | 57.94(0.04) | 56.32(0.0) | 56.4(0.95) | 60.58(0.0) | 52.23(2.84) |
| annthyroid | 90.14(1.08) | 76.77(0.0) | 78.45(0.0) | 88.95(1.61) | 66.02(0.0) | 90.28(1.52) | 92.81(0.0) | 77.35(7.51) |
| backdoor | 69.65(5.23) | 50.0(0.0) | 50.0(0.0) | 94.83(0.62) | 70.81(0.89) | 74.89(2.67) | 93.75(0.53) | 47.62(22.63) |
| breastw | 99.11(0.23) | **99.46(0.09)** | 99.14(0.22) | 59.07(15.44) | 99.23(0.17) | **99.5(0.08)** | 99.05(0.26) | 98.13(0.35) |
| campaign | 77.05(0.31) | 78.15(0.0) | 76.86(0.0) | 69.1(3.85) | 77.06(0.0) | 73.64(1.48) | 78.48(0.0) | 58.88(4.48) |
| cardio | 93.49(1.47) | 93.16(0.0) | 94.95(0.0) | 92.12(0.56) | 80.7(0.0) | 93.32(1.43) | 92.0(0.0) | 91.34(2.93) |
| cardiotocography | 67.61(2.21) | 66.35(0.0) | **79.3(0.0)** | 63.64(2.4) | 61.24(0.0) | 74.24(2.88) | 62.11(0.0) | 72.79(7.6) |
| celeba | 79.28(1.3) | 75.72(0.59) | 76.33(0.63) | 46.89(1.8) | 76.68(0.66) | 71.23(2.27) | 73.14(0.72) | 62.46(13.17) |
| census | 70.84(0.28) | 50.0(0.0) | 50.0(0.0) | 55.92(1.0) | 62.5(0.47) | 62.55(2.38) | **72.26(0.29)** | 51.12(11.19) |
| cover | 94.04(0.28) | 88.2(0.27) | 91.86(0.21) | **99.16(0.63)** | 71.11(0.82) | 86.31(2.07) | 97.54(0.15) | 94.93(3.07) |
| donors | 93.47(0.23) | 81.5(0.21) | 88.74(0.38) | 95.21(1.72) | 81.19(0.61) | 89.44(2.22) | 99.49(0.06) | 63.52(27.27) |
| fault | 59.0(1.29) | 49.14(0.0) | 50.37(0.0) | 48.32(0.95) | 53.06(0.0) | 55.86(2.03) | 58.73(0.0) | 50.27(1.88) |
| fraud | 94.91(1.08) | 94.3(1.41) | 94.89(1.27) | 94.83(1.56) | **95.02(0.67)** | 94.73(1.23) | **95.43(1.04)** | 89.05(8.12) |
| glass | 89.35(1.48) | 76.0(1.94) | 71.14(3.54) | 88.49(1.71) | 82.59(3.23) | 81.09(2.56) | 92.04(1.12) | 67.34(5.39) |
| hepatitis | 86.26(2.35) | 80.9(1.22) | 73.84(1.99) | 67.76(6.5) | 84.84(0.78) | 82.69(2.75) | 96.46(1.46) | 68.98(3.97) |
| http | 99.93(0.01) | 99.19(0.09) | 97.95(0.12) | 92.1(0.57) | 98.58(1.04) | 99.35(0.29) | **100.0(0.0)** | 47.72(45.49) |
| imdb | 49.94(0.01) | 51.05(0.0) | 46.88(0.0) | 49.53(0.1) | 49.94(0.0) | 49.53(0.78) | 50.08(0.0) | 47.23(2.24) |
| internetads | 65.16(0.08) | 65.94(0.0) | 66.01(0.0) | 71.38(2.32) | 49.18(0.0) | 47.87(2.11) | 68.08(0.0) | 58.73(3.84) |
| ionosphere | 96.78(1.5) | 78.32(2.13) | 71.77(1.43) | 94.47(2.1) | 70.68(2.85) | 91.21(1.37) | 97.44(0.98) | 85.56(3.59) |
| landsat | 57.21(0.26) | 49.29(0.0) | 42.01(0.0) | 66.38(0.17) | **73.21(0.0)** | 58.8(2.21) | 68.25(0.0) | 44.65(3.3) |
| letter | 33.24(0.66) | 36.53(0.0) | 45.37(0.0) | 44.84(1.0) | 35.91(0.0) | 32.04(1.64) | 35.43(0.0) | 30.2(0.94) |
| lymphography | 99.83(0.02) | 99.53(0.2) | 99.52(0.15) | 96.61(2.84) | 99.69(0.17) | 99.45(0.32) | 99.93(0.08) | 67.04(13.87) |
| magic.gamma | 75.81(0.0) | 68.0(0.0) | 63.58(0.0) | 84.19(0.72) | 74.53(0.0) | 77.09(1.29) | 83.27(0.0) | 70.53(1.36) |
| mammography | 84.74(0.02) | 90.59(0.0) | **90.67(0.0)** | 86.31(0.35) | 85.01(0.0) | 88.02(0.3) | 87.58(0.0) | 89.62(0.87) |
| mnist | 91.1(0.23) | 50.0(0.0) | 50.0(0.0) | 92.55(0.4) | 62.34(0.0) | 86.6(1.99) | **93.85(0.0)** | 64.74(7.74) |
| musk | **100.0(0.0)** | 99.71(0.0) | 99.87(0.0) | **100.0(0.0)** | **100.0(0.0)** | 90.58(6.2) | **100.0(0.0)** | 99.67(0.35) |
| optdigits | 83.52(1.69) | 50.0(0.0) | 50.0(0.0) | 96.27(0.49) | 89.92(0.0) | 81.07(3.28) | 93.72(0.0) | 32.77(8.62) |
| pageblocks | 91.23(0.11) | 80.85(0.0) | 87.95(0.0) | 91.11(0.3) | 65.62(0.0) | 82.64(0.89) | 89.65(0.0) | 83.62(2.6) |
| pendigits | 96.67(0.07) | 90.74(0.0) | 92.95(0.0) | 99.5(0.11) | 93.55(0.0) | 97.22(0.48) | **99.87(0.0)** | 92.13(1.02) |
| pima | 72.94(1.07) | 66.59(1.32) | 60.56(1.57) | 71.93(2.22) | 74.76(1.6) | 74.26(1.63) | 76.94(1.87) | 62.68(7.55) |
| satellite | 73.2(0.92) | 68.34(0.0) | 62.22(0.0) | 80.11(0.11) | 85.5(0.0) | 77.46(1.48) | 82.24(0.0) | 69.73(1.15) |
| satimage-2 | 99.42(0.01) | 97.92(0.0) | 97.09(0.0) | 99.47(0.03) | 97.95(0.0) | 99.12(0.21) | 99.71(0.0) | 98.67(0.52) |
| shuttle | 99.72(0.02) | 99.47(0.0) | 99.33(0.0) | 86.89(8.22) | 98.64(0.0) | 99.65(0.07) | 99.91(0.0) | 71.68(33.88) |
| skin | 91.82(0.21) | 47.21(0.18) | 49.14(0.18) | 78.39(0.91) | 76.92(0.35) | 89.42(0.58) | **99.49(0.08)** | 75.51(5.43) |
| smtp | 87.28(5.65) | 91.15(1.58) | 88.26(2.46) | 84.81(3.66) | 82.75(5.31) | 90.36(2.13) | 92.43(2.7) | 73.03(6.65) |
| spambase | 81.52(0.55) | 72.09(0.0) | 68.83(0.0) | 69.64(2.09) | 77.88(0.0) | **85.18(1.69)** | 83.36(0.0) | 72.39(6.9) |
| speech | 35.88(0.15) | 37.03(0.0) | 35.96(0.0) | 37.48(0.37) | 36.66(0.0) | 37.7(1.72) | 36.36(0.0) | 38.02(2.67) |
| stamps | 93.41(1.66) | 93.14(0.4) | 87.62(0.91) | 94.21(2.26) | 91.8(1.0) | 93.47(1.42) | 95.89(1.44) | 91.93(3.55) |
| thyroid | 98.54(0.06) | 93.81(0.0) | 97.55(0.0) | 93.15(1.52) | 98.65(0.0) | **98.96(0.2)** | 98.68(0.0) | 96.06(1.65) |
| vertebral | 54.43(1.97) | 26.34(2.49) | 41.95(4.78) | 64.13(2.15) | 40.09(3.97) | 45.64(3.83) | 57.67(3.58) | 31.66(5.28) |
| vowels | 78.72(4.01) | 52.82(0.0) | 61.47(0.0) | 85.32(1.16) | 53.31(0.0) | 61.83(0.62) | 82.21(0.0) | 55.52(8.1) |
| waveform | 72.93(0.9) | 72.36(0.0) | 59.44(0.0) | **76.95(1.07)** | 69.28(0.0) | 72.29(1.52) | 75.21(0.0) | 60.96(5.08) |
| wbc | 98.31(1.31) | 99.4(0.25) | 99.39(0.25) | 58.05(18.08) | 99.0(0.41) | 99.41(0.37) | 99.12(0.19) | 97.86(1.59) |
| wdbc | 98.73(0.43) | 99.18(0.21) | 96.72(0.71) | **99.63(0.18)** | 98.55(0.28) | 98.73(0.64) | 99.05(0.27) | 96.98(2.04) |
| wilt | 42.9(1.14) | 32.09(0.0) | 37.48(0.0) | 73.35(10.5) | 39.1(0.0) | 47.97(3.12) | 63.66(0.0) | 41.1(7.0) |
| wine | 97.75(0.59) | 86.37(4.37) | 73.86(5.42) | 97.93(0.87) | 95.63(2.63) | 93.92(1.79) | 99.19(0.22) | 90.94(4.75) |
| wpbc | 59.57(1.98) | 52.33(2.93) | 49.5(2.47) | 56.79(1.86) | 60.91(2.67) | 56.33(2.72) | 63.67(2.34) | 51.32(3.74) |
| yeast | **50.4(0.08)** | 38.88(0.0) | 44.64(0.0) | 46.4(1.33) | 42.88(0.0) | 41.8(0.75) | 44.74(0.0) | 46.51(5.8) |
| yelp | 63.8(0.07) | 60.21(0.0) | 57.39(0.0) | 67.06(0.12) | 59.95(0.0) | 61.07(0.53) | 68.07(0.0) | 56.26(3.78) |
| MNIST-C | 81.11(0.09) | 50.0(0.0) | 50.0(0.0) | **87.33(0.15)** | 70.43(0.0) | 76.75(1.45) | 84.11(0.0) | 69.39(5.31) |
| FashionMNIST | 89.1(0.15) | 50.0(0.0) | 50.0(0.0) | **91.67(0.11)** | 75.42(0.0) | 84.15(1.07) | 89.87(0.0) | 79.28(4.1) |
| CIFAR10 | 67.87(0.21) | 54.98(0.0) | 56.88(0.0) | **70.31(0.21)** | 57.89(0.0) | 64.04(0.93) | 67.53(0.0) | 61.62(4.35) |
| SVHN | 60.97(0.17) | 50.0(0.0) | 50.0(0.0) | **63.93(0.14)** | 54.65(0.0) | 58.99(0.88) | 61.69(0.0) | 54.5(4.02) |
| MVTec-AD | 79.96(2.02) | 50.0(0.0) | 50.0(0.0) | 80.45(2.2) | 75.97(1.84) | 77.38(1.95) | 81.51(1.82) | 72.31(3.37) |
| 20news | 57.1(1.23) | 52.92(0.43) | 54.11(0.24) | 60.25(0.7) | 53.57(0.35) | 54.91(1.17) | 57.36(0.66) | 53.45(4.0) |
| agnews | 62.82(0.08) | 55.05(0.0) | 55.11(0.0) | **74.64(0.09)** | 55.69(0.0) | 58.43(1.09) | 67.05(0.0) | 56.95(3.54) |

| | LOF | MCD | OCSVM | PCA | DAGMM | DeepSVDD | DROCC | GOAD | ICL |
|---|---|---|---|---|---|---|---|---|---|
| aloi | 48.76(0.0) | 48.54(0.35) | **54.29(0.0)** | 54.04(0.0) | 50.84(2.97) | 50.89(2.05) | 50.0(0.0) | 48.01(0.92) | 47.5(0.98) |
| amazon | 57.88(0.0) | 60.36(0.09) | 56.48(0.0) | 54.9(0.0) | 50.47(2.01) | 51.2(4.42) | 50.0(0.0) | 56.07(0.9) | 54.21(0.22) |
| annthyroid | 88.63(0.0) | 90.19(0.02) | 88.45(0.0) | 85.19(0.0) | 72.23(15.13) | 55.01(3.62) | 88.9(2.31) | 81.01(5.16) | 81.11(1.07) |
| backdoor | **95.33(0.25)** | 85.13(8.87) | 62.52(0.64) | 64.57(0.65) | 54.36(19.91) | 91.14(2.62) | 94.25(0.73) | 52.9(14.48) | 93.62(0.69) |
| breastw | 88.91(6.8) | 98.66(0.65) | 99.39(0.16) | 99.21(0.17) | 89.53(10.2) | 96.96(0.92) | 47.32(31.95) | 98.86(0.33) | 98.28(0.45) |
| campaign | 70.55(0.0) | 78.51(0.81) | 77.67(0.0) | 77.07(0.0) | 61.47(2.73) | 62.21(12.88) | 50.0(0.0) | 47.89(12.32) | **80.92(0.79)** |
| cardio | 92.21(0.0) | 82.82(0.85) | 95.61(0.0) | 96.54(0.01) | 77.92(8.85) | 65.43(4.37) | 62.14(23.76) | 96.01(0.27) | 80.01(2.12) |
| cardiotocography | 64.49(0.0) | 57.11(1.41) | 75.22(0.0) | 78.89(0.0) | 67.11(9.03) | 47.75(8.59) | 45.98(16.75) | 76.06(1.4) | 54.2(1.8) |
| celeba | 43.73(0.75) | **84.37(2.34)** | 79.79(0.77) | 80.53(0.7) | 63.81(4.28) | 56.17(22.46) | 68.89(1.11) | 43.8(10.49) | 72.21(0.82) |
| census | 58.46(1.06) | **74.14(1.94)** | 70.02(0.21) | 70.51(0.21) | 52.24(1.19) | 54.16(4.33) | 55.36(3.62) | 35.24(4.19) | 70.56(0.35) |
| cover | **99.18(0.1)** | 70.02(0.66) | 96.17(0.11) | 94.41(0.14) | 75.94(14.06) | 49.12(14.74) | 95.79(0.69) | 13.83(13.34) | 89.34(4.02) |
| donors | 96.97(0.24) | 81.93(10.78) | 92.09(0.22) | 88.12(0.57) | 62.15(16.19) | 72.95(17.81) | 74.18(22.09) | 33.57(16.0) | **99.9(0.05)** |
| fault | 47.42(0.0) | 59.44(3.79) | 57.21(0.0) | 55.87(0.0) | 52.85(7.19) | 54.31(1.62) | 55.73(5.33) | 58.89(0.61) | 60.63(0.5) |
| fraud | 94.35(1.41) | 91.1(1.79) | **95.61(0.69)** | **95.38(0.68)** | 85.33(6.76) | 83.13(6.6) | 50.0(0.0) | 69.75(21.3) | 92.78(1.46) |
| glass | 88.82(1.98) | 79.71(1.37) | 69.73(5.89) | 73.44(2.22) | 65.33(15.53) | 83.67(16.2) | 64.89(23.64) | 59.03(12.64) | **99.44(0.58)** |
| hepatitis | 66.92(7.01) | 80.64(4.23) | 90.58(1.79) | 84.48(2.29) | 70.22(6.66) | 99.57(0.24) | 51.8(17.93) | 84.5(3.25) | **99.94(0.13)** |
| http | 99.98(0.03) | 99.95(0.01) | **100.0(0.0)** | 99.95(0.01) | 91.78(17.91) | 61.31(51.49) | 50.0(0.0) | 99.68(0.13) | 98.24(3.45) |
| imdb | 49.57(0.0) | 51.24(0.18) | 48.72(0.0) | 47.97(0.0) | 48.6(0.38) | 49.97(5.71) | 51.35(2.05) | 48.46(0.65) | **52.34(0.49)** |
| internetads | 71.72(0.0) | 47.73(0.01) | 65.63(0.0) | 65.12(0.0) | 49.47(5.05) | 72.96(3.24) | 53.4(7.55) | 65.65(0.21) | 72.2(0.55) |
| ionosphere | 94.29(2.2) | 95.4(0.64) | 96.32(0.93) | 89.11(1.31) | 73.95(5.98) | 97.2(1.26) | 61.14(28.54) | 91.54(3.05) | **98.98(0.32)** |
| landsat | 66.58(0.0) | 56.78(6.08) | 47.98(0.0) | 43.9(0.0) | 56.27(3.46) | 59.44(1.41) | 53.86(2.57) | 40.52(2.32) | 65.13(0.44) |
| letter | 44.83(0.0) | 31.47(4.16) | 32.17(0.0) | 30.3(0.0) | 38.97(8.38) | 36.4(3.05) | 55.26(11.04) | 31.08(0.55) | 42.68(1.17) |
| lymphography | 98.21(0.75) | 98.88(0.55) | **100.0(0.0)** | 99.86(0.05) | 94.94(3.85) | 99.73(0.3) | 32.42(37.75) | 99.89(0.08) | **100.0(0.0)** |
| magic.gamma | 83.4(0.0) | 73.67(0.12) | 74.25(0.0) | 70.64(0.0) | 59.23(4.32) | 62.97(1.07) | 78.83(0.66) | 69.46(2.39) | 75.56(0.42) |
| mammography | 85.52(0.0) | 72.87(0.64) | 88.63(0.0) | 89.93(0.0) | 76.03(14.65) | 71.5(7.4) | 81.82(1.94) | 69.94(8.59) | 71.87(9.11) |
| mnist | 92.93(0.0) | 88.3(1.03) | 90.56(0.0) | 90.21(0.0) | 72.19(7.16) | 66.37(11.03) | 83.13(1.64) | 90.07(0.35) | 90.11(1.13) |
| musk | **100.0(0.0)** | 93.91(2.55) | **100.0(0.0)** | **100.0(0.0)** | 95.01(4.27) | 99.99(0.01) | 32.99(33.18) | **100.0(0.0)** | 99.37(0.63) |
| optdigits | **96.65(0.0)** | 64.86(0.92) | 63.38(0.0) | 58.17(0.0) | 40.04(20.45) | 39.45(18.65) | 85.25(2.9) | 67.46(4.94) | **97.18(0.8)** |
| pageblocks | **91.3(0.0)** | 87.07(0.02) | 88.58(0.0) | 86.12(0.0) | 82.8(10.44) | 78.39(1.53) | **92.33(1.18)** | 88.05(1.19) | 88.39(0.77) |
| pendigits | 99.05(0.0) | 83.69(0.09) | 96.36(0.0) | 94.37(0.0) | 56.49(21.83) | 46.29(11.35) | 75.91(13.38) | 89.97(2.03) | 96.71(0.83) |
| pima | 70.53(2.19) | 73.64(1.37) | 71.53(1.79) | 72.28(1.99) | 54.54(6.0) | 57.99(2.85) | 47.53(16.52) | 62.31(13.7) | **79.68(3.06)** |
| satellite | 80.3(0.0) | 72.76(3.68) | 73.91(0.0) | 66.63(0.0) | 72.79(2.01) | 76.19(2.66) | 73.38(4.44) | 68.76(0.92) | 85.15(0.48) |
| satimage-2 | 99.38(0.0) | **99.92(0.0)** | 99.61(0.0) | 98.17(0.0) | 91.82(4.43) | 92.94(2.84) | 99.22(0.42) | 98.99(0.06) | 99.48(0.22) |
| shuttle | **99.98(0.0)** | 98.98(0.0) | 99.62(0.0) | 99.36(0.0) | 84.58(18.68) | 99.79(0.07) | 50.0(0.0) | 70.44(16.28) | 99.92(0.04) |
| skin | 86.34(1.77) | 88.37(0.26) | 90.25(0.24) | 59.73(0.31) | 67.91(30.02) | 59.95(4.47) | 89.48(1.07) | 64.95(2.25) | 6.58(0.63) |
| smtp | 93.42(2.48) | **94.87(0.84)** | 84.65(4.49) | 81.81(7.32) | 87.08(5.31) | 85.24(6.6) | 57.13(15.93) | 78.78(13.12) | 74.36(7.07) |
| spambase | 73.23(0.0) | 80.69(3.02) | 81.7(0.0) | 81.4(0.0) | 69.41(4.42) | 70.24(5.02) | 75.37(4.45) | 81.78(0.36) | 83.53(0.45) |
| speech | 37.53(0.0) | 38.81(0.36) | 36.57(0.0) | 36.38(0.0) | **50.66(3.9)** | **48.88(2.88)** | **48.96(2.21)** | 36.63(1.14) | 48.86(2.72) |
| stamps | 93.74(2.36) | 84.93(2.01) | 93.72(1.74) | 92.7(1.68) | 80.11(11.56) | 71.09(3.68) | 50.15(22.93) | 81.46(15.36) | 96.68(1.09) |
| thyroid | 92.72(0.0) | 98.49(0.01) | 98.56(0.0) | 98.55(0.0) | 91.08(6.68) | 88.77(3.69) | 94.96(1.55) | 95.15(0.85) | 95.4(0.98) |
| vertebral | 64.3(1.31) | 47.1(1.82) | 50.47(2.23) | 42.08(3.49) | 50.6(11.93) | 44.79(1.71) | 43.84(24.28) | 46.73(8.74) | **79.19(5.06)** |
| vowels | 86.3(0.0) | 27.66(0.28) | 75.91(0.0) | 52.29(0.0) | 42.55(11.56) | 55.73(4.43) | 54.74(21.32) | 68.49(3.71) | 85.1(2.13) |
| waveform | **76.0(0.0)** | 58.39(0.03) | 70.44(0.0) | 64.68(0.0) | 51.89(7.39) | 59.94(2.58) | 67.7(5.0) | 64.99(3.1) | 68.68(3.66) |
| wbc | 80.52(4.47) | 98.88(1.02) | **99.63(0.19)** | 99.35(0.13) | 86.76(15.33) | 91.44(3.64) | 44.24(28.21) | 99.14(0.2) | **99.66(0.36)** |
| wdbc | **99.62(0.21)** | 97.03(0.47) | 99.34(0.23) | 99.14(0.22) | 73.78(25.27) | 99.31(0.4) | 40.08(33.97) | 98.96(0.25) | **99.78(0.27)** |
| wilt | 68.81(0.0) | 81.72(0.01) | 34.81(0.0) | 26.07(0.0) | 41.81(7.29) | 34.41(1.7) | 49.49(12.62) | 51.38(5.33) | 76.42(3.46) |
| wine | 98.36(0.38) | 97.28(1.8) | 97.82(0.45) | 93.79(1.58) | 66.17(39.61) | 92.16(4.34) | 43.8(32.33) | 94.11(1.86) | 99.87(0.29) |
| wpbc | 57.36(2.01) | 63.36(0.93) | 53.37(2.44) | 52.54(2.3) | 46.99(2.56) | 82.67(5.47) | 43.78(4.34) | 51.39(5.74) | **96.61(1.17)** |
| yeast | 45.79(0.0) | 43.05(0.1) | 44.84(0.0) | 43.24(0.0) | **51.03(3.92)** | 47.62(5.96) | 48.42(5.46) | **52.53(3.88)** | 48.98(2.39) |
| yelp | 67.2(0.0) | 66.15(0.04) | 62.08(0.0) | 59.16(0.0) | 49.87(1.25) | 49.9(3.3) | 50.67(1.15) | 61.07(0.98) | 55.79(0.5) |
| MNIST-C | **87.21(0.0)** | 75.26(1.24) | 79.55(0.0) | 78.35(0.0) | 63.71(6.37) | 64.7(4.6) | 57.2(11.1) | 79.34(0.41) | 85.62(0.52) |
| FashionMNIST | **91.6(0.0)** | 84.37(1.11) | 88.16(0.0) | 87.6(0.0) | 70.8(4.89) | 75.45(2.22) | 51.58(14.44) | 88.03(0.24) | 90.56(0.27) |
| CIFAR10 | **70.3(0.0)** | 65.15(0.58) | 67.79(0.0) | 67.42(0.0) | 53.97(2.96) | 56.12(2.2) | 49.63(3.46) | 67.53(0.72) | 63.6(0.83) |
| SVHN | **63.82(0.0)** | 58.87(0.65) | 61.25(0.0) | 60.79(0.0) | 53.36(2.12) | 53.92(3.05) | 49.99(2.47) | 60.75(0.49) | 61.7(0.5) |
| MVTec-AD | 80.36(2.14) | 86.75(2.12) | 77.44(2.03) | 76.37(1.91) | 64.69(5.63) | 89.55(2.19) | 60.52(11.42) | 77.06(2.1) | **94.75(0.84)** |
| 20news | 60.19(0.62) | **62.85(1.63)** | 56.25(0.62) | 54.39(0.41) | 51.48(4.08) | 55.64(4.17) | 52.77(4.09) | 54.92(1.04) | **61.3(1.05)** |
| agnews | **74.58(0.0)** | 67.98(0.25) | 60.64(0.0) | 56.9(0.0) | 51.03(3.6) | 49.83(5.63) | 49.65(0.76) | 59.87(0.86) | 62.55(0.47) |

| | PlanarFlow | VAE | GANomaly | SLAD | DIF | DDPM | DTE-NP | DTE-IG | DTE-C |
|---|---|---|---|---|---|---|---|---|---|
| aloi | 48.52(2.34) | 54.04(0.0) | 53.94(1.65) | 50.76(0.29) | 51.1(0.0) | 49.91(0.35) | 51.19(0.46) | 50.87(1.17) | 50.44(0.19) |
| amazon | 49.94(2.12) | 54.9(0.0) | 53.42(0.94) | 52.01(0.06) | 51.43(0.34) | 55.09(0.1) | **60.82(0.0)** | 51.93(3.5) | 56.71(2.15) |
| annthyroid | 93.19(2.14) | 85.44(0.0) | 67.52(5.54) | 93.34(0.36) | 88.36(0.0) | 88.82(1.34) | 92.9(0.25) | 87.6(4.46) | **97.52(0.15)** |
| backdoor | 76.03(11.56) | 64.67(0.74) | 87.17(1.43) | 50.0(0.0) | 83.73(0.76) | 80.93(0.56) | 93.31(1.7) | 94.02(1.46) | 91.65(1.73) |
| breastw | 97.93(0.82) | 99.22(0.18) | 94.75(2.73) | **99.53(0.13)** | 56.34(11.9) | 98.7(0.43) | 99.28(0.12) | 78.65(11.1) | 92.78(1.75) |
| campaign | 69.75(3.75) | 77.07(0.0) | 69.21(3.28) | 76.75(0.16) | 57.87(0.0) | 74.51(0.42) | 78.79(0.25) | 74.81(1.69) | 77.95(1.11) |
| cardio | 88.9(0.93) | **96.55(0.0)** | 86.06(4.31) | 83.05(1.1) | 68.25(0.0) | 86.94(1.96) | 91.8(0.59) | 73.79(14.09) | 87.26(1.0) |
| cardiotocography | 69.88(5.65) | 78.9(0.0) | 62.77(8.62) | 47.32(0.31) | 41.79(0.0) | 54.54(2.96) | 63.76(1.88) | 52.44(3.67) | 60.13(2.59) |
| celeba | 71.64(7.91) | 80.32(0.48) | 52.25(11.79) | 67.43(1.64) | 66.69(3.55) | 78.56(1.99) | 70.4(0.37) | 74.51(2.39) | **82.18(2.38)** |
| census | 59.33(2.89) | 70.52(0.22) | 68.07(3.66) | 57.91(10.84) | 61.45(2.04) | 70.15(0.23) | 72.1(0.4) | 61.79(4.94) | 69.62(0.91) |
| cover | 47.52(8.02) | 94.35(0.15) | 76.35(19.11) | 73.97(13.8) | 57.69(5.24) | 98.35(0.66) | 97.73(0.55) | 95.83(1.55) | 97.76(1.28) |
| donors | 91.64(3.52) | 88.6(0.25) | 75.34(11.69) | 88.51(5.51) | 90.04(1.79) | 82.5(1.83) | 99.26(0.29) | 99.25(0.6) | 98.15(0.37) |
| fault | 57.51(5.15) | 55.87(0.0) | 59.5(5.14) | **63.93(0.19)** | 62.31(0.0) | 61.09(1.16) | 58.64(0.65) | 59.42(1.46) | 59.46(1.45) |
| fraud | 90.72(2.26) | **95.47(0.75)** | 93.25(2.85) | 94.58(1.01) | 82.58(2.27) | 93.65(0.92) | **95.64(1.05)** | 90.79(3.24) | 93.52(1.53) |
| glass | 85.33(6.2) | 72.55(1.66) | 79.77(8.72) | 86.04(5.69) | 96.45(1.39) | 66.67(13.63) | 89.64(3.54) | 98.53(0.86) | 92.42(2.3) |
| hepatitis | 95.8(1.67) | 84.84(2.27) | 87.39(7.55) | **99.93(0.15)** | 96.07(2.32) | 97.74(1.18) | 93.22(3.9) | **99.93(0.15)** | 98.78(0.88) |
| http | 99.38(0.08) | 99.94(0.01) | 50.14(34.85) | 99.91(0.08) | 99.36(0.07) | **100.0(0.0)** | 99.98(0.03) | 80.72(43.08) | 99.45(0.1) |
| imdb | 49.23(2.75) | 47.97(0.0) | 51.58(0.71) | 51.26(0.1) | 51.4(0.49) | 47.91(0.1) | 50.43(0.0) | 50.97(3.41) | 48.05(2.22) |
| internetads | 70.87(0.84) | 65.12(0.0) | 69.86(0.23) | 75.94(0.11) | 49.33(0.34) | 65.76(0.06) | 69.96(2.22) | 71.52(3.89) | **77.57(1.54)** |
| ionosphere | 96.86(1.21) | 89.76(1.2) | 93.89(2.03) | 98.21(0.6) | 93.59(2.06) | 94.6(0.85) | 97.77(1.39) | 95.15(3.6) | 95.42(0.58) |
| landsat | 50.85(2.13) | 54.22(9.45) | 55.34(10.03) | 65.03(0.18) | 56.56(0.0) | 51.37(1.0) | 68.2(1.75) | 44.72(5.52) | 52.79(1.63) |
| letter | 38.73(3.53) | 30.23(0.0) | 34.02(0.74) | 36.8(0.41) | **74.07(0.0)** | 38.05(1.12) | 34.38(0.98) | 39.86(2.29) | 36.72(0.95) |
| lymphography | 99.58(0.51) | 99.88(0.09) | 99.09(0.86) | **100.0(0.01)** | 99.84(0.22) | 99.94(0.09) | 99.93(0.09) | 99.98(0.05) | 98.99(0.4) |
| magic.gamma | 74.12(2.75) | 70.64(0.0) | 59.18(1.6) | 72.0(0.01) | 63.86(0.0) | 85.97(1.08) | 83.57(0.76) | 86.46(1.12) | **87.5(0.9)** |
| mammography | 78.93(5.71) | 89.58(0.17) | 85.54(7.43) | 74.51(0.67) | 73.87(0.2) | 81.01(2.04) | 87.62(0.09) | 84.64(3.47) | 86.42(1.72) |
| mnist | 81.9(2.67) | 90.21(0.0) | 77.9(6.44) | 89.73(0.44) | 50.21(0.0) | 87.27(3.22) | **94.02(0.42)** | 80.78(5.91) | 87.43(2.48) |
| musk | 76.65(18.72) | **100.0(0.0)** | **100.0(0.0)** | **100.0(0.0)** | 97.76(0.0) | **100.0(0.0)** | **100.0(0.0)** | 94.22(12.93) | **100.0(0.0)** |
| optdigits | 34.12(8.29) | 58.17(0.0) | 74.3(12.58) | 95.28(0.17) | 48.64(0.0) | 90.76(2.12) | 94.28(1.67) | 79.81(9.35) | 82.38(2.68) |
| pageblocks | 84.85(1.19) | 86.16(0.0) | 72.83(9.94) | 87.86(0.01) | 87.39(0.0) | 86.93(0.42) | 89.32(0.26) | 85.66(1.77) | 89.89(0.55) |
| pendigits | 83.45(6.13) | 94.5(0.0) | 67.93(21.6) | 94.61(0.41) | 89.71(0.0) | 98.11(0.23) | 99.61(0.17) | 96.96(1.63) | 97.79(0.68) |
| pima | 72.17(2.71) | 73.18(1.93) | 60.45(3.01) | 60.62(4.21) | 55.19(6.92) | 70.27(2.28) | **81.5(2.57)** | 68.59(3.93) | 69.88(1.95) |
| satellite | 72.3(1.73) | 74.14(0.25) | 79.87(0.56) | **87.49(0.1)** | 66.91(0.0) | 77.7(0.53) | 82.11(0.67) | 76.52(2.68) | 78.61(0.7) |
| satimage-2 | 96.67(0.57) | 98.98(0.04) | 97.57(1.19) | 99.77(0.0) | 86.9(0.0) | 99.62(0.16) | 99.67(0.0) | 95.34(1.84) | 99.34(0.07) |
| shuttle | 86.5(5.68) | 99.35(0.0) | 97.53(0.74) | 99.9(0.01) | 99.76(0.0) | 99.91(0.01) | 99.93(0.02) | 99.86(0.12) | 99.75(0.0) |
| skin | 91.27(7.09) | 66.05(0.19) | 48.48(2.87) | 91.05(2.61) | 87.55(0.76) | 88.74(4.4) | 98.86(0.46) | 98.71(1.13) | 91.77(0.22) |
| smtp | 84.22(6.88) | 81.93(5.67) | 54.55(5.98) | 92.15(1.87) | **95.51(1.36)** | **95.43(1.26)** | 92.98(2.91) | 81.64(9.87) | **95.27(1.28)** |
| spambase | 82.26(3.35) | 81.4(0.0) | 82.57(1.34) | **84.86(0.2)** | 41.31(0.0) | 64.54(0.8) | 83.74(0.69) | 77.5(3.37) | 83.01(0.41) |
| speech | 48.56(4.54) | 36.38(0.0) | 38.7(3.43) | 41.37(0.94) | **52.29(3.42)** | 36.96(0.86) | 41.37(0.0) | 39.57(1.54) | 38.17(0.57) |
| stamps | 87.3(10.69) | 93.28(1.28) | 66.2(7.84) | 81.97(8.29) | 84.92(4.03) | 91.84(4.15) | **97.87(0.37)** | 93.38(3.31) | 91.6(2.04) |
| thyroid | 98.42(0.51) | 98.55(0.0) | 94.78(3.22) | 95.27(0.24) | 96.29(0.0) | 97.95(0.22) | 98.63(0.04) | 89.43(11.73) | 98.74(0.14) |
| vertebral | 49.78(7.82) | 42.63(2.75) | 50.67(10.71) | 44.96(9.25) | 57.21(3.04) | 70.67(6.28) | 54.3(15.47) | **74.63(7.58)** | 66.41(1.9) |
| vowels | 54.59(10.55) | 52.12(0.02) | 63.11(9.82) | 85.02(0.02) | **88.48(0.0)** | 86.38(1.94) | 81.42(1.44) | 85.68(3.16) | 86.93(2.25) |
| waveform | 64.8(2.42) | 64.84(0.0) | 75.95(7.58) | 48.92(0.08) | 50.63(0.0) | 62.17(2.64) | 74.48(0.0) | 73.68(2.68) | 65.21(1.05) |
| wbc | 95.97(0.81) | 99.25(0.27) | 95.91(3.0) | **99.78(0.18)** | 97.46(1.82) | 99.2(0.39) | 99.54(0.28) | 90.98(11.64) | 80.53(7.15) |
| wdbc | 98.86(0.76) | 99.14(0.36) | 96.22(3.1) | 99.49(0.25) | 68.03(7.24) | 99.3(0.17) | **99.52(0.38)** | **99.6(0.31)** | 98.48(0.77) |
| wilt | 74.62(4.18) | 35.41(0.0) | 44.01(5.48) | 61.76(0.11) | 55.02(0.0) | 71.66(0.81) | 62.91(5.59) | **93.75(3.23)** | 85.1(1.13) |
| wine | 95.38(2.4) | 94.25(1.47) | 74.32(28.64) | **100.0(0.0)** | 91.09(2.25) | 99.61(0.09) | 99.44(0.97) | 99.95(0.11) | 99.97(0.04) |
| wpbc | 57.46(2.93) | 54.42(2.76) | 60.11(5.48) | **95.53(2.21)** | 82.51(3.26) | 66.49(3.17) | 83.16(13.46) | 70.71(9.06) | 68.88(2.7) |
| yeast | 45.07(3.33) | 42.39(0.01) | 47.64(6.7) | 48.69(0.11) | 38.44(0.0) | 49.13(2.85) | 44.58(0.33) | 48.58(2.89) | 47.08(1.1) |
| yelp | 53.6(2.03) | 59.14(0.04) | 56.11(1.6) | 54.88(0.19) | 48.36(0.23) | 59.3(0.09) | **68.66(0.0)** | 57.32(4.3) | 59.94(5.32) |
| MNIST-C | 71.21(1.5) | 78.35(0.01) | 80.08(1.16) | 83.51(0.11) | 54.05(2.27) | 80.12(0.14) | 84.74(0.0) | 79.92(3.93) | 86.1(0.84) |
| FashionMNIST | 82.19(0.96) | 87.6(0.0) | 89.43(0.52) | 89.94(0.04) | 65.53(1.77) | 88.52(0.07) | 90.14(0.0) | 84.32(2.44) | 90.21(0.55) |
| CIFAR10 | 62.75(1.09) | 67.42(0.0) | 67.19(0.84) | 66.6(0.1) | 52.06(1.59) | 67.91(0.13) | 67.82(0.0) | 62.4(3.33) | 68.53(1.59) |
| SVHN | 58.87(0.87) | 60.79(0.0) | 61.25(0.77) | 60.87(0.07) | 52.98(0.67) | 61.37(0.08) | 62.13(0.0) | 59.19(2.5) | 62.91(1.1) |
| MVTec-AD | 72.8(2.39) | 76.21(1.81) | 81.1(1.8) | 93.24(1.09) | 81.99(2.78) | 78.0(1.99) | 89.66(1.51) | 85.88(3.63) | 89.39(2.0) |
| 20news | 51.59(3.0) | 54.59(0.74) | 59.68(1.87) | 59.47(0.8) | 53.18(2.56) | 54.88(0.52) | 59.97(0.74) | 58.28(6.15) | **64.3(3.17)** |
| agnews | 50.15(1.19) | 56.9(0.0) | 58.63(1.14) | 58.43(0.07) | 52.02(0.89) | 57.82(0.12) | 67.98(0.0) | 56.55(4.15) | 68.16(3.22) |

Table 14: Average F1 score and standard deviations over five seeds for the semi-supervised setting on ADBench.

| | CBLOF | COPOD | ECOD | FeatureBagging | HBOS | IForest | kNN | LODA | LOF | MCD | OCSVM | PCA |
|---|---|---|---|---|---|---|---|---|---|---|---|---|
| aloi | 53.69(0.15) | 49.5 aloi | 6.74(0.08) | 4.58(0.0) | 4.44(0.0) | **8.93(0.57)** | 7.43(0.0) | 4.2(0.26) | 5.9(0.0) | 6.6(1.58) | 8.16(0.0) | 3.41(0.14) |
| 5.12(0.68) | 4.2(0.2) | | | | | | | | | | | |
| amazon | 11.52(0.5) | 11.4(0.0) | 10.0(0.0) | 10.0(0.14) | 10.6(0.0) | 11.28(0.64) | 11.4(0.0) | 10.64(1.18) | 10.0(0.0) | 11.32(0.23) | **12.0(0.0)** | 11.0(0.0) |
| annthyroid | 56.7(3.3) | 31.65(0.0) | 38.39(0.0) | 50.67(5.76) | 35.96(0.0) | 55.02(4.22) | 61.99(0.0) | 46.78(5.94) | 49.63(0.0) | 50.37(0.0) | 53.56(0.0) | 50.0(0.0) |
| backdoor | 7.74(1.1) | 0.0(0.0) | 0.0(0.0) | 58.53(7.63) | 6.93(0.61) | 4.07(2.4) | 52.01(1.92) | 4.64(4.15) | 72.42(2.18) | 19.49(27.36) | 7.94(0.85) | 8.3(1.0) |
| breastw | 95.78(0.27) | 96.41(0.34) | 94.63(0.53) | 60.99(14.7) | **96.93(0.34)** | **96.94(0.46)** | 95.77(0.19) | 95.67(0.47) | 85.4(5.93) | 95.84(0.67) | **96.66(1.1)** | 95.78(0.45) |
| campaign | 49.29(0.2) | 49.27(0.0) | 48.38(0.0) | 37.15(6.73) | 47.91(0.0) | 43.7(0.91) | 50.37(0.0) | 30.74(5.47) | 42.24(0.0) | 48.33(1.62) | 49.59(0.0) | 48.84(0.0) |
| cardio | 70.0(5.04) | 70.45(0.0) | 73.86(0.0) | 62.95(3.04) | 56.25(0.0) | 67.5(3.32) | 61.93(0.0) | 63.41(3.82) | 62.5(0.0) | 59.09(0.0) | 70.45(0.0) | **76.14(0.0)** |
| cardiotocography | 51.42(3.49) | 48.28(0.0) | **62.88(0.0)** | 48.41(1.57) | 41.42(0.0) | 56.14(2.75) | 46.35(0.0) | 55.11(7.7) | 48.28(0.0) | 36.48(1.78) | 57.94(0.0) | 61.59(0.0) |
| celeba | 25.32(7.14) | 22.78(0.91) | 22.81(0.78) | 2.92(0.89) | 22.68(0.93) | 17.33(2.29) | 17.19(0.83) | 13.26(8.39) | 1.91(0.47) | 25.12(4.44) | **27.37(0.74)** | **27.17(0.49)** |
| census | 21.46(0.28) | 0.0(0.0) | 0.0(0.0) | 3.47(1.3) | 10.77(0.62) | 10.54(1.4) | 22.52(0.51) | 14.15(8.41) | 13.09(0.42) | **29.45(3.37)** | 20.67(0.38) | 20.82(0.33) |
| cover | 13.99(1.06) | 18.82(0.81) | 24.46(1.11) | 79.41(10.6) | 10.75(1.26) | 11.61(1.24) | 65.1(2.15) | 24.19(12.26) | **82.4(2.16)** | 3.44(0.31) | 24.55(1.58) | 16.24(1.53) |
| donors | 48.48(1.33) | 41.37(0.99) | 44.6(1.04) | 56.11(14.4) | 24.36(3.68) | 43.46(3.54) | 94.91(0.67) | 21.04(27.81) | 74.47(2.04) | 33.32(14.21) | 39.52(1.55) | 37.3(1.8) |
| fault | 56.4(0.87) | 50.82(0.0) | 51.56(0.0) | 50.4(0.85) | 53.64(0.0) | 53.64(1.34) | 55.57(0.0) | 51.59(1.84) | 50.67(0.0) | 56.37(1.62) | 55.13(0.0) | 55.27(0.0) |
| fraud | 34.39(0.6) | 46.16(3.48) | 37.81(2.09) | 67.65(4.19) | 41.51(4.73) | 28.03(4.09) | 45.22(4.87) | 45.06(11.59) | 59.47(4.5) | 56.19(3.88) | 41.52(5.56) | 33.26(1.7) |
| glass | 23.75(14.31) | 19.06(8.56) | 15.77(8.65) | 22.45(7.76) | 27.68(10.39) | 16.18(7.01) | 25.87(13.76) | 14.6(4.71) | 20.46(8.79) | 16.25(9.09) | 14.97(7.84) | 15.77(8.65) |
| hepatitis | 66.93(8.55) | 53.78(0.81) | 37.6(3.88) | 41.27(12.6) | 58.08(1.24) | 54.0(5.54) | 81.29(5.04) | 46.76(9.9) | 41.95(10.62) | 49.49(11.04) | 66.61(4.93) | 60.56(7.59) |
| http | 91.29(1.52) | 2.16(1.15) | 2.05(1.29) | 0.0(0.0) | 3.64(3.65) | 25.8(22.46) | **100.0(0.0)** | 1.05(0.96) | 96.78(3.13) | 93.05(1.51) | 99.78(0.3) | 92.71(1.43) |
| imdb | 6.96(0.09) | 6.6(0.0) | 5.0(0.0) | 6.56(0.3) | 6.4(0.0) | 6.2(0.49) | 5.4(0.0) | 7.04(1.08) | 6.4(0.0) | 7.44(0.22) | 5.8(0.0) | 5.6(0.0) |
| internetads | 45.76(0.24) | 50.0(0.0) | 50.0(0.0) | 54.35(3.71) | 27.17(0.0) | 26.41(4.44) | 51.9(0.0) | 41.36(2.56) | 54.62(0.0) | 33.42(0.0) | 46.2(0.0) | 45.65(0.0) |
| ionosphere | 91.96(2.14) | 69.53(2.51) | 64.5(2.11) | 87.65(2.26) | 69.49(1.68) | 83.43(3.5) | 90.47(2.17) | 77.34(4.54) | 87.53(2.54) | 88.64(1.2) | **92.62(1.51)** | 78.99(2.57) |
| landsat | 38.33(0.19) | 33.83(0.0) | 30.76(0.0) | **53.97(0.17)** | 52.14(0.0) | 43.27(1.34) | 51.46(0.0) | 36.89(4.89) | 53.64(0.0) | 47.7(9.54) | 38.56(0.0) | 33.98(0.0) |
| letter | 1.0(0.0) | 4.0(0.0) | 9.0(0.0) | 8.6(1.34) | 6.0(0.0) | 3.8(1.1) | 1.0(0.0) | 1.2(0.45) | 10.0(0.0) | 2.4(0.89) | 1.0(0.0) | 1.0(0.0) |
| lymphography | 89.28(1.85) | 88.47(4.27) | 86.67(6.42) | 65.34(17.16) | 88.49(3.88) | 85.05(4.72) | 94.5(6.47) | 24.05(20.79) | 74.87(7.44) | 83.73(4.92) | **100.0(0.0)** | 90.86(4.08) |
| magic.gamma | 69.24(0.01) | 62.86(0.0) | 59.72(0.0) | 76.85(0.77) | 67.19(0.0) | 69.64(1.25) | 76.17(0.0) | 65.48(1.31) | 76.08(0.0) | 67.89(0.12) | 68.38(0.0) | 65.19(0.0) |
| mammography | 49.23(0.0) | 52.69(0.0) | **53.08(0.0)** | 39.38(0.97) | 16.92(0.0) | 39.23(2.48) | 40.38(0.0) | 47.92(2.01) | 38.46(0.0) | 2.62(0.63) | 41.92(0.0) | 44.62(0.0) |
| mnist | 65.94(0.44) | 0.0(0.0) | 0.0(0.0) | 68.89(1.44) | 24.14(0.0) | 52.6(4.64) | **71.86(0.0)** | 33.8(7.68) | 71.43(0.0) | 55.97(2.61) | 64.29(0.0) | 63.86(0.0) |
| musk | **100.0(0.0)** | 87.63(0.0) | 92.78(0.0) | **100.0(0.0)** | **100.0(0.0)** | 35.88(24.78) | **100.0(0.0)** | 90.72(5.41) | **100.0(0.0)** | 53.61(14.3) | **100.0(0.0)** | **100.0(0.0)** |
| optdigits | 1.6(0.37) | 0.0(0.0) | 0.0(0.0) | 47.33(4.45) | 40.67(0.0) | 12.8(6.28) | 21.33(0.0) | 1.07(1.67) | **53.33(0.0)** | 0.0(0.0) | 0.67(0.0) | 0.67(0.0) |
| pageblocks | 65.29(0.28) | 36.67(0.0) | 49.22(0.0) | 63.45(1.67) | 12.35(0.0) | 42.63(2.29) | 59.02(0.0) | 46.82(3.32) | 65.88(0.0) | 57.57(0.11) | 55.69(0.0) | 46.86(0.0) |
| pendigits | 49.23(0.7) | 35.26(0.0) | 43.59(0.0) | 83.33(3.51) | 41.03(0.0) | 57.95(4.75) | **90.38(0.0)** | 42.05(6.31) | 76.28(0.0) | 14.36(0.35) | 53.21(0.0) | 44.23(0.0) |
| pima | 68.78(2.76) | 63.16(2.25) | 58.54(1.89) | 68.43(3.04) | 69.6(2.76) | 69.57(2.68) | 70.56(2.49) | 61.14(5.31) | 66.75(3.0) | 70.86(2.2) | 68.59(2.02) | 69.3(2.92) |
| satellite | 64.02(0.07) | 60.71(0.0) | 56.63(0.0) | 72.6(0.08) | 75.98(0.0) | 67.12(0.64) | 71.81(0.0) | 65.01(1.17) | 72.64(0.0) | 63.15(5.11) | 67.34(0.0) | 62.67(0.0) |
| satimage-2 | 92.96(0.0) | 80.28(0.0) | 78.87(0.0) | 84.51(1.0) | 83.1(0.0) | 89.58(1.61) | 90.14(0.0) | 88.73(1.0) | 81.69(0.0) | **95.77(0.0)** | 91.55(0.0) | 87.32(0.0) |
| shuttle | 96.31(0.16) | 96.07(0.0) | 91.8(0.0) | 30.91(41.52) | 95.07(0.0) | 96.71(0.53) | 98.23(0.0) | 53.11(44.55) | 98.41(0.0) | 84.62(0.0) | 96.5(0.0) | 95.78(0.0) |
| skin | 81.39(0.38) | 20.2(0.58) | 22.0(0.36) | 59.02(1.62) | 58.3(0.29) | 78.06(0.72) | **96.35(0.62)** | 55.77(10.14) | 70.8(2.09) | 76.76(0.35) | 80.02(0.42) | 37.91(0.84) |
| smtp | **69.5(4.43)** | 0.0(0.0) | **69.5(4.43)** | 0.0(0.0) | 0.0(0.0) | 0.0(0.0) | **69.5(4.43)** | 8.76(5.05) | **65.82(5.88)** | 0.0(0.0) | **69.5(4.43)** | **69.5(4.43)** |
| spambase | 78.78(0.49) | 71.59(0.0) | 69.51(0.0) | 71.52(1.85) | 74.93(0.0) | 80.49(1.34) | 80.52(0.0) | 71.03(5.1) | 73.97(0.0) | 77.7(2.09) | 78.56(0.0) | 78.5(0.0) |
| speech | 1.64(0.0) | 3.28(0.0) | 3.28(0.0) | 2.95(0.73) | **4.92(0.0)** | 3.93(2.49) | 3.28(0.0) | 2.3(1.87) | 3.28(0.0) | 2.62(1.47) | 3.28(0.0) | 3.28(0.0) |
| stamps | 64.39(11.86) | 67.23(4.51) | 49.48(5.09) | 64.7(12.55) | 57.55(6.45) | 63.62(8.81) | 75.47(9.45) | 60.16(12.64) | 63.52(13.2) | 30.97(8.32) | 63.44(9.99) | 57.86(9.09) |
| thyroid | 74.19(1.08) | 30.11(0.0) | 59.14(0.0) | 40.22(11.57) | **77.42(0.0)** | **80.43(3.08)** | 75.27(0.0) | 70.75(5.18) | 52.69(0.0) | 73.12(0.0) | 75.27(0.0) | 74.19(0.0) |
| vertebral | 25.73(3.79) | 0.28(0.63) | 12.61(2.46) | 33.33(8.67) | 9.53(5.46) | 15.84(2.0) | 23.82(5.02) | 8.42(4.46) | 33.68(6.21) | 17.32(5.22) | 20.37(3.6) | 13.93(1.31) |
| vowels | 19.6(2.61) | 6.0(0.0) | 22.0(0.0) | 35.2(4.15) | 8.0(0.0) | 15.2(3.63) | 26.0(0.0) | 10.4(3.29) | 34.0(0.0) | 0.0(0.0) | 28.0(0.0) | 12.0(0.0) |
| waveform | 26.4(1.52) | 9.0(0.0) | 7.0(0.0) | **29.6(2.19)** | 8.0(0.0) | 10.2(2.17) | 27.0(0.0) | 7.4(2.79) | **28.0(0.0)** | 9.0(0.0) | 13.0(0.0) | 9.0(0.0) |
| wbc | 80.68(9.87) | 82.4(5.65) | 82.4(5.65) | 6.29(14.06) | 80.41(7.38) | 88.25(2.41) | 86.41(3.23) | 68.8(16.05) | 20.27(9.64) | 79.13(14.06) | **89.84(2.99)** | 87.28(5.1) |
| wdbc | 69.65(7.72) | 79.55(3.91) | 51.13(9.94) | **87.11(5.67)** | 67.95(6.63) | 70.91(11.05) | 78.7(2.32) | 52.65(20.79) | **85.63(6.59)** | 58.73(7.26) | 80.34(2.47) | 78.78(1.28) |
| wilt | 1.09(0.43) | 1.56(0.0) | 4.28(0.0) | 19.14(13.28) | 0.0(0.0) | 2.02(0.33) | 2.33(0.0) | 0.86(0.58) | 16.73(0.0) | 7.78(0.0) | 1.17(0.0) | 1.56(0.0) |
| wine | 75.73(7.02) | 56.13(3.76) | 39.27(8.98) | 82.7(5.26) | 77.67(8.64) | 71.05(4.25) | 87.16(5.61) | 55.33(13.28) | 80.84(3.22) | 74.67(15.13) | 78.28(3.76) | 66.01(5.57) |
| wpbc | 44.45(2.96) | 33.52(3.96) | 36.21(1.91) | 38.86(5.39) | 44.59(3.59) | 36.63(1.93) | 49.09(2.1) | 37.28(3.79) | 41.26(4.76) | 41.28(3.93) | 35.79(1.33) | 33.62(3.01) |
| yeast | **51.36(0.11)** | 42.6(0.0) | 46.35(0.0) | 47.5(1.51) | 44.38(0.0) | 44.46(0.86) | 46.75(0.0) | 48.05(3.25) | 47.73(0.0) | 46.27(0.18) | 46.55(0.0) | 43.39(0.0) |
| yelp | 13.4(0.2) | 16.0(0.0) | 13.4(0.0) | **20.72(0.23)** | 16.2(0.0) | 15.8(0.62) | 18.8(0.0) | 13.8(2.17) | **20.6(0.0)** | 12.52(0.3) | 15.2(0.0) | 16.2(0.0) |
| MNIST-C | 42.94(0.16) | 0.0(0.0) | 0.0(0.0) | **53.21(0.6)** | 23.24(0.0) | 34.7(2.59) | 46.3(0.0) | 34.48(5.52) | **52.96(0.0)** | 25.69(4.92) | 42.32(0.0) | 41.11(0.0) |
| FashionMNIST | 57.14(0.32) | 0.0(0.0) | 0.0(0.0) | **63.78(0.56)** | 33.65(0.0) | 45.33(2.4) | 59.05(0.0) | 48.9(3.89) | **63.3(0.0)** | 37.35(5.77) | 56.44(0.0) | 55.62(0.0) |
| CIFAR10 | 23.48(0.53) | 9.7(0.0) | 10.19(0.0) | **27.22(0.71)** | 14.9(0.0) | 18.19(1.18) | 22.85(0.0) | 20.42(3.07) | **27.07(0.0)** | 17.91(2.06) | 22.81(0.0) | 22.62(0.0) |
| SVHN | 18.65(0.37) | 0.0(0.0) | 0.0(0.0) | **19.66(0.45)** | 13.1(0.0) | 16.45(1.06) | 18.95(0.0) | 15.3(2.34) | **19.23(0.0)** | 14.34(1.94) | 18.43(0.0) | 18.27(0.0) |
| MVTec-AD | 66.48(3.06) | 62.92(4.9) | 62.92(4.9) | 67.48(3.26) | 62.65(2.52) | 64.6(2.65) | 67.37(3.1) | 60.38(3.87) | 67.29(3.16) | 72.35(3.2) | 64.6(2.67) | 63.41(2.78) |
| 20news | 12.83(2.11) | 10.89(0.77) | 10.76(1.05) | **16.64(1.39)** | 9.38(0.78) | 10.49(1.66) | 14.61(1.48) | 9.95(1.97) | **16.77(1.37)** | 14.72(1.95) | 11.31(1.1) | 10.5(1.24) |
| agnews | 15.35(0.25) | 11.55(0.0) | 11.45(0.0) | 30.53(0.37) | 11.45(0.0) | 12.48(0.65) | 20.1(0.0) | 13.52(1.55) | **30.6(0.0)** | 13.08(0.32) | 13.7(0.0) | 12.25(0.0) |

| | DAGMM | DeepSVDD | DROCC | GOAD | ICL | PlanarFlow | VAE | GANomaly | SLAD | DIF | DDPM | DTE-NP | DTE-IG | DTE-C |
|---|---|---|---|---|---|---|---|---|---|---|---|---|---|---|
| aloi | 7.29(0.0) | 7.63(0.0) | 5.98(1.67) | 5.17(0.92) | 0.0(0.0) | 5.73(1.45) | 4.91(0.56) | 3.83(0.72) | 7.63(0.0) | **9.35(2.27)** | 5.32(0.11) | 3.91(0.0) | 6.76(0.19) | 5.82(0.07) |
| 5.12(0.68) | | | | | | | | | | | | | | |
| amazon | 9.48(1.23) | 11.76(2.19) | 0.0(0.0) | 11.56(0.26) | 9.4(0.32) | 9.48(1.62) | 11.0(0.0) | 8.96(0.74) | 10.16(0.22) | 9.32(0.66) | 11.08(0.11) | 10.8(0.0) | 10.48(2.37) | 11.8(1.53) |
| annthyroid | 45.66(16.44) | 23.33(5.12) | 57.42(2.53) | 55.77(4.62) | 49.44(3.88) | 60.0(7.76) | 50.19(0.0) | 34.42(7.48) | 65.99(0.62) | 58.99(0.0) | 57.23(2.96) | 61.84(1.59) | 48.8(7.04) | **77.72(0.5)** |
| backdoor | 5.25(3.94) | 82.96(3.28) | 85.44(1.14) | 4.79(3.47) | **87.15(1.1)** | 36.73(22.17) | 8.5(1.27) | 21.92(20.41) | 0.0(0.0) | 20.28(2.02) | 9.6(0.62) | 51.48(17.26) | 84.45(2.16) | 82.58(2.44) |
| breastw | 83.52(11.1) | 91.85(0.77) | 48.27(26.55) | 95.66(0.34) | 95.91(0.68) | 94.09(1.69) | 96.12(0.47) | 90.05(3.37) | **96.87(0.65)** | 56.81(4.58) | 95.04(0.75) | **96.66(0.71)** | 74.03(10.32) | 88.18(2.86) |
| campaign | 34.13(3.43) | 37.89(12.9) | 0.0(0.0) | 22.62(9.05) | 51.03(0.73) | 42.11(2.88) | 48.84(0.0) | 40.92(4.2) | 49.83(0.0) | 27.11(0.0) | 50.4(0.68) | 50.98(0.61) | 47.85(2.0) | **52.12(0.62)** |
| cardio | 53.07(6.55) | 38.41(4.19) | 46.93(23.41) | 74.89(0.93) | 52.16(5.49) | 59.77(1.94) | **76.14(0.0)** | 58.75(3.47) | 60.8(0.0) | 27.27(0.0) | 61.7(1.73) | 63.07(0.0) | 36.82(14.29) | 58.3(0.76) |
| cardiotocography | 52.32(9.99) | 37.08(5.46) | 33.95(12.37) | 59.96(1.08) | 38.93(2.89) | 49.4(6.53) | 61.59(0.0) | 46.35(9.46) | 33.82(0.47) | 31.33(0.0) | 38.84(2.75) | 46.78(1.44) | 31.67(2.63) | 38.37(2.23) |
| celeba | 14.19(5.61) | 8.43(5.5) | 8.62(0.84) | 3.98(2.98) | 12.69(1.83) | 17.91(7.49) | **27.04(0.41)** | 11.07(7.99) | 13.69(1.66) | 10.79(1.42) | 26.02(2.89) | 15.81(0.69) | 19.11(5.61) | 17.35(3.48) |
| census | 14.46(2.49) | 19.28(1.44) | 15.55(1.44) | 4.96(2.13) | 23.96(0.54) | 13.86(2.4) | 20.76(0.27) | 18.27(3.91) | 8.65(11.85) | 14.44(1.82) | 20.27(0.48) | 22.21(0.6) | 17.47(2.81) | 17.43(2.38) |
| cover | 12.16(11.56) | 3.43(3.44) | 41.87(7.55) | 0.0(0.0) | 39.96(12.71) | 2.59(2.48) | 16.21(1.68) | 25.69(34.37) | 9.14(8.13) | 1.19(0.85) | 76.86(1.33) | 66.84(7.4) | 77.79(3.91) | 71.04(9.46) |
| donors | 21.94(14.54) | 41.44(30.4) | 29.37(27.54) | 4.29(4.13) | **97.22(1.04)** | 47.84(14.58) | 37.75(0.93) | 18.98(16.47) | 55.86(8.74) | 37.4(5.83) | 25.01(7.13) | 92.9(2.83) | 93.11(3.02) | 82.17(2.54) |
| fault | 53.22(4.51) | 54.92(1.38) | 56.7(4.72) | 55.96(0.68) | 57.59(0.56) | 57.65(4.55) | 55.22(0.08) | 56.76(4.09) | 60.06(0.24) | **61.96(0.0)** | 58.45(1.15) | 56.2(0.4) | 55.81(0.56) | 56.23(1.73) |
| fraud | 20.88(22.32) | 58.11(14.77) | 0.0(0.0) | 37.28(25.8) | 57.43(5.97) | 66.55(9.18) | 34.46(3.22) | 61.17(11.07) | 47.39(4.57) | 4.59(3.73) | **73.16(2.29)** | 48.36(5.41) | 55.61(10.96) | 68.23(13.11) |
| glass | 13.72(16.1) | 45.39(21.87) | 15.48(13.14) | 20.15(11.0) | **87.83(9.1)** | 19.56(11.2) | 18.03(5.11) | 24.83(10.99) | 35.03(7.61) | 60.41(14.55) | 32.81(13.36) | 24.58(16.83) | 78.4(13.6) | 37.46(6.39) |
| hepatitis | 47.51(7.54) | 93.82(1.29) | 29.29(17.28) | 57.86(7.77) | **99.64(0.79)** | 79.75(3.37) | 60.51(6.9) | 65.51(10.66) | **99.64(0.79)** | 81.1(4.43) | 86.69(4.26) | 79.03(11.95) | **99.64(0.79)** | 92.8(3.32) |
| http | 48.95(34.37) | 25.0(22.46) | 0.0(0.0) | 56.39(17.61) | 60.7(53.56) | 14.43(12.31) | 91.94(1.25) | 18.99(41.3) | 88.49(9.31) | 14.18(12.07) | 99.68(0.3) | 97.43(3.53) | 78.82(42.94) | 24.59(15.3) |
| imdb | 8.92(1.37) | 9.96(2.71) | 4.4(6.03) | 5.64(0.65) | 10.56(0.54) | 9.24(1.45) | 5.6(0.0) | 6.96(0.55) | 10.24(0.22) | **11.44(0.77)** | 5.56(0.09) | 5.2(0.0) | 10.36(5.13) | 7.24(1.62) |
| internetads | 31.85(5.57) | 54.29(5.92) | 38.42(6.06) | 46.14(0.52) | 55.87(0.91) | 55.82(1.78) | 45.65(0.0) | 52.72(0.54) | 57.83(0.3) | 31.14(0.15) | 45.87(0.35) | 53.21(3.77) | 54.95(5.79) | **64.78(2.48)** |
| ionosphere | 69.33(4.23) | **93.07(1.23)** | 60.19(21.57) | 83.42(5.88) | **94.18(1.62)** | 90.83(1.6) | 79.8(2.26) | 86.17(2.5) | **92.65(1.26)** | 85.86(1.47) | 88.64(1.58) | 91.51(2.09) | 89.68(4.23) | 89.58(0.9) |
| landsat | 40.95(4.11) | 42.18(2.53) | 40.8(3.61) | 32.96(1.19) | **53.82(0.35)** | 35.77(1.67) | 38.77(4.59) | 35.12(11.16) | 46.93(0.04) | 34.43(0.0) | 40.23(1.02) | 51.22(2.55) | 30.26(4.37) | 38.29(2.94) |
| letter | 8.6(4.39) | 5.0(1.58) | 13.6(8.62) | 1.2(0.45) | 7.2(0.84) | 3.8(2.39) | 1.0(0.0) | 1.2(0.84) | 1.6(0.55) | **28.0(0.0)** | 3.6(0.89) | 1.0(0.0) | 3.4(0.55) | 2.4(1.67) |
| lymphography | 67.58(11.63) | 89.82(9.49) | 26.15(35.82) | 93.13(5.46) | **100.0(0.0)** | 91.07(9.32) | 92.96(5.55) | 83.31(12.43) | 99.47(1.18) | 94.4(7.77) | 95.19(6.67) | 95.78(5.81) | 97.89(4.71) | 82.01(3.83) |
| magic.gamma | 57.35(2.82) | 59.88(0.89) | 72.61(0.67) | 62.72(1.48) | 69.55(0.47) | 67.82(1.62) | 65.25(0.0) | 56.95(1.27) | 65.95(0.02) | 62.02(0.0) | 78.88(0.95) | 76.46(0.81) | 79.31(1.11) | **80.78(0.8)** |
| mammography | 26.85(20.2) | 31.62(10.21) | 32.69(2.68) | 35.62(5.7) | 17.38(3.62) | 22.0(10.88) | 45.0(0.0) | 36.85(22.78) | 22.15(1.26) | 16.85(0.42) | 24.62(4.04) | 42.38(1.03) | 35.31(7.05) | 37.31(3.91) |
| mnist | 44.66(8.53) | 43.31(11.21) | 57.29(2.09) | 63.91(1.13) | 64.89(1.95) | 52.14(4.56) | 63.86(0.0) | 41.83(5.0) | 67.0(0.39) | 21.29(0.0) | 60.37(3.89) | **72.6(1.21)** | 50.43(8.41) | 58.46(2.88) |
| musk | 70.72(21.53) | 99.18(1.13) | 12.16(17.44) | **100.0(0.0)** | 83.3(8.97) | 35.05(28.17) | **100.0(0.0)** | **100.0(0.0)** | **100.0(0.0)** | 70.1(0.0) | **100.0(0.0)** | **100.0(0.0)** | 88.66(25.36) | **100.0(0.0)** |
| optdigits | 0.27(0.6) | 0.0(0.0) | 20.13(7.2) | 0.4(0.37) | **57.73(8.41)** | 0.0(0.0) | 0.67(0.0) | 4.93(4.07) | 39.87(0.73) | 0.67(0.0) | 28.4(7.05) | 27.2(10.73) | 27.07(18.57) | 10.93(3.39) |
| pageblocks | 57.92(11.77) | 54.71(3.06) | **68.43(1.73)** | 50.24(1.07) | 64.9(1.29) | 54.35(2.08) | 46.86(0.0) | 39.8(16.8) | 60.2(0.0) | 61.96(0.0) | 50.31(0.73) | 59.29(0.26) | 54.59(2.54) | 62.12(0.47) |
| pendigits | 13.97(16.56) | 12.31(12.2) | 19.23(5.21) | 41.54(3.49) | 61.15(5.82) | 14.23(6.24) | 44.23(0.0) | 15.51(20.48) | 44.36(1.05) | 26.92(0.0) | 64.62(2.62) | 83.46(6.02) | 59.74(9.95) | 56.03(8.28) |
| pima | 54.05(5.38) | 55.95(2.36) | 50.01(13.57) | 59.19(11.77) | **73.54(2.63)** | 67.92(2.87) | 70.45(2.76) | 58.93(5.42) | 58.87(2.87) | 54.47(4.52) | 66.62(2.63) | **74.69(2.25)** | 65.17(3.7) | 65.25(3.25) |
| satellite | 65.13(1.63) | 67.76(2.88) | 67.52(3.93) | 63.58(0.48) | 74.99(0.46) | 66.01(1.44) | 66.17(0.11) | 69.98(0.81) | **78.24(0.13)** | 63.85(0.0) | 73.74(0.27) | 71.91(0.99) | 70.57(3.05) | 72.33(0.63) |
| satimage-2 | 50.42(33.88) | 73.24(6.68) | 76.34(13.45) | 90.7(0.77) | 88.45(1.54) | 62.25(5.49) | 88.17(0.77) | 76.62(15.83) | 88.73(0.0) | 0.0(0.0) | 78.59(5.12) | 90.14(0.0) | 78.31(5.23) | 66.48(2.71) |
| shuttle | 67.9(24.09) | 98.11(0.09) | 0.0(0.0) | 56.26(30.2) | **98.82(0.16)** | 46.15(11.43) | 95.78(0.0) | 91.15(8.04) | 98.47(0.05) | 97.86(0.0) | 98.3(0.08) | 98.3(0.1) | **98.78(0.09)** | 97.99(0.02) |
| skin | 55.71(28.67) | 43.26(2.59) | 78.44(1.35) | 52.05(1.57) | 1.09(0.97) | 78.59(11.23) | 44.73(0.84) | 31.3(2.79) | 74.55(2.88) | 72.67(0.95) | 73.42(5.35) | 95.08(1.16) | 93.36(2.92) | 82.23(0.43) |
| smtp | 26.32(34.3) | 34.0(23.28) | 13.75(30.75) | 48.56(12.43) | 6.96(12.38) | 0.0(0.0) | **69.59(4.42)** | 0.0(0.0) | **69.59(4.42)** | **68.05(5.72)** | 56.19(11.77) | **69.59(4.42)** | 37.91(24.53) | **69.5(4.43)** |
| spambase | 68.45(3.55) | 69.55(3.79) | 73.94(3.09) | 78.81(0.63) | 79.27(0.49) | 77.62(3.62) | 78.49(0.03) | 79.33(1.09) | **81.5(0.13)** | 50.98(0.0) | 63.55(0.95) | 80.67(0.48) | 75.18(2.65) | 80.02(0.23) |
| speech | 3.28(1.16) | 1.31(1.37) | 2.95(2.14) | 2.95(1.37) | 2.62(1.87) | 1.64(1.16) | 3.28(0.0) | 1.97(1.37) | **6.23(2.69)** | 4.59(1.8) | 2.95(1.37) | **4.92(0.0)** | 1.97(1.8) | 3.93(1.47) |
| stamps | 47.02(25.21) | 37.07(9.74) | 28.03(26.42) | 52.72(15.8) | 77.17(7.83) | 51.82(17.69) | 61.44(6.81) | 32.27(14.5) | 50.99(12.77) | 43.72(9.88) | 62.98(12.09) | **85.93(2.97)** | 70.23(11.35) | 57.97(11.6) |
| thyroid | 65.38(11.86) | 65.59(8.6) | 69.03(3.68) | 74.19(1.32) | 56.13(8.72) | 69.89(3.72) | 74.19(0.0) | 52.9(20.51) | 71.18(1.18) | 51.61(0.0) | 75.48(2.07) | 74.84(0.96) | 47.74(10.66) | 75.48(0.9) |
| vertebral | 21.19(13.5) | 16.71(5.9) | 16.99(21.41) | 18.25(10.64) | **63.39(5.17)** | 21.53(8.46) | 14.07(1.08) | 19.76(4.38) | 14.2(10.67) | 36.37(4.21) | 37.54(12.52) | 21.56(14.78) | 46.58(10.24) | 42.13(11.49) |
| vowels | 5.6(6.69) | 20.8(4.15) | 13.6(12.2) | 23.6(2.19) | 24.4(8.29) | 14.4(5.18) | 12.0(0.0) | 22.8(15.4) | 38.8(4.38) | **50.0(0.0)** | 41.6(5.37) | 29.6(0.89) | 36.0(2.45) | 37.2(4.15) |
| waveform | 4.6(1.67) | 14.6(3.58) | 26.6(6.11) | 9.8(2.39) | 26.8(5.81) | **28.6(4.39)** | 8.0(0.0) | 12.0(7.21) | 2.2(1.1) | 5.0(0.0) | 12.0(1.22) | 26.0(0.0) | 24.8(4.27) | 12.2(2.77) |
| wbc | 46.16(26.61) | 54.17(11.28) | 26.61(27.57) | 86.45(4.97) | **92.88(4.57)** | 55.68(15.83) | **88.42(3.21)** | 64.11(13.61) | **92.2(4.48)** | 71.77(11.9) | 86.03(5.16) | **89.35(3.45)** | 62.64(10.31) | 32.49(10.28) |
| wdbc | 32.5(29.05) | **83.34(7.3)** | 8.7(19.44) | 75.82(5.69) | **90.52(9.27)** | 75.08(11.73) | 78.69(6.98) | 57.89(10.72) | **85.23(6.41)** | 3.3(5.65) | 79.28(7.36) | **85.05(6.83)** | **89.47(7.99)** | 68.08(14.3) |
| wilt | 5.68(3.33) | 0.62(0.35) | 1.48(1.39) | 12.45(2.0) | 35.18(2.59) | 3.27(4.59) | 1.95(0.0) | 6.15(5.11) | 7.0(0.0) | 10.51(0.0) | 20.23(0.91) | 2.41(1.49) | **64.75(4.54)** | 17.59(2.54) |
| wine | 48.54(37.49) | 69.81(9.29) | 12.78(16.95) | 65.52(5.99) | 99.32(1.52) | 68.31(14.39) | 67.9(6.31) | 42.72(36.22) | **100.0(0.0)** | 48.58(10.34) | 90.31(3.17) | 92.13(11.85) | 99.32(1.52) | 98.29(2.42) |
| wpbc | 33.18(3.98) | 70.22(6.11) | 31.9(5.22) | 34.22(3.42) | **90.55(0.97)** | 42.84(4.39) | 36.48(3.41) | 45.61(5.66) | 87.91(3.64) | 67.99(4.49) | 50.71(4.98) | 68.16(14.02) | 59.51(9.36) | 57.76(6.44) |
| yeast | **51.99(3.47)** | 49.47(5.66) | 48.8(3.77) | **53.21(2.43)** | 50.26(1.38) | 47.77(2.46) | 44.26(0.18) | 49.51(4.78) | 49.27(0.22) | 43.79(0.0) | 50.85(2.02) | 46.15(0.44) | 49.66(2.28) | 49.23(1.12) |
| yelp | 8.2(2.18) | 10.4(1.77) | 6.88(6.41) | 15.36(0.38) | 8.88(0.73) | 11.12(1.38) | 16.2(0.0) | 13.36(1.31) | 7.12(0.11) | 8.16(0.33) | 16.08(0.18) | 19.6(0.0) | 14.6(2.17) | 14.72(1.51) |
| MNIST-C | 25.17(8.13) | 34.45(3.0) | 30.54(9.37) | 42.11(0.45) | 52.07(1.25) | 34.67(1.69) | 41.11(0.01) | 44.5(1.76) | 47.66(0.22) | 12.0(1.59) | 42.44(0.25) | 47.51(0.0) | 46.21(5.11) | 48.68(1.5) |
| FashionMNIST | 33.43(6.11) | 47.96(2.33) | 32.21(12.83) | 56.25(0.6) | 62.9(1.07) | 48.05(1.55) | 55.62(0.0) | 59.45(1.3) | 59.77(0.26) | 18.37(2.0) | 56.67(0.27) | 59.68(0.0) | 54.66(2.68) | 59.22(0.99) |
| CIFAR10 | 13.35(2.59) | 16.46(1.96) | 14.97(2.42) | 22.88(0.89) | 20.59(1.37) | 17.6(1.31) | 22.62(0.0) | 24.27(1.43) | 24.67(0.48) | 10.33(1.42) | 22.98(0.41) | 23.19(0.0) | 19.97(2.49) | 23.79(1.09) |
| SVHN | 12.99(2.1) | 15.2(1.66) | 13.18(1.68) | 18.49(0.42) | **19.55(0.99)** | 17.53(0.97) | 18.27(0.0) | 19.19(0.89) | 19.18(0.24) | 11.37(0.78) | 18.73(0.23) | 19.21(0.0) | 17.59(1.78) | **19.49(0.61)** |
| MVTec-AD | 51.88(5.78) | 76.78(3.79) | 51.15(10.03) | 63.83(3.2) | **82.63(2.26)** | 60.38(3.19) | 63.11(2.64) | 67.28(2.76) | **81.77(2.71)** | 66.34(4.21) | 65.02(2.84) | 75.69(2.88) | 75.87(4.33) | 78.98(3.08) |
| 20news | 9.61(2.52) | 13.64(3.63) | 12.55(3.73) | 10.7(1.03) | **16.43(1.74)** | 9.08(3.33) | 10.79(1.83) | 14.54(2.93) | 14.09(1.43) | 10.12(2.67) | 10.55(1.27) | **17.83(1.58)** | 14.33(4.65) | **18.95(4.69)** |
| agnews | 10.77(3.43) | 10.66(3.57) | 3.72(4.16) | 13.15(0.69) | 17.79(0.72) | 9.78(1.2) | 12.25(0.0) | 14.44(0.8) | 13.33(0.14) | 10.71(1.01) | 12.52(0.18) | 20.7(0.0) | 14.11(3.85) | 23.93(3.44) |

Table 15: Average AUC PR and standard deviations over five seeds for the semi-supervised setting on ADBench.

| | CBLOF | COPOD | ECOD | FeatureBagging | HBOS | IForest | kNN | LODA | LOF | MCD | OCSVM | PCA | DAGMM | DeepSVDD | DROCC | GOAD | ICL | PlanarFlow | VAE | GANomaly | SLAD | DIF | DDPM | DTE-NP | DTE-IG | DTE-C |
|---|---|---|---|---|---|---|---|---|---|---|---|---|---|---|---|---|---|---|---|---|---|---|---|---|---|---|
| aloi | | | | | | | | | | | | | | | | | | | | | | | | | | |
| amazon | | | | | | | | | | | | | | | | | | | | | | | | | | |
| annthyroid | | | | | | | | | | | | | | | | | | | | | | | | | | |
| backdoor | | | | | | | | | | | | | | | | | | | | | | | | | | |
| breastw | | | | | | | | | | | | | | | | | | | | | | | | | | |
| campaign | | | | | | | | | | | | | | | | | | | | | | | | | | |
| cardio | | | | | | | | | | | | | | | | | | | | | | | | | | |
| cardiotocography | | | | | | | | | | | | | | | | | | | | | | | | | | |
| celeba | | | | | | | | | | | | | | | | | | | | | | | | | | |
| census | | | | | | | | | | | | | | | | | | | | | | | | | | |
| cover | | | | | | | | | | | | | | | | | | | | | | | | | | |
| donors | | | | | | | | | | | | | | | | | | | | | | | | | | |
| fault | | | | | | | | | | | | | | | | | | | | | | | | | | |
| fraud | | | | | | | | | | | | | | | | | | | | | | | | | | |
| glass | | | | | | | | | | | | | | | | | | | | | | | | | | |
| hepatitis | | | | | | | | | | | | | | | | | | | | | | | | | | |
| http | | | | | | | | | | | | | | | | | | | | | | | | | | |
| imdb | | | | | | | | | | | | | | | | | | | | | | | | | | |
| internetads | | | | | | | | | | | | | | | | | | | | | | | | | | |
| ionosphere | | | | | | | | | | | | | | | | | | | | | | | | | | |
| landsat | | | | | | | | | | | | | | | | | | | | | | | | | | |
| letter | | | | | | | | | | | | | | | | | | | | | | | | | | |
| lymphography | | | | | | | | | | | | | | | | | | | | | | | | | | |
| magic.gamma | | | | | | | | | | | | | | | | | | | | | | | | | | |
| mammography | | | | | | | | | | | | | | | | | | | | | | | | | | |
| mnist | | | | | | | | | | | | | | | | | | | | | | | | | | |
| musk | | | | | | | | | | | | | | | | | | | | | | | | | | |
| optdigits | | | | | | | | | | | | | | | | | | | | | | | | | | |
| pageblocks | | | | | | | | | | | | | | | | | | | | | | | | | | |
| pendigits | | | | | | | | | | | | | | | | | | | | | | | | | | |
| pima | | | | | | | | | | | | | | | | | | | | | | | | | | |
| satellite | | | | | | | | | | | | | | | | | | | | | | | | | | |
| satimage-2 | | | | | | | | | | | | | | | | | | | | | | | | | | |
| shuttle | | | | | | | | | | | | | | | | | | | | | | | | | | |
| skin | | | | | | | | | | | | | | | | | | | | | | | | | | |
| smtp | | | | | | | | | | | | | | | | | | | | | | | | | | |
| spambase | | | | | | | | | | | | | | | | | | | | | | | | | | |
| speech | | | | | | | | | | | | | | | | | | | | | | | | | | |
| stamps | | | | | | | | | | | | | | | | | | | | | | | | | | |
| thyroid | | | | | | | | | | | | | | | | | | | | | | | | | | |
| vertebral | | | | | | | | | | | | | | | | | | | | | | | | | | |
| vowels | | | | | | | | | | | | | | | | | | | | | | | | | | |
| waveform | | | | | | | | | | | | | | | | | | | | | | | | | | |
| wbc | | | | | | | | | | | | | | | | | | | | | | | | | | |
| wilt | | | | | | | | | | | | | | | | | | | | | | | | | | |
| wine | | | | | | | | | | | | | | | | | | | | | | | | | | |
| wpbc | | | | | | | | | | | | | | | | | | | | | | | | | | |
| yeast | | | | | | | | | | | | | | | | | | | | | | | | | | |
| MNIST-C | | | | | | | | | | | | | | | | | | | | | | | | | | |
| FashionMNIST | | | | | | | | | | | | | | | | | | | | | | | | | | |
| CIFAR10 | | | | | | | | | | | | | | | | | | | | | | | | | | |
| SVHN | | | | | | | | | | | | | | | | | | | | | | | | | | |
| MVTec-AD | | | | | | | | | | | | | | | | | | | | | | | | | | |
| 20news | | | | | | | | | | | | | | | | | | | | | | | | | | |
| agnews | | | | | | | | | | | | | | | | | | | | | | | | | | |

Table 16: Average AUC ROC and standard deviations over five seeds for the unsupervised setting on ADBench.

Table 17: Average F1 score and standard deviations over five seeds for the unsupervised setting on ADBench.

Table 18: Average AUC PR and standard deviations over five seeds for the unsupervised setting on ADBench.

| | CBLOF | COPOD | ECOD | FeatureBagging | HBOS | IForest | KNN | LODA | LOF | MCD | OCSVM | PCA | DAGMM | DeepSVDD | DROCC | GOAD | ICL | PlanarFlow | VAE | GANomaly | SLAD | DIF | DDPM | DTE-NP | DTE-IG | DTE-C |
|---|---|---|---|---|---|---|---|---|---|---|---|---|---|---|---|---|---|---|---|---|---|---|---|---|---|---|
| aloi | 3.74(0.07) | 3.13(0.0) | 3.29(0.0) | 10.36(0.45) | 3.38(0.05) | 3.39(0.03) | 4.76(0.02) | 3.27(0.29) | 9.69(0.28) | 3.22(0.05) | 3.92(0.14) | 3.72(0.03) | 3.31(0.26) | 3.44(0.27) | 3.04(0.0) | 3.28(0.23) | 4.6(0.43) | 3.23(0.1) | 3.7(0.0) | 4.38(0.29) | 3.65(0.1) | 3.06(0.15) | 3.59(0.02) | 5.55(0.02) | 3.95(0.1) | 3.28(0.07) |
| amazon | 6.06(0.06) | 5.96(0.01) | 5.5(0.01) | 5.8(0.11) | 5.87(0.02) | 5.83(0.09) | 6.22(0.01) | 5.44(0.43) | 5.79(0.13) | 6.21(0.04) | 5.89(0.01) | 5.69(0.02) | 4.94(0.25) | 4.6(0.31) | 5.0(0.0) | 5.83(0.2) | 5.23(0.16) | 5.04(0.2) | 5.69(0.0) | 5.61(0.11) | 5.08(0.04) | 5.21(0.18) | 5.71(0.01) | 6.22(0.08) | 5.49(0.4) | 5.72(0.54) |
| annthyroid | 16.94(0.78) | 17.43(0.1) | 27.21(0.44) | 21.68(6.66) | 22.79(0.86) | 11.23(3.56) | 22.41(0.47) | 10.08(7.77) | 25.8(0.11) | 6.21(0.0) | 18.75(0.28) | 19.55(1.07) | 24.99(7.1) | 4.6(0.0) | 18.55(2.55) | 13.12(3.87) | 5.23(0.16) | 33.61(8.07) | 9.16(0.01) | 12.44(0.05) | 5.08(0.04) | 5.21(0.0) | 29.74(23.6) | 22.82(0.34) | 43.84(3.25) | 48.07(1.28) |
| backdoor | 54.65(1.42) | 2.48(0.05) | 2.48(0.05) | 21.66(6.86) | 5.15(0.09) | 4.54(0.72) | 47.92(1.45) | 9.82(7.1) | 35.82(2.43) | 12.15(6.59) | 53.14(1.28) | 53.41(3.0) | 24.99(7.1) | 37.23(5.52) | 2.48(0.05) | 34.69(3.95) | 71.71(1.3) | 33.61(8.07) | 52.57(1.21) | 54.05(6.18) | 2.48(0.05) | 2.5(0.1) | 53.68(12.06) | 47.29(1.45) | 38.03(6.2) | 67.01(0.84) |
| breastw | 88.99(6.33) | 98.87(0.33) | 98.24(0.39) | 28.44(1.29) | 95.64(1.41) | 96.26(0.95) | 93.21(0.85) | 95.29(3.23) | 29.65(2.09) | 96.23(1.15) | 83.69(1.55) | 94.55(0.9) | 4.94(0.25) | 48.29(3.06) | 77.57(4.69) | 82.59(7.94) | 63.45(5.52) | 90.76(0.05) | 89.46(8.33) | 89.78(0.83) | 5.08(0.04) | 34.72(0.0) | 99.68(0.28) | 92.09(0.62) | 77.03(2.76) | 71.52(3.8) |
| campaign | 28.38(0.93) | 36.64(0.84) | 36.46(0.87) | 18.44(1.29) | 35.26(0.47) | 21.78(3.16) | 13.05(0.53) | 13.06(5.47) | 12.88(5.47) | 39.51(0.52) | 89.69(1.55) | 28.38(0.0) | 6.84(0.42) | 16.89(3.63) | 32.34(1.59) | 25.61(0.61) | 26.92(1.96) | 10.41(3.57) | 28.41(0.0) | 10.65(0.86) | 2.48(0.05) | 34.12(0.0) | 53.68(5.28) | 92.02(0.74) | 33.46(0.47) | 32.42(1.11) |
| cardio | 48.23(1.68) | 57.59(0.51) | 56.68(0.74) | 16.09(1.04) | 45.80(0.86) | 55.88(4.43) | 40.17(1.51) | 42.78(0.47) | 15.89(1.81) | 36.44(5.19) | 33.57(0.67) | 60.87(0.73) | 19.28(7.44) | 17.72(6.67) | 25.78(2.78) | 53.96(5.29) | 10.84(1.25) | 47.07(11.95) | 61.0(0.12) | 33.44(20.15) | 18.46(2.5) | 9.57(0.68) | 27.84(5.61) | 37.62(0.74) | 18.35(6.15) | 26.81(0.87) |
| cardiotocography | 33.53(5.06) | 40.29(2.63) | 50.23(0.37) | 27.64(0.53) | 36.1(0.67) | 43.62(2.11) | 32.37(0.29) | 41.63(0.41) | 18.78(0.89) | 40.26(7.31) | 40.83(0.26) | 36.28(2.59) | 27.14(4.98) | 17.72(0.85) | 25.78(2.59) | 40.26(7.31) | 27.15(0.91) | 47.52(0.07) | 47.52(0.07) | 39.15(0.74) | 17.24(0.0) | 34.72(0.0) | 33.84(3.2) | 31.16(0.49) | 23.68(3.68) | 27.55(1.23) |
| celeba | 6.88(2.06) | 9.28(0.59) | 9.5(0.55) | 2.37(0.28) | 8.95(0.56) | 6.26(0.41) | 6.07(0.27) | 4.65(3.19) | 1.81(0.02) | 9.17(1.68) | 10.28(0.48) | 60.87(0.73) | 4.42(1.27) | 3.11(1.51) | 4.66(0.22) | 2.09(0.91) | 4.48(0.3) | 6.55(3.45) | 11.2(0.73) | 2.9(2.18) | 3.18(0.27) | 2.25(0.1) | 9.25(1.47) | 5.19(0.23) | 5.77(1.59) | 7.68(0.83) |
| census | 8.75(0.28) | 6.23(0.16) | 6.23(0.16) | 6.11(0.18) | 7.36(0.09) | 7.31(0.49) | 7.31(0.49) | 6.52(2.72) | 6.87(0.23) | 7.17(0.08) | 8.53(0.23) | 8.66(0.23) | 6.17(0.3) | 7.54(1.22) | 4.83(6.28) | 5.8(0.42) | 6.17(0.3) | 7.36(0.54) | 7.54(1.22) | 8.58(1.32) | 8.23(1.24) | 6.2(0.5) | 8.56(0.23) | 8.56(0.23) | 8.3(0.3) | 8.09(0.3) |
| cover | 6.98(0.21) | 7.58(0.69) | 6.47(0.07) | 11.27(0.07) | 7.36(0.19) | 31.76(0.49) | 7.31(0.49) | 8.5(0.12) | 9.17(0.18) | 1.98(0.1) | 9.21(0.0) | 6.84(0.0) | 6.17(0.3) | 4.31(0.22) | 4.83(6.28) | 3.81(4.35) | 2.25(0.45) | 0.38(0.26) | 7.41(0.02) | 7.41(0.02) | 2.1(0.0) | 4.9(0.01) | 4.5(0.03) | 4.78(0.18) | 2.49(0.09) | 6.59(0.78) |
| donors | 14.77(0.42) | 20.94(0.53) | 26.47(0.61) | 12.04(0.75) | 13.47(1.11) | 12.44(0.93) | 18.21(0.17) | 25.47(3.62) | 10.86(0.23) | 14.13(4.81) | 13.94(0.3) | 16.61(0.62) | 8.58(3.98) | 11.24(7.73) | 12.28(3.73) | 9.85(0.88) | 11.87(1.73) | 24.07(2.38) | 16.48(0.33) | 12.27(8.05) | 8.09(0.53) | 5.91(0.16) | 14.33(0.84) | 18.83(0.17) | 16.35(6.18) | 13.95(3.79) |
| fault | 47.3(3.1) | 31.26(0.16) | 32.54(0.17) | 39.57(1.23) | 35.97(6.44) | 39.45(0.61) | 52.21(0.73) | 33.65(2.46) | 10.86(0.23) | 33.35(0.73) | 40.08(0.34) | 33.16(0.61) | 36.11(4.21) | 49.62(2.49) | 37.52(3.2) | 37.77(1.09) | 47.27(1.26) | 44.27(1.85) | 33.98(0.07) | 34.86(3.3) | 3.56(0.07) | 35.41(0.41) | 39.2(0.85) | 53.23(0.39) | 41.7(3.59) | 42.18(2.19) |
| fraud | 14.53(3.21) | 25.17(5.67) | 21.54(4.92) | 12.34(0.75) | 20.88(5.54) | 14.49(5.34) | 16.88(5.49) | 33.68(2.23) | 0.26(0.05) | 48.76(3.53) | 33.16(0.61) | 14.91(3.13) | 8.44(1.19) | 25.02(7.18) | 0.16(0.01) | 25.72(8.34) | 12.66(2.66) | 44.74(9.78) | 15.67(4.11) | 15.62(7.69) | 17.96(6.53) | 0.18(0.03) | 14.58(3.63) | 13.68(4.1) | 18.81(8.21) | 64.75(5.97) |
| glass | 14.36(3.18) | 11.05(2.58) | 18.33(6.17) | 32.89(0.84) | 16.07(4.53) | 21.78(4.38) | 16.74(2.54) | 33.63(3.19) | 14.42(6.7) | 33.35(1.52) | 12.98(4.31) | 11.33(2.11) | 1.06(0.5) | 10.06(3.39) | 37.52(3.2) | 12.58(0.18) | 12.23(5.53) | 11.33(0.89) | 9.99(2.56) | 14.85(5.17) | 13.37(2.61) | 4.61(1.41) | 7.29(1.12) | 20.57(6.59) | 11.64(2.19) | 16.82(4.63) |
| hepatitis | 30.16(3.57) | 28.02(4.37) | 14.47(0.73) | 53.93(3.4) | 32.16(3.17) | 32.12(2.46) | 39.12(2.64) | 21.47(3.72) | 21.39(2.64) | 36.14(4.8) | 27.13(2.3) | 49.99(2.39) | 36.82(2.99) | 16.99(1.58) | 31.09(6.15) | 31.09(6.15) | 23.56(2.34) | 13.37(0.09) | 31.38(2.73) | 30.36(7.55) | 38.15(3.94) | 0.40(0.04) | 64.22(20.75) | 57.57(2.24) | 21.49(8.14) | 25.73(4.5) |
| http | 46.43(3.33) | 4.96(0.03) | 4.87(0.03) | 4.69(1.83) | 30.19(3.04) | 88.63(8.26) | 0.98(0.69) | 0.41(0.07) | 4.95(2.1) | 86.46(1.79) | 35.59(2.55) | 49.99(2.39) | 4.86(0.1) | 9.34(9.72) | 0.37(0.03) | 44.13(3.94) | 9.08(7.28) | 36.27(2.48) | 47.66(2.57) | 21.15(4.39) | 38.15(3.94) | 0.40(0.04) | 64.22(20.75) | 2.41(0.99) | 29.53(19.62) | 44.03(16.11) |
| imdb | 4.74(0.03) | 4.74(0.0) | 4.48(0.0) | 4.87(0.04) | 4.74(0.01) | 4.68(0.04) | 4.67(0.02) | 4.58(0.06) | 4.88(0.06) | 4.38(0.08) | 4.69(0.09) | 4.86(0.1) | 4.86(0.1) | 5.31(0.07) | 5.01(0.03) | 4.68(0.05) | 5.28(0.09) | 4.59(0.0) | 4.59(0.0) | 4.79(0.22) | 5.19(0.04) | 5.12(0.15) | 4.59(0.01) | 4.69(0.03) | 4.74(0.52) | 4.67(0.32) |
| internetads | 29.65(0.08) | 50.47(0.2) | 50.54(0.2) | 18.19(1.9) | 52.27(0.3) | 48.62(4.28) | 29.64(0.09) | 24.18(1.77) | 23.2(1.2) | 34.36(5.4) | 29.09(0.14) | 27.56(0.76) | 20.73(1.63) | 28.78(0.13) | 19.73(1.54) | 28.78(0.13) | 23.66(0.9) | 26.19(0.35) | 29.56(0.0) | 34.52(1.55) | 26.26(1.06) | 18.6(0.83) | 29.46(0.05) | 29.01(0.46) | 27.53(2.43) | 30.18(2.54) |
| ionosphere | 88.1(2.92) | 66.28(3.2) | 63.34(2.3) | 82.05(3.0) | 35.26(2.03) | 77.92(3.9) | 91.09(0.79) | 71.94(0.7) | 80.67(4.01) | 94.66(0.24) | 82.91(0.84) | 72.08(2.42) | 47.34(2.6) | 39.24(6.87) | 72.27(10.53) | 72.02(3.9) | 47.19(2.26) | 82.36(0.99) | 73.01(1.74) | 86.59(2.41) | 80.73(2.48) | 36.53(1.6) | 63.39(3.49) | 92.04(1.9) | 27.53(2.43) | 87.96(2.24) |
| landsat | 21.23(0.78) | 17.66(0.05) | 16.57(0.05) | 24.65(0.49) | 23.07(0.23) | 19.93(0.66) | 25.75(0.23) | 19.37(0.66) | 24.99(0.55) | 25.31(0.07) | 17.58(0.05) | 16.33(0.13) | 23.11(1.27) | 36.15(4.04) | 27.23(0.36) | 18.54(1.66) | 47.19(0.83) | 18.65(0.43) | 18.65(0.43) | 22.07(2.21) | 30.85(3.31) | 20.92(0.14) | 19.99(0.13) | 25.45(0.27) | 22.27(3.23) | 22.34(0.89) |
| letter | 16.64(1.01) | 6.84(0.03) | 7.1(0.05) | 44.53(3.22) | 7.79(0.21) | 8.59(0.16) | 20.31(0.72) | 8.26(1.17) | 43.32(2.85) | 17.38(0.35) | 11.27(0.27) | 7.62(0.12) | 8.27(2.66) | 9.94(1.37) | 25.22(4.92) | 9.85(0.8) | 20.8(3.38) | 15.27(4.23) | 7.11(0.01) | 14.64(7.07) | 30.96(3.04) | 34.8(0.49) | 36.69(1.64) | 25.51(1.41) | 18.09(2.29) | 25.65(0.6) |
| lymphography | 91.49(6.57) | 90.69(2.49) | 89.39(2.04) | 9.0(7.08) | 91.91(3.02) | 77.22(5.71) | 89.44(6.58) | 49.0(9.58.67) | 13.52(9.57) | 76.67(5.33) | 29.09(0.14) | 93.51(4.78) | 45.41(15.81) | 26.44(9.97) | 46.31(19.05) | 89.72(6.59) | 23.69(0.93) | 69.26(4.35) | 93.67(4.31) | 86.59(2.41) | 66.11(8.36) | 18.6(0.83) | 93.71(20.13) | 80.51(0.21) | 38.81(9.67) | 38.14(14.88) |
| magic.gamma | 66.61(0.05) | 58.8(0.04) | 53.34(0.05) | 53.87(0.79) | 61.74(0.15) | 63.77(0.37) | 72.35(0.14) | 57.87(1.31) | 51.98(0.44) | 63.15(0.09) | 62.51(0.11) | 58.88(0.09) | 45.01(3.53) | 49.93(1.07) | 62.71(0.72) | 32.59(4.63) | 54.76(1.7) | 69.24(4.35) | 59.12(0.0) | 34.52(1.55) | 50.99(1.35) | 35.4(0.33) | 65.14(1.91) | 72.98(0.13) | 65.74(2.99) | 66.4(0.97) |
| mammography | 13.95(2.79) | 43.02(0.41) | 43.54(0.39) | 7.01(0.99) | 13.24(1.15) | 21.76(4.58) | 18.06(0.92) | 21.76(4.58) | 8.48(0.72) | 3.58(0.25) | 18.65(0.74) | 20.44(1.39) | 11.06(0.12) | 26.44(9.97) | 72.27(10.33) | 46.31(0.67) | 47.19(2.28) | 33.19(0.14) | 19.82(0.0) | 86.59(2.41) | 50.99(1.35) | 2.44(0.28) | 9.83(0.76) | 17.45(0.99) | 8.2(2.53) | 17.02(1.42) |
| mnist | 38.6(0.17) | 9.23(0.0) | 9.21(0.0) | 24.11(1.05) | 10.90(0.12) | 29.03(4.81) | 48.06(0.52) | 23.54(4.31) | 8.48(0.72) | 25.31(0.07) | 38.54(0.33) | 38.84(0.94) | 21.85(3.55) | 35.84(0.0) | 23.19(1.08) | 35.76(4.34) | 23.19(1.08) | 25.94(0.68) | 38.1(0.0) | 26.03(4.5) | 9.21(0.0) | 37.88(1.09) | 33.78(1.09) | 39.99(0.75) | 42.15(3.73) | 36.76(2.05) |
| musk | 100.0(0.08) | 36.91(4.05) | 47.47(1.53) | 13.95(7.85) | 99.87(0.08) | 94.47(9.05) | 70.81(0.35) | 84.13(7.56) | 11.77(5.21) | 99.15(1.13) | 100.0(0.0) | 99.95(0.02) | 50.02(29.29) | 10.74(13.43) | 19.57(7.39) | 99.97(0.07) | 12.84(4.41) | 39.11(37.87) | 99.98(0.0) | 100.0(0.0) | 26.35(7.41) | 39.44(0.77) | 98.38(1.16) | 43.36(3.14) | 13.68(4.74) | 55.3(2.58) |
| optdigits | 5.92(0.26) | 2.88(0.0) | 2.88(0.0) | 3.62(0.78) | 19.18(1.06) | 4.61(0.81) | 2.18(0.09) | 2.05(0.34) | 3.54(0.69) | 2.24(0.21) | 2.65(0.08) | 2.7(0.03) | 2.61(1.34) | 3.89(2.58) | 3.16(0.28) | 3.94(0.72) | 2.0(0.01) | 1.75(0.2) | 2.68(0.0) | 2.67(1.13) | 3.04(0.24) | 3.03(0.22) | 2.82(0.35) | 2.14(0.09) | 2.82(0.35) | 55.49(4.05) |
| pageblocks | 54.67(5.46) | 37.03(0.41) | 51.96(0.39) | 34.11(2.79) | 31.84(4.58) | 46.37(1.42) | 55.58(0.65) | 40.05(8.6) | 29.16(2.08) | 61.69(0.68) | 53.07(0.57) | 52.46(1.66) | 25.52(5.14) | 28.49(3.65) | 24.09(8.58) | 37.29(4.72) | 28.49(3.65) | 53.76(4.73) | 51.32(0.01) | 32.59(7.06) | 40.38(3.46) | 49.27(2.07) | 66.16(0.76) | 52.96(0.97) | 50.72(9.01) | 50.72(9.01) |
| pendigits | 19.17(0.42) | 17.71(1.05) | 26.6(0.57) | 34.1(2.79) | 24.17(0.83) | 26.01(4.72) | 9.58(2.01) | 18.06(8.48) | 3.81(0.79) | 6.91(0.19) | 22.57(1.06) | 21.86(0.8) | 5.63(4.69) | 7.5(1.6) | 1.27(0.46) | 37.29(4.72) | 28.49(3.65) | 64.02(2.57) | 19.62(13.14) | 41.26(3.36) | 4.52(1.09) | 8.75(1.47) | 47.6(7.13) | 18.86(3.3) | 4.46(0.91) | 4.36(1.06) |
| pima | 48.38(0.73) | 53.62(2.38) | 48.38(2.46) | 41.22(2.23) | 50.12(2.65) | 56.32(3.34) | 52.99(3.69) | 49.19(4.08) | 48.39(5.5) | 53.19(5.6) | 47.54(2.19) | 35.57(3.87) | 17.66(6.12) | 47.64(7.25) | 21.05(3.05) | 37.18(2.12) | 9.08(7.28) | 59.55(0.78) | 31.84(2.33) | 41.28(3.53) | 45.02(0.09) | 37.05(7.9) | 46.16(0.76) | 52.83(2.87) | 43.74(3.29) | 44.68(2.5) |
| satellite | 65.64(6.27) | 57.04(0.08) | 52.62(0.1) | 37.77(0.72) | 68.78(0.47) | 64.88(1.51) | 58.16(0.35) | 61.27(4.3) | 38.1(0.7) | 76.80(1.13) | 65.44(0.16) | 60.61(0.17) | 32.67(25.13) | 40.57(4.8) | 46.45(0.98) | 65.83(2.86) | 45.14(1.92) | 59.55(0.78) | 70.55(0.27) | 41.26(3.93) | 52.89(1.37) | 31.60(0.48) | 66.16(0.76) | 50.44(3.18) | 37.96(2.66) | 52.91(3.46) |
| satimage-2 | 97.21(0.03) | 79.7(0.94) | 66.62(1.58) | 4.23(2.71) | 76.01(4.1) | 91.75(0.85) | 13.28(0.76) | 85.74(7.48) | 4.08(2.5) | 68.24(3.21) | 96.53(0.02) | 87.19(0.1) | 28.92(20.61) | 5.15(4.69) | 11.75(3.86) | 11.54(1.69) | 10.18(2.82) | 48.36(8.72) | 81.24(2.27) | 61.22(32.46) | 34.44(6.41) | 1.32(0.11) | 57.86(0.25) | 10.93(1.18) | 37.4(0.96) | 13.84(3.38) |
| shuttle | 18.38(2.14) | 88.12(2.34) | 90.5(0.14) | 8.08(1.88) | 96.47(0.15) | 97.62(0.41) | 19.31(0.46) | 16.83(19.06) | 19.31(0.46) | 84.1(0.05) | 90.72(0.06) | 91.33(0.15) | 13.58(3.41) | 14.86(7.6) | 7.15(0.0) | 13.58(9.72) | 13.48(3.81) | 34.57(2.26) | 9.54(0.0) | 90.07(8.72) | 63.27(30.21) | 7.22(0.07) | 77.88(7.67) | 72.17(0.58) | 24.72(13.77) | 62.6(0.55) |
| skin | 28.86(3.15) | 17.86(0.69) | 18.27(0.1) | 9.87(0.0) | 23.52(0.49) | 23.56(0.41) | 29.0(0.18) | 29.07(0.16) | 23.01(0.17) | 49.01(0.81) | 22.01(0.0) | 17.32(4.0) | 22.56(6.4) | 4.64(0.0) | 28.42(1.09) | 35.76(4.34) | 22.14(2.08) | 33.53(2.26) | 19.33(0.1) | 18.17(1.34) | 33.54(3.14) | 5.70(2.22) | 17.54(0.55) | 18.65(0.26) | 31.19(2.34) | 30.24(1.74) |
| smtp | 40.32(5.33) | 3.70(0.01) | 41.7(0.0) | 21.66(12.23) | 5.05(2.2) | 44.0(2.5) | 4.92(0.07) | 31.21(0.04) | 0.53(0.08) | 35.76(4.34) | 38.25(8.36) | 3.22(0.0) | 24.01(4.1) | 48.84(1.46) | 0.43(0.51) | 35.76(4.34) | 0.4(0.75) | 41.07(5.45) | 18.17(1.34) | 10.36(0.49) | 42.54(4.67) | 5.70(2.22) | 50.23(9.75) | 41.07(5.45) | 1.16(2.2) | 42.15(3.73) |
| spambase | 18.7(0.02) | 54.37(0.16) | 51.82(0.17) | 34.39(0.06) | 51.77(1.22) | 48.75(1.64) | 41.54(0.17) | 38.65(5.96) | 35.95(0.33) | 34.89(2.17) | 40.21(0.07) | 40.93(0.51) | 38.87(3.27) | 45.64(4.64) | 38.32(0.09) | 38.73(3.86) | 37.01(0.74) | 43.32(5.7) | 40.95(0.02) | 42.94(1.74) | 38.51(0.29) | 39.46(0.77) | 38.73(0.71) | 40.69(0.22) | 39.86(3.19) | 40.04(1.52) |
| speech | 1.87(0.02) | 1.88(0.07) | 1.96(0.01) | 2.18(0.15) | 2.29(0.14) | 2.05(0.34) | 1.85(0.02) | 1.6(0.2) | 1.91(0.11) | 1.9(0.11) | 1.85(0.03) | 1.84(0.0) | 2.18(0.43) | 1.8(0.15) | 2.03(0.73) | 1.93(0.42) | 1.66(2.25) | 1.75(0.2) | 1.84(0.0) | 2.22(0.74) | 1.72(0.1) | 1.74(0.27) | 2.04(0.43) | 1.88(0.15) | 1.9(0.21) | 2.0(0.33) |
| stamps | 21.06(2.78) | 39.78(4.75) | 32.35(3.22) | 14.26(4.1) | 33.18(3.9) | 34.72(4.5) | 31.69(3.92) | 25.73(6.26) | 15.27(4.4) | 25.73(6.26) | 31.76(4.47) | 36.46(4.3) | 19.78(8.33) | 23.64(0.84) | 24.09(8.58) | 28.5(0.19) | 11.66(2.25) | 28.36(3.04) | 23.65(3.65) | 27.83(4.72) | 16.53(5.78) | 12.64(1.4) | 27.25(4.34) | 21.88(11.01) | 23.48(11.01) | 22.61(4.68) |
| thyroid | 27.17(0.59) | 17.84(0.9) | 47.18(0.25) | 6.92(8.88) | 50.12(2.95) | 56.22(3.46) | 39.12(2.16) | 18.92(2.65) | 1.82(4.6) | 50.98(5.97) | 32.89(2.07) | 30.17(0.2) | 21.44(2.8) | 21.44(2.8) | 33.44(1.18) | 31.78(3.97) | 21.92(3.39) | 23.97(1.24) | 21.23(1.24) | 23.97(1.24) | 32.74(3.42) | 21.46(4.43) | 17.55(0.58) | 27.25(4.34) | 14.05(3.08) | 14.05(3.08) |
| vertebral | 12.34(0.98) | 8.5(1.2) | 4.13(0.0) | 12.37(3.08) | 9.12(1.05) | 9.68(1.0) | 9.51(1.18) | 13.5(1.1) | 13.2(5.67) | 16.64(4.96) | 17.93(0.17) | 16.97(0.5) | 7.29(0.01) | 9.78(0.58) | 9.63(0.92) | 9.22(4.15) | 11.54(1.69) | 15.43(1.59) | 17.21(0.01) | 17.07(3.82) | 15.81(1.16) | 17.24(0.46) | 17.75(0.19) | 29.45(0.06) | 13.31(5.54) | 11.92(1.7) |
| vowels | 16.01(1.03) | 3.43(0.05) | 8.28(0.54) | 31.42(8.14) | 7.83(0.89) | 16.23(6.18) | 44.32(0.55) | 12.73(3.85) | 58.16(0.75) | 8.54(6.45) | 19.58(1.16) | 6.87(0.26) | 4.07(1.98) | 3.71(0.39) | 29.53(0.31) | 15.42(7.74) | 21.92(3.39) | 29.53(8.97) | 6.96(0.08) | 5.56(3.1) | 32.74(3.42) | 5.14(0.33) | 5.78(0.23) | 50.44(3.18) | 16.57(4.43) | 41.71(2.32) |
| waveform | 12.23(1.76) | 5.69(0.14) | 4.04(0.03) | 10.97(0.25) | 4.83(0.11) | 5.63(0.92) | 13.28(0.76) | 3.51(0.46) | 4.0(2.5) | 3.95(0.12) | 5.25(0.11) | 4.41(0.02) | 3.06(0.51) | 6.11(0.96) | 4.46(0.0) | 4.24(0.82) | 21.06(4.43) | 43.06(2.69) | 4.46(0.01) | 5.78(4.83) | 2.38(0.04) | 5.14(0.05) | 5.78(9.23) | 3.74(0.96) | 3.73(0.96) | 4.28(0.59) |
| wbc | 69.07(11.79) | 88.13(2.34) | 88.19(2.42) | 3.72(0.48) | 72.26(6.35) | 58.04(2.62) | 74.27(6.66) | 74.22(7.61) | 89.79(0.28) | 83.92(1.36) | 81.72(1.15) | 91.33(4.96) | 32.67(25.13) | 4.46(0.0) | 3.94(2.57) | 7.63(6.15) | 21.06(4.43) | 43.06(2.69) | 89.23(4.82) | 70.18(14.01) | 70.59(20.16) | 4.87(1.2) | 5.78(9.23) | 72.17(13.46) | 34.84(17.79) | 19.35(3.2) |
| wben | 28.86(3.15) | 88.13(2.34) | 49.27(0.01) | 8.19(2.44) | 72.82(0.35) | 76.21(2.2) | 76.21(6.92) | 72.69(4.12) | 3.51(0.49) | 3.3(1.34) | 84.17(1.0) | 91.34(0.96) | 94.91(0.33) | 88.16(1.66) | 3.39(2.57) | 6.47(1.51) | 21.06(4.43) | 43.06(2.69) | 89.23(4.82) | 46.95(0.38) | 36.36(6.0) | 48.71(2.3) | 48.27(10.6) | 26.52(7.83) | 23.58(2.07) | 15.67(1.38) |
| wilt | 4.01(0.12) | 3.7(0.01) | 4.17(0.0) | 5.05(2.16) | 3.94(0.15) | 4.4(0.25) | 4.92(0.07) | 3.66(0.48) | 3.54(0.01) | 83.31(0.34) | 3.54(0.01) | 3.22(0.01) | 4.73(0.02) | 4.64(0.17) | 4.05(0.02) | 5.36(4.08) | 10.88(1.46) | 5.05(2.16) | 3.64(0.0) | 4.19(0.08) | 10.36(0.49) | 5.70(2.22) | 7.62(0.85) | 5.35(0.07) | 21.14(6.69) | 16.29(1.47) |
| wine | 17.04(2.72) | 36.39(6.24) | 19.45(3.2) | 6.06(0.5) | 3.94(0.15) | 20.69(4.89) | 8.05(0.89) | 24.99(9.9) | 6.42(1.66) | 73.74(14.77) | 13.48(2.11) | 26.39(5.02) | 12.04(7.37) | 11.65(1.15) | 12.64(7.4) | 22.94(4.04) | 8.67(1.18) | 8.64(0.64) | 23.65(3.65) | 17.63(13.88) | 6.77(1.17) | 8.18(0.0) | 7.45(2.14) | 7.37(1.21) | 6.39(1.93) | 10.27(3.41) |
| wpbc | 22.74(2.24) | 23.37(1.06) | 21.66(1.22) | 20.57(1.44) | 24.11(1.66) | 23.73(1.92) | 23.44(1.4) | 22.65(1.74) | 20.98(1.73) | 75.66(2.17) | 22.15(1.31) | 22.86(1.58) | 21.44(2.55) | 24.02(1.97) | 23.38(2.93) | 21.44(2.8) | 24.02(1.97) | 23.97(1.24) | 21.23(1.24) | 23.97(1.24) | 23.55(0.58) | 23.21(1.36) | 22.36(3.11) | 22.72(1.72) | 23.58(2.07) | 23.14(3.11) |
| yeast | 31.39(0.56) | 30.79(0.15) | 33.19(0.08) | 32.59(0.99) | 32.79(0.5) | 30.39(0.49) | 29.36(0.48) | 33.01(2.79) | 31.51(0.78) | 29.76(0.48) | 30.33(0.38) | 30.17(0.2) | 35.27(2.8) | 35.05(2.67) | 28.51(1.8) | 33.22(2.65) | 31.84(1.18) | 15.43(1.98) | 32.28(0.01) | 33.59(3.64) | 34.04(1.07) | 34.48(1.53) | 32.04(0.49) | 29.45(0.66) | 30.64(1.82) | 30.61(1.7) |
| yelp | 17.26(2.17) | 4.99(0.0) | 5.0(0.0) | 12.8(0.38) | 4.83(0.11) | 5.18(0.49) | 19.06(0.11) | 35.8(0.37) | 12.65(0.33) | 24.55(0.82) | 4.9(0.24) | 16.97(0.5) | 3.8(0.24) | 18.09(3.66) | 9.63(0.92) | 6.78(0.07) | 15.81(1.16) | 29.73(1.53) | 17.21(0.01) | 6.95(0.02) | 7.9(0.14) | 17.24(0.46) | 17.75(0.19) | 19.22(0.15) | 14.05(3.08) | 5.0(0.97) |
| MNIST-C | 32.89(1.4) | 6.45(0.02) | 6.74(0.03) | 12.8(0.38) | 19.4(0.3) | 5.63(0.03) | 10.23(0.03) | 18.04(4.4) | 18.8(0.76) | 6.28(0.63.4) | 10.19(0.24) | 31.88(0.91) | 13.82(4.52) | 18.09(3.66) | 9.78(0.58) | 32.76(0.93) | 15.81(1.16) | 29.73(1.53) | 32.28(0.01) | 30.68(0.35) | 32.37(3.59) | 5.14(0.33) | 17.75(0.66) | 19.22(0.15) | 21.25(4.22) | 26.74(1.66) |
| CIFAR10 | 10.34(0.16) | 5.0(0.0) | 5.0(0.0) | 4.83(0.37) | 4.83(0.11) | 4.99(0.03) | 5.0(0.03) | 8.57(1.48) | 8.57(1.48) | 8.41(0.69) | 10.19(0.24) | 5.0(0.01) | 6.17(0.82) | 7.29(0.98) | 5.43(0.04) | 4.2(0.43) | 6.84(0.03) | 6.98(0.02) | 7.21(0.02) | 10.48(0.55) | 10.38(0.27) | 5.24(0.31) | 5.46(2.22) | 7.70(0.66) | 7.77(0.06) | 9.17(0.5) |
| FashionMNIST | 68.01(0.3) | 5.0(0.0) | 5.0(0.0) | 6.35(0.03) | 7.49(0.07) | 8.9(0.51) | 7.94(0.01) | 6.4(0.88) | 8.26(0.14) | 6.28(0.63) | 7.81(0.16) | 7.38(0.24) | 6.17(0.82) | 6.25(0.54) | 6.3(0.86) | 5.8(0.26) | 6.84(0.03) | 6.98(0.02) | 7.96(0.72) | 7.96(0.02) | 7.91(0.14) | 5.0(0.2) | 5.0(0.2) | 8.01(0.04) | 6.89(0.62) | 7.73(0.34) |
| SVHN | 10.67(0.17) | 5.0(0.0) | 5.0(0.0) | 5.4(1.24) | 5.4(1.24) | 8.9(0.51) | 10.23(0.03) | 6.31(0.85) | 6.3(0.86) | 8.4(0.05.36) | 6.38(0.14) | 6.24(0.23) | 6.18(0.62) | 4.64(0.04) | 5.55(0.74) | 6.28(0.3) | 6.84(0.03) | 6.98(0.02) | 10.07(0.0) | 6.98(0.02) | 10.38(0.27) | 5.77(1.05) | 6.25(0.21) | 10.36(0.09) | 6.89(0.62) | 15.67(1.38) |
| MVTec-AD | 73.8(0.1) | 6.09(0.29) | 5.03(0.01) | 5.14(3.52) | 5.43(0.17) | 7.94(0.02) | 6.90(0.36) | 6.42(0.03) | 6.34(0.06.14) | 6.29(0.03.4) | 7.81(0.16) | 6.9(0.27) | 5.9(0.57) | 5.27(0.72) | 6.3(0.86) | 6.3(0.86) | 6.89(0.24) | 5.0(0.2) | 6.98(0.02) | 6.55(0.07) | 7.9(0.14) | 5.77(0.11) | 6.25(0.21) | 7.16(0.49) | 6.01(1.38) | 7.73(0.34) |
| 20news | 6.66(0.42) | 6.69(0.29) | 6.17(0.12) | 5.2(0.32) | 6.05(0.21) | 6.24(0.36) | 6.90(0.36) | 6.24(0.07) | 6.23(0.14) | 7.15(0.62) | 6.38(0.44) | 6.24(0.23) | 6.28(0.3) | 6.26(0.33) | 6.28(0.3) | 6.28(0.3) | 6.89(0.24) | 5.63(0.69) | 6.27(0.3) | 6.55(0.71) | 6.3(0.08) | 5.77(0.11) | 6.25(0.21) | 7.16(0.49) | 6.01(1.38) | 6.84(0.87) |
| agnews | 7.24(0.07) | 5.85(0.01) | 5.76(0.0) | 12.51(0.62) | 5.87(0.0) | 6.36(0.22) | 8.16(0.03) | 6.42(0.35) | 12.47(0.61) | 7.7(0.1) | 6.78(0.07) | 6.11(0.03) | 5.31(0.69) | 5.27(0.72) | 5.1(0.21) | 6.53(0.49) | 6.89(0.24) | 5.0(0.2) | 6.11(0.0) | 8.81(0.79) | 6.3(0.08) | 6.17(0.01) | 6.17(0.01) | 8.45(0.06) | 6.28(1.28) | 7.55(1.04) |

