# OpenReview forum: "On Diffusion Modeling for Anomaly Detection"
_ICLR.cc/2024/Conference — ICLR 2024 spotlight_

### Official Review · Reviewer_udy2 · 2023-10-31

**Soundness:** 3 good
**Presentation:** 3 good
**Contribution:** 2 fair
**Rating:** 6
**Confidence:** 5

**Summary:**

In this paper, the authors investigate diffusion modeling for anomaly detection and introduce an efficient approach called Diffusion Time Estimation (DTE), showing its competitive performance and improved inference times compared to traditional methods and deep learning techniques.

**Strengths:**

1. This paper is well-organized and easy to follow.
2. Investigating the diffusion model to facilitate anomaly detection is an interesting and promising research issue.
3. The authors conduct comprehensive experiments to demonstrate the effectiveness of the proposed method.

**Weaknesses:**

1. The contribution of this paper appears limited. There are several existing works on anomaly detection utilizing diffusion models, such as those mentioned below:
[1] Unsupervised Surface Anomaly Detection with Diffusion Probabilistic Model, ICCV23.
[2] Feature Prediction Diffusion Model for Video Anomaly Detection, ICCV23.
[3] DiffusionAD: Denoising Diffusion for Anomaly Detection, Arxiv.
[4] Diffusion models for medical anomaly detection, MCCAI22.
Can the authors point out the unique contributions of this paper compared with them?
2. The motivation and rationale for introducing the diffusion model into anomaly detection are somewhat unclear. The authors should emphasize it.
3. Authors should compare with some latest AD methods proposed in 2023. Besides, it would be more convincing to include diffusion model-based anomaly detection approaches.
4. The reviewer has thoroughly reviewed the experimental results presented in the appendix. The authors have done a comprehensive experiment by including the results from dozens of datasets in this paper. However, the reviewer observed that the proposed method did not consistently outperform other methods on each individual dataset. It achieved the best performance only on a subset of the datasets.

**Questions:**

See weakness.

---

> ### Author Response · Authors · 2023-11-20
>
> We thank the reviewer for their careful and thorough review.
>
> > The contribution of this paper appears limited. There are several existing works on anomaly detection utilizing diffusion models, such as those mentioned below: [1] Unsupervised Surface Anomaly Detection with Diffusion Probabilistic Model, ICCV23. [2] Feature Prediction Diffusion Model for Video Anomaly Detection, ICCV23. [3] DiffusionAD: Denoising Diffusion for Anomaly Detection, Arxiv. [4] Diffusion models for medical anomaly detection, MCCAI22. Can the authors point out the unique contributions of this paper compared with them?
>
> The main difference between DTE and other applications of diffusion model to anomaly detection, including [1-4] is that DTE does not learn a generative model. We note in section 3 that “we propose a much simpler approach that does not require modeling the reverse diffusion process but instead models the distribution over diffusion time corresponding to noisy input samples.” As a result, not only this is orders of magnitude faster than learning a generative diffusion model (Figure 1 and 11), but it also performs better (Figure 5). Another distinguishing feature is that we are focused on finding anomalous samples rather than identifying localized anomalies within images.
> We would also like to clarify that papers [1,2] are published after the May 28, 2023, which is ICLR cut-off deadline for citation, and papers [3,4] are already cited and discussed in Section 5. Nevertheless, we have added a discussion of [1] to section 5 and cited both [1,2].
>
> > The motivation and rationale for introducing the diffusion model into anomaly detection are somewhat unclear. The authors should emphasize it.
>
> Historically, all generative modelling techniques are used for anomaly detection since they can implicitly identify low-probability samples. Diffusion models are not an exception. As we explain in Section 3, “The reverse diffusion process implicitly learns the score function of the data distribution and can be used for the likelihood-based identification of anomalies.” We then further investigate this application of diffusion modelling for anomaly detection and find an alternative use of diffusion, explained in the previous answer, leading us to DTE.
>
> > Authors should compare with some latest AD methods proposed in 2023. Besides, it would be more convincing to include diffusion model-based anomaly detection approaches.
>
> ADBench was used as our benchmark and was published at NeurIPS 2022. As discussed in Section 5, We include all baselines from that benchmark in addition to ICL (ICLR 2022), DROCC (ICML 2020), GOAD (ICLR 2020). We updated the paper using two new 2023 baselines that were published after the cut-off deadline: SLAD [1] and DIF [2]. DTE outperforms both methods overall on ADBench. We found that methods created specifically for image anomaly detection tend to perform poorly on tabular datasets, and most of the recent papers are tailored for images. As discussed in the paper, the latest anomaly detection methods, such as [3] and [4], use variations on vanilla DDPM, where the difference between the original input and its reconstruction is used. We do compare to this baseline and find DTE superior to both in efficiency and accuracy.
>
> [1] Hongzuo Xu, Yijie Wang, Juhui Wei, Songlei Jian, Yizhou Li, and Ning Liu. Fascinating supervisory
> signals and where to find them: Deep anomaly detection with scale learning. In Proceedings of
> the 40th International Conference on Machine Learning, ICML’23
>
> [2] Hongzuo Xu, Guansong Pang, Yijie Wang, and Yongjun Wang. Deep isolation forest for anomaly
> detection. IEEE Transactions on Knowledge and Data Engineering, pp. 1–14, 2023a.
>
> [3] Julia Wolleb, Florentin Bieder, Robin Sandk ̈uhler, and Philippe C Cattin. Diffusion models for
> medical anomaly detection. In Medical Image Computing and Computer Assisted Intervention–
> MICCAI 2022: 25th International Conference, Proceedings,
> Part VIII, pp. 35–45. Springer, 2022.
>
> [4] Julian Wyatt, Adam Leach, Sebastian M. Schmon, and Chris G. Willcocks. Anoddpm: Anomaly
> detection with denoising diffusion probabilistic models using simplex noise. In 2022 IEEE/CVF
> Conference on Computer Vision and Pattern Recognition Workshops (CVPRW), pp. 649–655,
> 2022.
>
> > The reviewer has thoroughly reviewed the experimental results presented in the appendix. The authors have done a comprehensive experiment by including the results from dozens of datasets in this paper. However, the reviewer observed that the proposed method did not consistently outperform other methods on each individual dataset. It achieved the best performance only on a subset of the datasets.
>
> Due to dataset diversity in ADBench, it is indeed challenging for a single method to consistently outperform others across all datasets. Our method shows competitive performance and, in many cases, outperforms other methods on an aggregate score across various datasets while remaining the most efficient in inference time.

---

> > ### Comment · Reviewer_udy2 · 2023-11-22
> > **Reply to authors.**
> >
> > Thanks for the authors’ reply.  The authors have clarified the contribution and motivation. Although the performance improvement is still the reviewer’s concern, it has shown effectiveness in many datasets and inference time. Besides, after carefully checking for the responses of other reviewers. I think the authors have addressed most of their concerns. Therefore, I want to increase my rating.

---

### Official Review · Reviewer_vggP · 2023-10-31

**Soundness:** 3 good
**Presentation:** 3 good
**Contribution:** 3 good
**Rating:** 8
**Confidence:** 4

**Summary:**

The paper proposes a non-parametric approach based on the inverse Gamma distribution of diffusion time for noisy input, achieving accurate predictions and ranking anomalies similarly to kNN. Additionally, a parametric strategy employs a deep neural network for large datasets, demonstrating competitive performance and significantly improving inference time. Pre-trained embeddings for images are found to enhance diffusion-based methods, highlighting the potential advantage of using latent space diffusion. The evaluation on ADBench, a benchmark for anomaly detection datasets, shows promising results in comparison to prior work.

**Strengths:**

+ The proposed approach offers a simpler alternative that avoids modeling the reverse diffusion process. Instead, it focuses on modeling the distribution over diffusion time associated with noisy input samples. The assumption is that anomalies are distant from the data manifold, leading to higher density for larger timesteps in the distribution.
+ Both non-parametric and parametric strategies are employed for DTE based anomaly detection, and the parametric strategies achieve a tradeoff between accuracy and inference time.
+ The evaluation is conducted on ADBench, as well as additional image datasets such as Visa, CIFAR-10, and MNIST.

**Weaknesses:**

-The performance in the semi-supervised setting is more competitive compared to the unsupervised setting. This indicates that DTE benefits from labeled data, allowing for a more accurate modeling of the distribution of diffusion time.

**Questions:**

How does varying the ratio of labeled data in the semi-supervised setting affect the performance of DTE? Can this method be extend to more challenging tasks, such as the localization of anomaly in data?

---

> ### Author Response · Authors · 2023-11-20
>
> We thank the reviewer for their feedback.
>
> > The performance in the semi-supervised setting is more competitive compared to the unsupervised setting. This indicates that DTE benefits from labeled data, allowing for a more accurate modeling of the distribution of diffusion time.
>
> > How does varying the ratio of labeled data in the semi-supervised setting affect the performance of DTE? Can this method be extend to more challenging tasks, such as the localization of anomaly in data?
>
> Since the question and the point raised under weakness are related, we answer both at the same time: As mentioned in Section 2, the semi-supervised setting used in the paper is also known as one-class classification, where the model only has access to normal samples when training and no labeled data. The ratio is the number of normal samples used for training on the normal samples used for testing. Increasing the ratio simply increases the training data, which should increase the model performance, but can make the test less accurate. This setting is more competitive compared to the unsupervised setting since it does not have any anomalies in its training data, allowing the model to learn the distribution of the normal data more accurately.

---

### Official Review · Reviewer_ahnY · 2023-11-02

**Soundness:** 3 good
**Presentation:** 3 good
**Contribution:** 2 fair
**Rating:** 6
**Confidence:** 3

**Summary:**

The work explores the use of diffusion models for anomaly detection (AD), and proposes an AD method based on diffusion time estimation (DTE), with three models under the DTE framework introduced. The DTE models, particularly the DNN-based parametric model, can achieve desired detection performance while substantially reducing the inference time. The models are evaluated on 57 datasets and show comparable performance compared to a set of 19 baseline/SOTA methods in both semi-supervised and unsupervised settings.

**Strengths:**

- The work is well motivated and easy-to-follow.
- The idea of using diffusion time estimation (DTE) for AD is interesting and new. It also provides a way for learning parametric DTE models that allow efficient inference time.
- The proposed DTE models generally perform substantially better than the popular diffusion model DDPM, and show comparable performance to a large number of competing methods on 57 tabular datasets.

**Weaknesses:**

- The performance of the DTE models seems to be upper bounded by the simple kNN-based AD method. There are a number of kNN-based AD methods, including some deep methods like Refs [1]. It would be helpful for the empirical evidence support if these advanced kNN variants are included in the empirical comparison.
- The models rely on distance in original feature space, and they would fail to work if the data lies in very high-dimensional space, e.g., datasets with hundreds of thousands of features or millions of features.
- Since the method is based on generative models, it is important to discuss and compare with other generative model-based AD methods, such as GAN-based methods, to highlight the advantages of the proposed method.
- Since the models directly work on tabular datasets, it is misleading to claim that the evaluation is performed on diverse tabular, image, and natural language datasets.
- The work may be improved by having more discussion on recent diffusion model-based AD studies, such as [2-4].



**Refs**
- [1] Learning representations of ultrahigh-dimensional data for random distance-based outlier detection. In Proceedings of the 24th ACM SIGKDD international conference on knowledge discovery & data mining (pp. 2041-2050).
- [2] Unsupervised Surface Anomaly Detection with Diffusion Probabilistic Model. In Proceedings of the IEEE/CVF International Conference on Computer Vision (pp. 6782-6791).
- [3] Feature Prediction Diffusion Model for Video Anomaly Detection. In Proceedings of the IEEE/CVF International Conference on Computer Vision (pp. 5527-5537).
- [4] Multimodal Motion Conditioned Diffusion Model for Skeleton-based Video Anomaly Detection. In Proceedings of the IEEE/CVF International Conference on Computer Vision (pp. 10318-10329).

**Questions:**

Please see the weaknesses.

---

> ### Author Response · Authors · 2023-11-20
>
> We thank the reviewer for their detailed feedback and suggestions.
>
> > The performance of the DTE models seems to be upper bounded by the simple kNN-based AD method. There are a number of kNN-based AD methods, including some deep methods like Refs [1]. It would be helpful for the empirical evidence support if these advanced kNN variants are included in the empirical comparison.
>
> Thanks for the pointer; we have now cited this paper and briefly discussed it. We would like to point out that while kNN remains one of the top-performing methods, DTE (parametric) is not upper-bounded by kNN: DTE (categorical, inverse Gamma) outperforms kNN on several datasets, as shown in Tables 13-18. Moreover, as we show, while DTE (parametric) is comparable with kNN/DTE(non-parametric), it has a significant efficiency advantage (Figure 1 and 11).
>
> The referenced paper uses unsupervised embedding followed by kNN in the embedding space. Appendix D presents ablations that compare kNN/DTE (non-parametric) and DTE parametric on embeddings produced through self-supervised learning and pre-trained models. Indeed, many of the datasets in ADBench are embeddings produced by pre-trained models. We generally observe better performance for all methods when using high-quality embeddings.
>
> > The models rely on distance in original feature space, and they would fail to work if the data lies in very high-dimensional space, e.g., datasets with hundreds of thousands of features or millions of features.
>
> It is informative to consider this point for diffusion models used for generative modeling: there again, while using latent diffusion can improve results for high-dimensional inputs, diffusion modeling remains competitive in the observation space. Similarly, as also discussed in answer to previous question, using embedding improves the performance of our method. We would also like to note that distance to nearest neighbours (in case of DTE non-parametric) can still be reliable in high-dimensions due to manifold hypothesis (i.e., distance to closes point in high-dimension resembles the distance on data-manifold and therefore the distance in any isometric embedding). Overall, we do observe that for high-dimensional data, performance on learned embeddings is better; see Appendix D.
>
> > Since the method is based on generative models, it is important to discuss and compare with other generative model-based AD methods, such as GAN-based methods, to highlight the advantages of the proposed method.
>
> We agree with the reviewer. In addition to the DDPM and the normalizing flows baselines, we have added a comparison on both semi-supervised and unsupervised AD to a GAN-based method from ADBench and VAE-based method (implementation from PyOD). DTE outperforms all these methods in terms of effectiveness and efficiency. We’ve added a sentence identifying the AD baselines that rely on generative models, per the reviewer’s suggestion.
>
> > Since the models directly work on tabular datasets, it is misleading to claim that the evaluation is performed on diverse tabular, image, and natural language datasets.
>
> We have reworded this sentence, which now reads “tabular data and embeddings of images and natural language datasets.”
>
> > The work may be improved by having more discussion on recent diffusion model-based AD studies, such as [2-4].
>
> Thanks for the pointers; we have cited them and briefly discussed [1,2] in the updated paper. We would also like to note that two of these papers are on video anomaly detection, and all were published after May 28, 2023, which is the cut-off date according to ICLR guidelines.
>
> [1] Learning representations of ultrahigh-dimensional data for random distance-based outlier detection. In Proceedings of the 24th ACM SIGKDD international conference on knowledge discovery & data mining (pp. 2041-2050).
>
> [2] Unsupervised Surface Anomaly Detection with Diffusion Probabilistic Model. In Proceedings of the IEEE/CVF International Conference on Computer Vision (pp. 6782-6791).
>
> [3] Feature Prediction Diffusion Model for Video Anomaly Detection. In Proceedings of the IEEE/CVF International Conference on Computer Vision (pp. 5527-5537).
>
> [4] Multimodal Motion Conditioned Diffusion Model for Skeleton-based Video Anomaly Detection. In Proceedings of the IEEE/CVF International Conference on Computer Vision (pp. 10318-10329).

---

> > ### Comment · Reviewer_ahnY · 2023-11-23
> > **Follow-up comments**
> >
> > Thanks for the rebuttal. It helps address my concerns on comparison on other generative models, relevance to other DM-based AD studies, and partly the high-dimensional AD issues. My concern on comparison to more advanced kNN-based AD methods is not properly addressed though.
> >
> > Overall, I like the proposed idea in that it is intuitive and interesting, and generally effective on diverse datasets. Even though the method might not be the state-of-the-art on some cases, it provides some new insights into how DMs could be exploited for AD. I therefore retain my positive rating for this work.

---

### Official Review · Reviewer_BHmN · 2023-11-07

**Soundness:** 3 good
**Presentation:** 3 good
**Contribution:** 4 excellent
**Rating:** 8
**Confidence:** 4

**Summary:**

The manuscript proposes *diffusion time estimation* (DTE) for the task of unsupervised and semi-supervised point anomaly detection. DTE assumes a data point to be produced by diffusion process and estimates the distribution of the denoising time step required to reconstruct the data point. The mean of mode of the distribution is regarded as the anomaly score of the data point.

In addition to its effectiveness demonstrated in prior works, DTE avoids the actual denoising process redundant for anomaly detection and directly estimates the extend to which the sample appers to be anomalous. With this keen insight, the manuscript provides detailed derivation of the posterior distribution of variance of time (decided by time step) given an input image assumed to be produced by a diffusion process. Based on the derivation, the manuscript designs one non-parametric model and two parametric models (regressive and categorical respectively).

The performances of the three models are evaluated on 57 datasets from ADBench demonstrating the capabilities of DTE for the task of anomaly detection and its advantages over DDPM in quaility and efficacy.

**Strengths:**

* The paper provides a new perspective for adopting diffusion modeling in the field of anomaly detection
* The paper provides one parametric and two non-parametric practical models for the task of anomaly detection
* The methods proposed in the paper (DTE) achieve significate margins in both performance and efficacy compared with DDPM

**Weaknesses:**

* The advantage of DTE methods hold should be demonstrated quantitatively. As the DTE methods perform worse than kNN in both quality and efficiency, quantitative results are recommended to demonstrate the distinctions in scalability of DTE methods
* The presentation in Figure 3 needs optimization. To demonstrate the small difference between non-parametric estimate and analytical posterior, the visualization of residual part seems more straightforward.

**Questions:**

See *Weaknesses*.

---

> ### Author Response · Authors · 2023-11-20
>
> We thank the reviewer for their insightful comments.
>
> > The advantage of DTE methods hold should be demonstrated quantitatively. As the DTE methods perform worse than kNN in both quality and efficiency, quantitative results are recommended to demonstrate the distinctions in scalability of DTE methods
>
> This is a misunderstanding. As seen in figure 1, DTE (parametric) is significantly more efficient than KNN, which we show is similar to DTE (non-parametric). Exact computation times for training and inference are given in Appendix C.4. In terms of performance, the reviewer is correct that the average performance of DTE (parametric) and KNN is comparable. This is where we believe the efficiency of DTE can facilitate its applications to large datasets.
>
> > The presentation in Figure 3 needs optimization. To demonstrate the small difference between non-parametric estimate and analytical posterior, the visualization of residual part seems more straightforward.
>
> Thank you for the suggestion. It makes sense, and we will improve this in the next revision (we are considering several alternatives.)

---

### Author Response · Authors · 2023-11-20

We would like to thank all reviewers for their thorough reviews and valuable feedback on our paper. We appreciate the opportunity to address their concerns and clarify aspects of our work.

Some changes in the revision:
- Citation for [1-6] in related work
- Added VAE and GAN baselines in the results for comparison with other generative models
- Added SLAD [5] and DIF [6] baselines for comparison against more recent methods.

[1] Learning representations of ultrahigh-dimensional data for random distance-based outlier detection. In Proceedings of the 24th ACM SIGKDD international conference on knowledge discovery & data mining (pp. 2041-2050).

[2] Unsupervised Surface Anomaly Detection with Diffusion Probabilistic Model. In Proceedings of the IEEE/CVF International Conference on Computer Vision (pp. 6782-6791).

[3] Feature Prediction Diffusion Model for Video Anomaly Detection. In Proceedings of the IEEE/CVF International Conference on Computer Vision (pp. 5527-5537).

[4] Multimodal Motion Conditioned Diffusion Model for Skeleton-based Video Anomaly Detection. In Proceedings of the IEEE/CVF International Conference on Computer Vision (pp. 10318-10329).

[5] Fascinating supervisory signals and where to find them: Deep anomaly detection with scale learning. In Proceedings of the 40th International Conference on Machine Learning, ICML’23

[6] Deep isolation forest for anomaly detection. IEEE Transactions on Knowledge and Data Engineering, pp. 1–14, 2023a.

---

### Meta-Review · Area_Chair_Mf7F · 2023-12-08

**Metareview:**

The authors present a simplified diffusion model for anomaly detection that remedies the need for the reverse diffusion (reconstruction) and instead aims to model the distribution of noisy instances over the diffusion process. The paper is technically interesting and provides empirical results on several data sets.

**Justification For Why Not Higher Score:**

Technically not overwhelmingly novel but solid. Reviewers hassled with empirical performance. Nevertheless, discarding the 'decoding diffusion' and modeling distribution of 'encodings' appears a nice insight and may inspire others.

**Justification For Why Not Lower Score:**

All reviewers recommend acceptance.

---

### Decision · Program_Chairs · 2024-01-16

Accept (spotlight)